# ON THE $O(1/T)$ CONVERGENCE OF ALTERNATING GRADIENT DESCENT–ASCENT IN BILINEAR GAMES

**Tianlong Nan**
IEOR Department, Columbia University
New York, NY 10027, USA
tianlong.nan@columbia.edu

**Shuvomoy Das Gupta**
CMOR Department, Rice University
Houston, TX 77005, USA
sd158@rice.edu

**Garud Iyengar**
IEOR Department, Columbia University
New York, NY 10027, USA
garud@ieor.columbia.edu

**Christian Kroer**
IEOR Department, Columbia University
New York, NY 10027, USA
christian.kroer@columbia.edu

## ABSTRACT

We study the alternating gradient descent-ascent (AltGDA) algorithm in two-player zero-sum games. Alternating methods, where players take turns to update their strategies, have long been recognized as simple and practical approaches for learning in games, exhibiting much better numerical performance than their simultaneous counterparts. However, our theoretical understanding of alternating algorithms remains limited, and results are mostly restricted to the unconstrained setting. We show that for two-player zero-sum games that admit an interior Nash equilibrium, AltGDA converges at an $O(1/T)$ ergodic convergence rate when employing a small constant stepsize. This is the first result showing that alternation improves over the simultaneous counterpart of GDA in the constrained setting. For games without an interior equilibrium, we show an $O(1/T)$ local convergence rate with a constant stepsize that is independent of any game-specific constants. In a more general setting, we develop a performance estimation programming (PEP) framework to jointly optimize the AltGDA stepsize along with its worst-case convergence rate. The PEP results indicate that AltGDA may achieve an $O(1/T)$ convergence rate for a finite horizon $T$, whereas its simultaneous counterpart appears limited to an $O(1/\sqrt{T})$ rate.

## 1 INTRODUCTION

No-regret learning is one of the premier approaches for computing game-theoretic equilibria in multi-agent games. It is the primary method employed for solving extremely large-scale games, and was used for computing superhuman poker AIs (Bowling et al., 2015; Moravčík et al., 2017; Brown & Sandholm, 2018; 2019), as well as human-level AIs for Stratego (Perolat et al., 2022) and Diplomacy (FAIR et al., 2022).

In theory it is known that no-regret learning dynamics can converge to a Nash equilibrium at a rate of $O(1/T)$ through the use of *optimistic* learning dynamics, such as optimistic gradient descent-ascent or optimistic multiplicative weights (Rakhlin & Sridharan, 2013a;b; Syrgkanis et al., 2015). Nonetheless, the practice of solving large games has mostly focused on theoretically slower methods that guarantee only an $O(1/\sqrt{T})$ convergence rate in the worst case, notably the CFR regret decomposition framework (Zinkevich et al., 2007) combined with variants of the *regret matching* algorithm (Hart & Mas-Colell, 2000; Tammelin, 2014; Farina et al., 2021). A critical "trick" for achieving fast practical performance with these methods is the idea of *alternation*, whereby the regret minimizers for the two players take turns updating their strategies and observing performance, rather than the simultaneous strategy updates traditionally employed in the classical folk-theorem that reduces Nash equilibrium computation in a two-player zero-sum game to a regret minimization problem in repeated play.

Initially, alternation was employed as a numerical trick that greatly improved performance (e.g., in Tammelin et al. (2015)), and was eventually shown not to *hurt* performance in theory (Farina et al., 2019; Burch et al., 2019). Yet its great practical performance begs the question of whether alternation provably *helps* performance. The first such result in a game context (and more generally for *constrained* bilinear saddle-point problems), was given by Wibisono et al. (2022), where they show that alternating *mirror descent* with a Legendre regularizer guarantees $O(T^{1/3})$ regret, and thus $O(1/T^{2/3})$ convergence to equilibrium. This bound was later tightened by Katona et al. (2024). A Legendre regularizer is, loosely speaking, one that guarantees that the updates in mirror descent never touch the boundary. This is satisfied by the entropy regularizer, which leads to the multiplicative weights algorithm, but not by the Euclidean regularizer in the constrained setting, and thus not for alternating gradient descent-ascent (AltGDA). In practice, AltGDA often achieves better performance than Legendre-based methods (Kroer, 2020), and the practically-successful regret-matching methods are also more akin to GDA than multiplicative weights (Farina et al., 2021).

In spite of recent progress on alternation, it remains an open question whether AltGDA achieves a speedup over simultaneous GDA for game solving, which is known to achieve $O(1/\sqrt{T})$ convergence. More generally, it is unknown whether any of the standard learning methods that touch the boundary during play benefit from alternation. Empirically, there is evidence suggesting this may be the case. For instance, Kroer (2020) observed that the empirical performance of AltGDA exhibits $O(1/T)$ behavior on random matrix games. In this paper, we demonstrate that an $O(1/T)$ convergence rate can be achieved in various settings, thereby providing the first set of theoretical results supporting the success of AltGDA in solving games and constrained minimax problems.

**Contributions.** The contribution of this paper is three-fold.

- We show that AltGDA achieves a $O(1/T)$ rate of convergence in bilinear games with an interior Nash equilibrium. Our result shows that alternation is enough to achieve a $O(1/T)$ rate of convergence, whereas every prior result achieving a $O(1/T)$ rate of convergence for two-player zero-sum games required some form of optimism.
- We prove that AltGDA converges locally at an $O(1/T)$ rate in *any* bilinear game. Moreover, in this case, we can set a constant stepsize that is independent of any game-specific constant.
- By leveraging the techniques of performance estimation programming (PEP) framework, we numerically compute worst-case convergence bounds for AltGDA by formulating the problem as SDPs. We present the numerically optimal fixed stepsizes for each $T$, and the corresponding optimal worst-case convergence bounds. Our methodology is the first instance of stepsize optimization of such performance estimation problems for primal-dual algorithms involving linear operators.

## 2 RELATED WORK

**AltGDA in unconstrained minimax problems.** Bailey et al. (2020) studied AltGDA in unconstrained bilinear problems, and showed an $O(1/T)$ convergence rate. They also proposed a useful energy function that is a constant along the AltGDA trajectory. Proving a $O(1/T)$ convergence rate is easier in the unconstrained setting, where the pair of strategies $(\mathbf{0}, \mathbf{0})$ is guaranteed to be a Nash equilibrium no matter the payoff matrix. More discussion is given in Section 5.

Zhang et al. (2022) established local linear convergence rates for both unconstrained strongly-convex strongly-concave (SCSC) minimax problems. Lee et al. (2024) studied AltGDA for unconstrained smooth SCSC minimax problems. More recently, Feng et al. (2025) studied AltGDA with momentum in unconstrained smooth minimax problems.

**AltGDA in constrained bilinear games.** From the game theory context, the constrained setting is more important, because it is the one capturing standard solution concepts such as Nash equilibrium. Prior to our work, we are not aware of any theoretical results showing that alternation improves GDA compared to the simultaneous algorithm in constrained minimax problems. See also Orabona (2019) for an extended discussion of the history of alternation in game solving and optimization.

As a common technique in game-solving, alternation has been investigated in settings related to ours. Mertikopoulos et al. (2018) showed that the continuous-time dynamics (in their Section A.2) achieve

an $O(1/T)$ average regret bound. Cevher et al. (2023) study a novel no-regret learning setting that captures the type of regret sequences observed in alternating self play in two-player zero-sum games. They show a $O(T^{1/3})$ no-regret learning result for a somewhat complicated learning algorithm for the simplex, and show that $O(\log T)$ regret is possible when the simplex has two actions, through a reduction to learning on the Euclidean ball, where they show the same bound. Hait et al. (2025) generalize this result to any convex-concave zero-sum games. Recently, Lazarsfeld et al. (2025) prove a lower bound of $\Omega(1/\sqrt{T})$ for alternation in the context of fictitious play.

**Optimistic methods in constrained bilinear games.** As mentioned earlier, it is well known that an $O(1/T)$ convergence rate can be achieved by certain variants of extragraidient methods (Korpelevich, 1976) and optimistic GDA (Popov, 1980) (here simply called optimistic methods). For constrained bilinear games, the $O(1/T)$ convergence rate has been established by a long line of work for various optimistic methods, including Mirror-Prox (Nemirovski, 2004), Dual Extrapolation (Nesterov, 2007), Primal-Dual Hybrid Gradient (Chambolle & Pock, 2011), Accelerated Primal-Dual (Chen et al., 2014), and Adaptive Mirror-Prox (Antonakopoulos et al., 2019), among others. We emphasize that AltGDA is not theoretically superior to optimistic methods in general; rather, it is an appealing algorithmic choice that is widely used in practice.

**PEP for primal-dual algorithms.** There has been prior work using the SDP-based PEP framework to evaluate the performance of primal-dual algorithms involving a linear operator with known stepsize (Bousselmi et al., 2024; Zamani et al., 2024; Krivchenko et al., 2024), but they do not investigate optimizing the stepsize to get the best convergence bound. Das Gupta et al. (2024); Jang et al. (2023) proposed for optimizing stepsizes of first-order methods for minimizing a single function or sum of two functions, by using spatial branch-and-bound based frameworks. Unfortunately such frameworks can become prohibitively slow when it comes to optimizing primal-dual algorithms because of additional nonconvex coupling between the variables in the presence of the linear operator.

**Notation.** For vectors $\boldsymbol{a}, \boldsymbol{b} \in \mathbb{R}^d$, we write $\boldsymbol{a}^\top \boldsymbol{b}$ or $\langle \boldsymbol{a}, \boldsymbol{b} \rangle$ for the standard inner product and $\|\boldsymbol{a}\| = \sqrt{\boldsymbol{a}^\top \boldsymbol{a}}$ for the Euclidean norm. The spectral norm of a matrix $A$ is denoted by $\|A\|_2 = \sigma_{\max}(A)$, where $\sigma_{\max}(A)$ represents the largest singular value of $A$. We use $\|\boldsymbol{a}\|_1$ and $\|\boldsymbol{a}\|_2$ to denote $\ell_1$ and $\ell_2$ vector norms, respectively. Projection onto a compact convex set $\mathcal{X}$ is denoted by $\Pi_{\mathcal{X}}(x) = \operatorname{argmin}_{z \in \mathcal{X}} \|x - z\|_2^2$. We write $[d] = \{1, \ldots, d\}$ for any positive integer $d$.

## 3 PRELIMINARIES

We consider bilinear saddle point problems (SPPs) of the form

$$\min_{\boldsymbol{x} \in \mathcal{X}} \max_{\boldsymbol{y} \in \mathcal{Y}} \boldsymbol{y}^\top A \boldsymbol{x}, \tag{1}$$

where $\mathcal{X} \subseteq \mathbb{R}^n$ and $\mathcal{Y} \subseteq \mathbb{R}^m$ are compact convex sets and $A$ is an $n \times m$ matrix. We are especially interested in *bilinear two-player zero-sum games* (or *matrix games*), where $\mathcal{X} = \Delta_n = \{\boldsymbol{x} \in \mathbb{R}_+^n \mid \sum_{i=1}^n x_i = 1\}$ and $\mathcal{Y} = \Delta_m = \{\boldsymbol{y} \in \mathbb{R}_+^m \mid \sum_{j=1}^m y_j = 1\}$ are the probability simplexes. In the game context, Eq. (1) corresponds to a game in which two players (called the $x$-player and $y$-player) choose their strategies from decision sets $\Delta_n$ and $\Delta_m$, and the matrix $A$ encodes the payoff of the $y$ player (which the $x$ player wants to minimize).

We say $(\boldsymbol{x}^*, \boldsymbol{y}^*) \in \Delta_n \times \Delta_m$ is a *Nash equilibrium* (NE) or saddle point of the game if it satisfies

$$\boldsymbol{y}^\top A \boldsymbol{x}^* \leq (\boldsymbol{y}^*)^\top A \boldsymbol{x}^* \leq (\boldsymbol{y}^*)^\top A \boldsymbol{x} \quad \forall \boldsymbol{x} \in \Delta_n, \boldsymbol{y} \in \Delta_m. \tag{2}$$

By von Neumann's min-max theorem (v. Neumann, 1928), in every bilinear two-player zero-sum game, there always exists a Nash equilibrium, and a unique value $\nu^* := \min_{\boldsymbol{x} \in \Delta_n} \max_{\boldsymbol{y} \in \Delta_m} \boldsymbol{y}^\top A \boldsymbol{x} = \max_{\boldsymbol{y} \in \Delta_m} \min_{\boldsymbol{x} \in \Delta_n} \boldsymbol{y}^\top A \boldsymbol{x}$ which is called the *value of the game*. Furthermore, the set of NE is convex, and $\nu^* = \min_i (A^\top \boldsymbol{y}^*)_i = \max_j (A \boldsymbol{x}^*)_j$. We call an NE $(x^*, y^*)$ an *interior NE* if $x_i^* > 0$ for all $i \in [n]$ and $y_j^* > 0$ for all $j \in [m]$.

For a strategy pair $(\tilde{\boldsymbol{x}}, \tilde{\boldsymbol{y}}) \in \Delta_n \times \Delta_m$, we use the *duality gap* (or *saddle-point residual*) to measure the proximity to NE:

$$\mathtt{DualityGap}(\tilde{\boldsymbol{x}}, \tilde{\boldsymbol{y}}) := \left( \sup_{\boldsymbol{y} \in \Delta_m} \boldsymbol{y}^\top A \tilde{\boldsymbol{x}} - \tilde{\boldsymbol{y}}^\top A \tilde{\boldsymbol{x}} \right) + \left( \tilde{\boldsymbol{y}}^\top A \tilde{\boldsymbol{x}} - \inf_{\boldsymbol{x} \in \Delta_n} \tilde{\boldsymbol{y}}^\top A \boldsymbol{x} \right)$$

$$= \sup_{\boldsymbol{x} \in \Delta_n, \boldsymbol{y} \in \Delta_m} \left( \boldsymbol{y}^\top A \tilde{\boldsymbol{x}} - \tilde{\boldsymbol{y}}^\top A \boldsymbol{x} \right). \tag{Duality Gap}$$

---

**Algorithm 1** Alternating Gradient Descent-Ascent (AltGDA)

---

**input:** Number of iterations $T$, step size $\eta > 0$
**initialize:** $(\boldsymbol{x}^0, \boldsymbol{y}^0) \in \mathcal{X} \times \mathcal{Y}$
**for** $t = 0, \ldots, T - 1$ **do**
    $\boldsymbol{x}^{t+1} = \Pi_{\mathcal{X}}(\boldsymbol{x}^t - \eta A^{\top} \boldsymbol{y}^t)$
    $\boldsymbol{y}^{t+1} = \Pi_{\mathcal{Y}}(\boldsymbol{y}^t + \eta A \boldsymbol{x}^{t+1})$
**end for**
**output:** $(\frac{1}{T} \sum_{t=1}^{T} \boldsymbol{x}^t, \frac{1}{T} \sum_{t=1}^{T} \boldsymbol{y}^t) \in \mathcal{X} \times \mathcal{Y}$

---

By definition, $\texttt{DualityGap}(\tilde{\boldsymbol{x}}, \tilde{\boldsymbol{y}}) \geq 0$ for any $(\tilde{\boldsymbol{x}}, \tilde{\boldsymbol{y}}) \in \Delta_n \times \Delta_m$. Moreover, $\texttt{DualityGap}(\tilde{\boldsymbol{x}}, \tilde{\boldsymbol{y}}) = 0$ if and only if $(\tilde{\boldsymbol{x}}, \tilde{\boldsymbol{y}})$ is a Nash equilibrium.

For general bilinear SPPs as in Eq. (1), $\texttt{DualityGap}(\tilde{\boldsymbol{x}}, \tilde{\boldsymbol{y}}) = \sup_{\boldsymbol{x} \in \mathcal{X}, \boldsymbol{y} \in \mathcal{Y}} \left( \boldsymbol{y}^{\top} A \tilde{\boldsymbol{x}} - \tilde{\boldsymbol{y}}^{\top} A \boldsymbol{x} \right)$. A point $(\tilde{\boldsymbol{x}}, \tilde{\boldsymbol{y}}) \in \mathcal{X} \times \mathcal{Y}$ is called an $\varepsilon$-saddle point if $\texttt{DualityGap}(\tilde{\boldsymbol{x}}, \tilde{\boldsymbol{y}}) \leq \varepsilon$.

**AltGDA and SimGDA.** For solving Eq. (1), the alternating and simultaneous GDA (AltGDA and SimGDA) algorithms are simple and commonly used in practice. In AltGDA, the players take turns updating their strategies by performing a single *projected gradient descent* update based on their expected payoff for the current state. We state the AltGDA algorithm in Algorithm 1. In contrast, SimGDA updates both players' strategies simultaneously, using the expected payoff evaluated at the previous state. Compared to Algorithm 1, the inner projected gradient descent takes the form

$$\boldsymbol{x}^{t+1} = \Pi_{\mathcal{X}}(\boldsymbol{x}^t - \eta A^{\top} \boldsymbol{y}^t), \quad \boldsymbol{y}^{t+1} = \Pi_{\mathcal{Y}}(\boldsymbol{y}^t + \eta A \boldsymbol{x}^t). \qquad \text{(SimGDA Updates)}$$

## 4 PERFORMANCE ESTIMATION PROGRAMMING FOR ALTGDA

In this section, we present a computer-assisted methodology based on the PEP framework (Drori & Teboulle, 2014; Taylor et al., 2017b;a) along with results on PEP with linear operators (Bousselmi et al., 2024) to compute the tightest convergence rate of AltGDA numerically.

**Computing the worst-case performance with a known $\eta$.** We consider bilinear SPPs over compact convex sets as described by (1). The worst-case performance (or complexity) of AltGDA corresponds to the number of oracle calls the algorithm needs to find an $\varepsilon$-saddle point. Equivalently, we can measure AltGDA's worst-case performance by looking at the duality gap of the averaged iterates, i.e., $\texttt{DualityGap}(\frac{1}{T} \sum_{k=1}^{T} \boldsymbol{x}^t, \frac{1}{T} \sum_{k=1}^{T} \boldsymbol{y}^t) = \max_{\boldsymbol{x} \in \mathcal{X}, \boldsymbol{y} \in \mathcal{Y}} \left( \boldsymbol{y}^{\top} A (\frac{1}{T} \sum_{k=1}^{T} \boldsymbol{x}^t) - (\frac{1}{T} \sum_{k=1}^{T} \boldsymbol{y}^t)^{\top} A \boldsymbol{x} \right)$, where $\{(\boldsymbol{x}^t, \boldsymbol{y}^t)\}_{1 \leq t \leq T}$ are generated by AltGDA with stepsize $\eta$.

To keep the worst-case performance bounded, we need to bound the norm of $A$ and the radii of the compact convex sets $\mathcal{X}, \mathcal{Y}$. In particular, without loss of generality, we assume $\sigma_{\max}(A) \leq 1$. Let $R_x$ and $R_y$ be the radii of the sets $\mathcal{X}$ and $\mathcal{Y}$, respectively. Then, without loss of generality, we can set $R := \max\{R_x, R_y\} = 1$. This is due to a scaling argument: for any other finite value of $R$, the new performance measure will be $R^2 \times$ (worst-case performances for $R = 1$).

Let $\texttt{AltGDA}(\eta, \boldsymbol{x}^0, \boldsymbol{y}^0)$ denote the sequence of iterates generated by Algorithm 1 with stepsize $\eta$ starting from initial point $(\boldsymbol{x}^0, \boldsymbol{y}^0)$. Then, we can compute the worst-case performance of AltGDA with stepsize $\eta > 0$ and total iteration $T$ by the following *infinite-dimensional* nonconvex optimization problem:

$$\mathcal{P}_T(\eta) := \left( \begin{array}{ll} \underset{\substack{\{(\boldsymbol{x}^t, \boldsymbol{y}^t)\}_{0 \leq t \leq T} \subseteq \mathbb{R}^n \times \mathbb{R}^m, \\ \mathcal{X} \times \mathcal{Y} \subseteq \mathbb{R}^n \times \mathbb{R}^m, \\ A \in \mathbb{R}^{m \times n}, m, n \in \mathbb{N}.}}{\text{maximize}} & \dfrac{1}{T} \sum_{t=1}^{T} \left( \boldsymbol{y}^{\top} A \boldsymbol{x}^t - (\boldsymbol{y}^t)^{\top} A \boldsymbol{x} \right) \\[2em] \text{subject to} & \mathcal{X} \text{ is a convex compact set in } \mathbb{R}^n \text{ with radius } 1, \\ & \mathcal{Y} \text{ is a convex compact set in } \mathbb{R}^m \text{ with radius } 1, \\ & \sigma_{\max}(A) \leq 1, \\ & \{(\boldsymbol{x}^t, \boldsymbol{y}^t)\}_{1 \leq t \leq T} = \texttt{AltGDA}(\eta, \boldsymbol{x}^0, \boldsymbol{y}^0), \\ & (\boldsymbol{x}^0, \boldsymbol{y}^0), (\boldsymbol{x}, \boldsymbol{y}) \in \mathcal{X} \times \mathcal{Y}. \end{array} \right) \qquad \text{(INNER)}$$

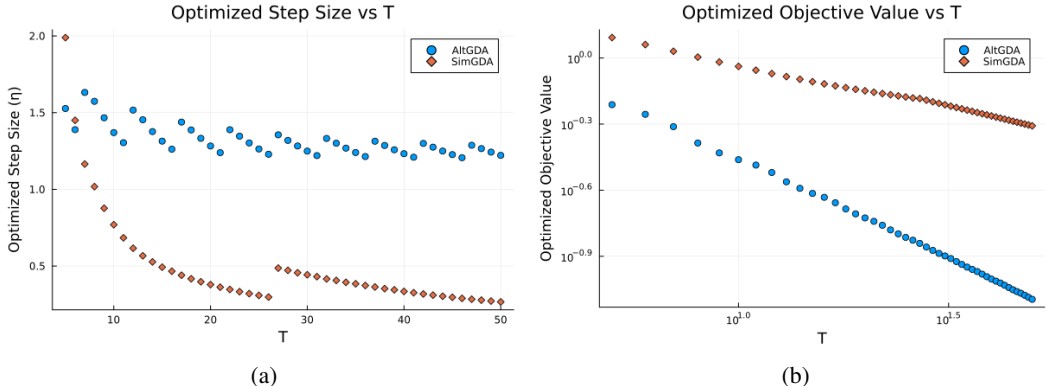

Figure 1: Optimized stepsizes and corresponding optimized objective values for $T = 5, 6, \ldots, 50$ via PEP. The left plot shows the optimized stepsizes. The optimized objective value in the right plot denotes the worst-case performance measure (i.e., duality gap of the averaged iterates) corresponding to the optimized stepsizes on log scale.

Problem (INNER) is intractable because it contains infinite-dimensional objects such as $\mathcal{X}, \mathcal{Y}$, and $A$ where dimensions $n, m$ are also variables. Nevertheless, for our setup, all possible iterates and their associated gradients up to $T$ can be captured by a finite collection of interpolation inequalities. These inequalities fully encode the entire class of admissible instances, thereby allowing (INNER) to be reduced to a finite-dimensional SDP, as elaborated in Appendix E. This SDP is also free from the dimensions $n$ and $m$ under a large-scale assumption. In other words, computing $\mathcal{P}_T(\eta)$ numerically provides us a tight dimension-independent convergence bound for AltGDA for a given $\eta$ and $T$.

**Best convergence rate with optimized $\eta$.** For a fixed $T$, the best convergence rate of AltGDA can be found by solving $\mathcal{P}_T^* = \text{minimize}_\eta \mathcal{P}_T(\eta)$. To solve this problem, we perform a grid-like search on the stepsize $\eta$ and solve the corresponding SDP for each of the finitely-many $\eta$ choice:

- Step 1: Set an initial search range $[\eta_{\min}, \eta_{\max}]$;
- Step 2: Pick $n$ points within this range such that their reciprocal is equally spaced, i.e., $n$ candidate stepsizes s.t. $\eta_{\min} = \eta_1 \leq \cdots \leq \eta_n = \eta_{\max}$ and $\frac{1}{\eta_1} - \frac{1}{\eta_2} = \cdots = \frac{1}{\eta_{n-1}} - \frac{1}{\eta_n}$;[1]
- Step 3: Compute the worst-case performance corresponding to each candidate stepsize, and denote the best stepsize as $\eta^*$;
- Step 4: Set an updated search range: $[\eta_{\min}, \eta_{\max}] \leftarrow [\eta^* - \alpha \frac{\eta_{\max} - \eta_{\min}}{n-1}, \eta^* + \alpha \frac{\eta_{\max} - \eta_{\min}}{n-1}]$;
- Step 5: Repeat Step 2 and Step 4 until $\eta_{\max} - \eta_{\min} \leq \varepsilon_\eta$.

Here, $\eta_{\min}, \eta_{\max}, n, \alpha, \varepsilon_\eta$ are hyperparameters to be fine-tuned. In our numerical experiments, we set $n = 20$, $\alpha = 1$ and $\varepsilon_\eta = 10^{-3}$; and fine-tuned $\eta_{\min}, \eta_{\max}$ based on different algorithms and time horizon $T$. Because the precision of the grid search $\varepsilon_\eta$ is not equal to exactly zero, we call our computed stepsize to be *optimized* rather than *optimal*.

**Results and discussion.** See Fig. 1 for the optimized stepsizes and corresponding worst-case performance. We also provide the data values to generate Fig. 1 in Appendix E.1.

From Fig. 1a, we observe a structured sequence of optimized stepsizes for AltGDA. The origin of this periodic optimized stepsize pattern is interesting in itself and worth exploring. Moreover, this phenomenon indicates the possibility of improving the convergence rate by employing iteration-dependent structured stepsize schedules in the minimax problems. Beyond this, we observe that the decay rate of the stepsizes scales as $O(1/(\log T)^\alpha)$ for some $\alpha > 0$, which indicates that the optimal convergence rate may hold with "nearly-constant" stepsizes. Fig. 1b shows that the optimized duality gap approaches a $O(1/T)$ convergence rate as $T$ increases. This suggests that AltGDA obtains a $O(1/T)$ convergence rate after a short transient phase. This finding also raises an interesting question about the origin of the initial convergence phase. In contrast, SimGDA exhibits a $O(1/\sqrt{T})$ convergence rate, even with an optimized stepsize schedule.

---

[1] By taking non-equally spaced points, we place greater emphasis on exploring the range of smaller step sizes.

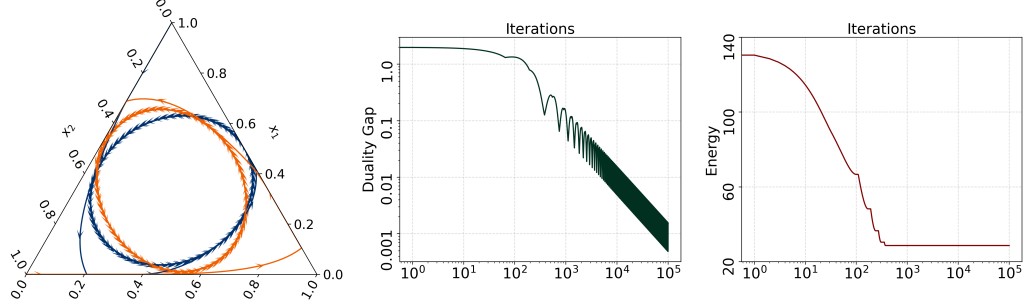

Figure 2: Numerical results on the rock-paper-scissor game. From left to right, we show the trajectories of the AltGDA iterates (in ternary plots), the changes in duality gaps, and the evolution of the energy functions.

The PEP literature provides us a potential solution to theoretically prove the tightest convergence rate for a given algorithm (Drori & Teboulle, 2014; Taylor et al., 2017b;a). A proof in this framework requires discovering analytical solutions to the optimal dual variables of the underlying SDPs, including proving semi-definiteness of the SDP matrices (Goujaud et al., 2023). For AltGDA, our attempts at a proof via this route lead to us observing rather intricate optimal dual variable structures that appear to make the proof difficult. As an alternative, we will show in the following sections that more classical proof approaches, with some interesting variations, can be used to show $O(1/T)$ convergence in several settings.

# 5 $O(1/T)$ CONVERGENCE RATE WITH AN INTERIOR NASH EQUILIBRIUM

In this section, we establish an $O(1/T)$ convergence rate of AltGDA for bilinear two-player zero-sum games that admit an interior NE. We begin by presenting the motivation and interpretation of the proof, followed by a sketch of the formal proof.

## 5.1 MOTIVATION AND INTERPRETATION

We will start by presenting some new observations about the trajectory generated by AltGDA, which is the inspiration for our proof. In contrast to the unconstrained setting (Bailey et al., 2020), the iterates of AltGDA do not necessarily cycle from the beginning, even in the presence of an interior NE. Fig. 2 shows the numerical behavior of AltGDA in the rock-paper-scissors game, which is a bilinear game admitting an interior NE. The left plot shows that the trajectories of the players' strategies exhibit two distinct phases. In the first phase, the orbit hits the boundary of the simplex and is "pushed back" into its interior. In the second phase, the orbit settles into a state where it cycles within the relative interior of the simplex and no longer touches the boundary.

We observe that this two-phase behavior can be captured by the following energy function with respect to any interior NE $(\boldsymbol{x}^*, \boldsymbol{y}^*)$ :[2]

$$\mathcal{E}(\boldsymbol{x}^t, \boldsymbol{y}^t) := \|\boldsymbol{x}^t - \boldsymbol{x}^*\|_2^2 + \|\boldsymbol{y}^t - \boldsymbol{y}^*\|_2^2 - \eta(\boldsymbol{y}^t)^\top A \boldsymbol{x}^t. \qquad \text{(Energy)}$$

We plot the evolution of $\mathcal{E}(\boldsymbol{x}^t, \boldsymbol{y}^t)$ on the right of Fig. 2. Interestingly, we find a correspondence between the "collision and friction" of the trajectory and the "energy decay" of $\mathcal{E}(\boldsymbol{x}^t, \boldsymbol{y}^t)$. In particular, the energy function admits a meaningful physical interpretation—it decays whenever the trajectory collides with and rubs against the boundary of the simplex.

Moreover, in the middle of Fig. 2, we see the duality gap decreases slowly when the energy decreases, and shrinks at an $O(1/T)$ rate after the energy function remains constant. This indicates the connection between the energy function and the convergence rate of the averaged iterate, which forms the foundation of our proof.

---

[2]While the energy function is dependent on the stepsize $\eta$, we write $\mathcal{E}(\boldsymbol{x}^t, \boldsymbol{y}^t)$ rather than $\mathcal{E}(\eta, \boldsymbol{x}^t, \boldsymbol{y}^t)$ to reduce the notational burden.

## 5.2 CONVERGENCE ANALYSIS

In classical optimization analysis, convergence guarantees are often established using some potential function: one first establishes an inequality showing that the duality gap at an arbitrary iteration is bounded by the change of a potential function plus some *summable* term, then telescopes this inequality to obtain the convergence rate. In contrast, our proof works with an inequality involving the duality gap at two successive iterates, as shown in the following lemma. The complete proofs in this section are deferred to Appendix B.

**Lemma 1.** *Let $\{(\boldsymbol{x}^t, \boldsymbol{y}^t)\}_{t=0,1,\dots}$ be a sequence of iterates generated by Algorithm 1 with $\eta > 0$. Then, for any $(\boldsymbol{x}, \boldsymbol{y}) \in \Delta_n \times \Delta_m$, we have*

$$\eta \left( \boldsymbol{y}^\top A \boldsymbol{x}^t - (\boldsymbol{y}^t)^\top A \boldsymbol{x} \right) \leq \psi_t(\boldsymbol{x}, \boldsymbol{y}) - \psi_{t+1}(\boldsymbol{x}, \boldsymbol{y}) + \eta \langle -A^\top \boldsymbol{y}^t , \boldsymbol{x}^{t+1} - \boldsymbol{x}^t \rangle$$
$$- \frac{1}{2} \| \boldsymbol{x}^{t+1} - \boldsymbol{x}^t \|_2^2 - \frac{1}{2} \| \boldsymbol{y}^{t+1} - \boldsymbol{y}^t \|_2^2, \ \text{for } t \geq 1, \qquad (3)$$

$$\eta \left( \boldsymbol{y}^\top A \boldsymbol{x}^{t+1} - (\boldsymbol{y}^{t+1})^\top A \boldsymbol{x} \right) \leq \phi_t(\boldsymbol{x}, \boldsymbol{y}) - \phi_{t+1}(\boldsymbol{x}, \boldsymbol{y}) + \eta \langle A \boldsymbol{x}^{t+1} , \boldsymbol{y}^{t+1} - \boldsymbol{y}^t \rangle$$
$$- \frac{1}{2} \| \boldsymbol{x}^{t+1} - \boldsymbol{x}^t \|_2^2 - \frac{1}{2} \| \boldsymbol{y}^{t+1} - \boldsymbol{y}^t \|_2^2, \ \text{for } t \geq 0, \qquad (4)$$

*where $\phi_t(\boldsymbol{x}, \boldsymbol{y}) := \frac{1}{2} \| \boldsymbol{x}^t - \boldsymbol{x} \|_2^2 + \frac{1}{2} \| \boldsymbol{y}^t - \boldsymbol{y} \|_2^2 + \eta (\boldsymbol{y}^t)^\top A \boldsymbol{x}$ and $\psi_t(\boldsymbol{x}, \boldsymbol{y}) := \frac{1}{2} \| \boldsymbol{x}^t - \boldsymbol{x} \|_2^2 + \frac{1}{2} \| \boldsymbol{y}^{t-1} - \boldsymbol{y} \|_2^2 - \frac{1}{2} \| \boldsymbol{y}^t - \boldsymbol{y}^{t-1} \|_2^2$.*

The main challenge in the proof is determining whether the sum of the residual terms on the right-hand sides of Eqs. (3) and (4) are summable, i.e., $\sum_{t=0}^\infty r_t < \infty$ where

$$r_t := \eta \langle -A^\top \boldsymbol{y}^t , \boldsymbol{x}^{t+1} - \boldsymbol{x}^t \rangle + \eta \langle A \boldsymbol{x}^{t+1} , \boldsymbol{y}^{t+1} - \boldsymbol{y}^t \rangle - \| \boldsymbol{x}^{t+1} - \boldsymbol{x}^t \|_2^2 - \| \boldsymbol{y}^{t+1} - \boldsymbol{y}^t \|_2^2$$
$$= \langle -\eta A^\top \boldsymbol{y}^t - \boldsymbol{x}^{t+1} + \boldsymbol{x}^t , \boldsymbol{x}^{t+1} - \boldsymbol{x}^t \rangle + \langle \eta A \boldsymbol{x}^{t+1} - \boldsymbol{y}^{t+1} + \boldsymbol{y}^t , \boldsymbol{y}^{t+1} - \boldsymbol{y}^t \rangle. \qquad (5)$$

In the unconstrained case, we have $r_t \equiv 0$ for all $t \geq 0$, and hence the $O(1/T)$ convergence rate follows directly. In contrast, in the constrained case, the first-order optimality conditions of the projection operators imply that $r_t \geq 0$. Therefore, it is not immediate whether $r_t$ is summable. To handle this, we exploit the connection between energy decay and the convergence rate of the duality gap, as shown in Fig. 2. In particular, when an interior NE exists, we show that the residual $r_t$ can be bounded by the decay of the energy function, as established in the following lemma.

**Lemma 2.** *Assume that the bilinear game admits an interior NE. Let $\{(\boldsymbol{x}^t, \boldsymbol{y}^t)\}_{t=0,1,\dots}$ be a sequence of iterates generated by Algorithm 1 with $\eta \leq \frac{1}{\|A\|_2} \min\{\min_{i \in [n]} x_i^*, \min_{j \in [m]} y_j^*\}$. Then, we have $0 \leq r_t \leq \mathcal{E}(\boldsymbol{x}^t, \boldsymbol{y}^t) - \mathcal{E}(\boldsymbol{x}^{t+1}, \boldsymbol{y}^{t+1})$ for all $t \geq 0$.*

By combining Lemmas 1 and 2, telescoping over $t = 0, 1, \dots, T$, and using the boundedness of $\phi, \psi, \mathcal{E}$, we obtain the $O(1/T)$ convergence rate.

**Theorem 1.** *Assume that the bilinear game admits an interior NE. Let $\{(\boldsymbol{x}^t, \boldsymbol{y}^t)\}_{t=0,1,\dots}$ be a sequence of iterates generated by Algorithm 1 with $\eta \leq \frac{1}{\|A\|_2} \min\{\min_{i \in [n]} x_i^*, \min_{j \in [m]} y_j^*\}$. Then, we have*

$$\texttt{DualityGap} \left( \frac{1}{T} \sum_{t=1}^T \boldsymbol{x}^t, \frac{1}{T} \sum_{t=1}^T \boldsymbol{y}^t \right) \leq \frac{9 + 4\eta \|A\|_2}{\eta T}.$$

Theorem 1 provides the first finite regret and $O(1/T)$ convergence rate result for AltGDA in constrained minimax problems. Although such a result has been known for several years in the unconstrained setting (Bailey et al., 2020), no better than $O(1/\sqrt{T})$ convergence rate has been established in the constrained case. Even for the broader class of alternating mirror descent algorithms, no instantiations of the algorithm were known to achieve a $O(1/T)$ convergence rate—despite having been observed numerically (Wibisono et al., 2022; Katona et al., 2024; Kroer, 2025).

Although our primary goal is to develop the theoretical foundations for AltGDA, we also include additional results relevant for practice in Appendix C. For instance, we present an adaptive step-size rule that does not require knowledge of the interior NE yet still achieves an $O(1/T)$ convergence rate in Appendix C.2.

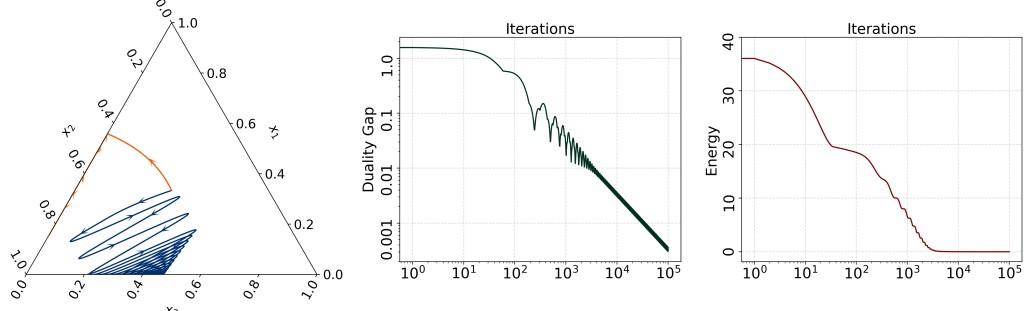

Figure 3: Numerical results on a $3 \times 3$ random matrix instance without an interior NE. The experimental setup is the same as in Fig. 2.

The trajectory of AltGDA exhibits more intricate behavior when the game does not have an interior NE. As shown in Fig. 3, the trajectory tends to approach the face of the simplex spanned by the NE with maximal support, which we refer to as the *essential face*. However, the trajectory does not converge to the essential face monotonically—it can leave the face after touching it. This non-monotonicity persists even after many iterations in our experiments, and, accordingly, the energy may increase on some iterations. In this case, the difference of the energy no longer yields an upper bound for $r_t$ as in Lemma 2.

## 6 LOCAL $O(1/T)$ CONVERGENCE RATE

As previously discussed, our $O(1/T)$ convergence rate only applies to games with an interior NE due to non-monotonicity of the energy function in the general case. Nevertheless, even without an interior NE, we show that in a local neighborhood of an NE, we can prove an $O(1/T)$ convergence rate with a constant stepsize. Notably, this stepsize is independent of any game-specific parameters.

Let $(\boldsymbol{x}^*, \boldsymbol{y}^*)$ be a NE with maximal support. Then we first partition each player's action set into two subsets: $I^* = \{i \in [n] \mid x_i^* > 0\}$ and $[n] \setminus I^*$; $J^* = \{j \in [m] \mid y_j^* > 0\}$ and $[m] \setminus J^*$, and introduce the following parameter measuring the gap between the suboptimal payoffs to the optimal payoff for both players[3]:

$$\delta := \min \left\{ \min_{i \notin I^*} \frac{(A^\top \boldsymbol{y}^*)_i - \nu^*}{\|A\|_2}, \min_{j \notin J^*} \frac{\nu^* - (A\boldsymbol{x}^*)_j}{\|A\|_2}, \min_{i \in I^*} x_i^*, \min_{j \in J^*} y_j^* \right\}. \quad (6)$$

If the equilibrium has full support, then $\delta > 0$ is the minimum probability of any action played in the full-support equilibrium. If there is no full-support equilibrium, then Mertikopoulos et al. (2018, Lemma C.3) show that for a maximum-support equilibrium we have that $\delta > 0$. Define $r_x = \min\{\frac{|I^*|}{n-|I^*|}, n\}$, $r_y = \min\{\frac{|J^*|}{m-|J^*|}, m\}$. and a local region[4]

$$S := \left\{ (\boldsymbol{x}, \boldsymbol{y}) \Big| \|\boldsymbol{x} - \boldsymbol{x}^*\|_2 \le \frac{\delta}{4}, \|\boldsymbol{y} - \boldsymbol{y}^*\|_2 \le \frac{\delta}{4}, \max_{i \notin I^*} x_i \le \frac{\eta\|A\|_2}{2} r_x\delta, \max_{j \notin J^*} y_j \le \frac{\eta\|A\|_2}{2} r_y\delta \right\}.$$

The following lemma establishes a separation between the entries in $I^*$ and $[n] \setminus I^*$; $J^*$ and $[m] \setminus J^*$. The complete proofs in this section are deferred to Appendix D.

**Lemma 3.** *If the current iterate $(\boldsymbol{x}, \boldsymbol{y}) \in S$, and the next iterate $(\boldsymbol{x}^+, \boldsymbol{y}^+)$ is generated by Algorithm 1 with the stepsize $\eta \le \frac{1}{2\|A\|_2}$, then we have (i) $x_i^+, x_i \ge \frac{\delta}{2}$ for all $i \in I^*$ and $y_j^+, y_j \ge \frac{\delta}{2}$ for all $j \in J^*$; (ii) $x_i^+ \le x_i$ for all $i \notin I^*$ and $y_j^+ \le y_j$ for all $j \notin J^*$.*

Next, we define an initial region:

$$S_0 := \left\{ (\boldsymbol{x}, \boldsymbol{y}) \Big| \|\boldsymbol{x} - \boldsymbol{x}^*\|_2 \le \frac{\delta}{8}, \|\boldsymbol{y} - \boldsymbol{y}^*\|_2 \le \frac{\delta}{8}, \max_{i \notin I^*} x_i \le \frac{c}{2} r_x\delta, \max_{j \notin J^*} y_j \le \frac{c}{2} r_y\delta \right\} \subset S, \quad (7)$$

---

[3]Note that the parameter $\delta$ is invariant under scaling of the payoff matrix $A$.

[4]The last two constraints are redundant when $|I^*| = n$ or $|J^*| = m$.

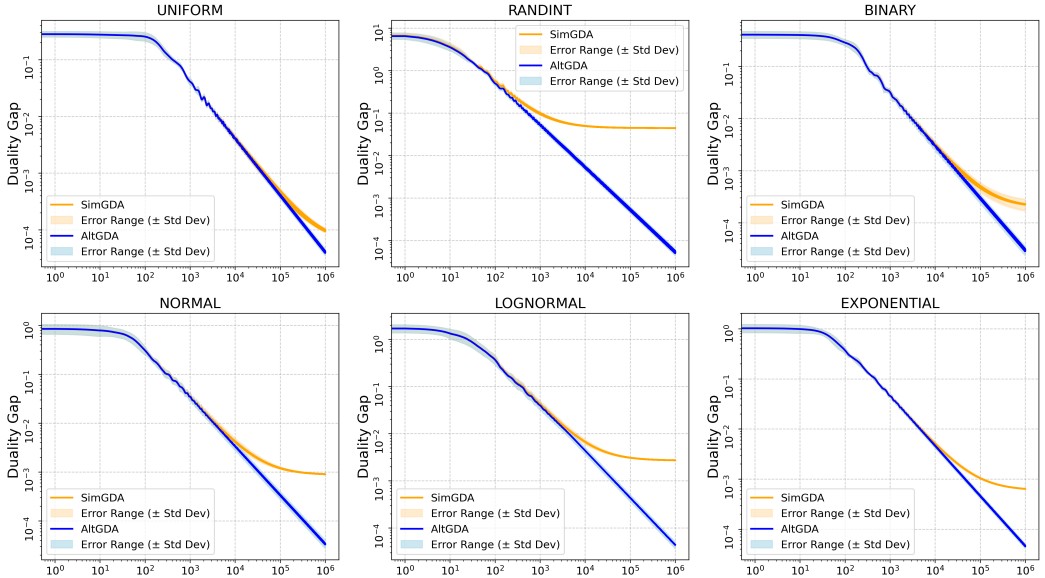

Figure 4: Numerical performances of AltGDA and SimGDA on $10 \times 20$ synthesized matrix games.

where $c = \min\{\eta\|A\|_2, \frac{\delta}{192|I^*|}, \frac{\delta}{192|J^*|}\}$. Also, for ease of presentation, we define a variant of the energy function: $\mathcal{V}(\boldsymbol{x}, \boldsymbol{y}) = \|\boldsymbol{x} - \boldsymbol{x}^*\|_2^2 + \|\boldsymbol{y} - \boldsymbol{y}^*\|_2^2 - \eta(\boldsymbol{y} - \boldsymbol{y}^*)^\top A(\boldsymbol{x} - \boldsymbol{x}^*)$.[5] In the following lemma, we prove that if we initialize AltGDA within $S_0$, then the sequence of iterates stays within $S$. With this in hand, we can derive an upper bound for the cumulative increase of the energy $\mathcal{V}$.

**Lemma 4.** *Let $\{(\boldsymbol{x}^t, \boldsymbol{y}^t)\}_{t \geq 0}$ be a sequence of iterates generated by Algorithm 1 with stepsize $\eta \leq \frac{1}{2\|A\|_2}$ and an initial point $(\boldsymbol{x}^0, \boldsymbol{y}^0) \in S_0$. Then, the iterates $\{(\boldsymbol{x}^t, \boldsymbol{y}^t)\}_{t \geq 0}$ stay within the local region $S$. Furthermore, for any $T > 0$, we have $\sum_{t=0}^{T} \left( \mathcal{V}(\boldsymbol{x}^{t+1}, \boldsymbol{y}^{t+1}) - \mathcal{V}(\boldsymbol{x}^t, \boldsymbol{y}^t) \right) \leq \frac{1}{128}\delta^2$.*

Combining this results with analogous inequalities as in Lemma 1, we obtain the local $O(1/T)$ convergence rate.

**Theorem 2.** *Let $\{(\boldsymbol{x}^t, \boldsymbol{y}^t)\}_{t \geq 0}$ be a sequence of iterates generated by Algorithm 1 with stepsize $\eta \leq \frac{1}{2\|A\|_2}$ and an initial point $(\boldsymbol{x}^0, \boldsymbol{y}^0) \in S_0$, where $S_0$ is defined in Eq. (7). Then, we have that*

$$\texttt{DualityGap}\left(\frac{1}{T}\sum_{t=1}^{T}\boldsymbol{x}^t, \frac{1}{T}\sum_{t=1}^{T}\boldsymbol{y}^t\right) \leq \frac{9 + 7\eta\|A\|_2 + (\delta^2/128)}{\eta T},$$

*where $\delta$ is defined in Eq. (6).*

## 7 NUMERICAL EXPERIMENTS

We conduct numerical experiments to compare the performance of AltGDA and SimGDA on bilinear matrix games, under a constant stepsize over a large time horizon.

We evaluate AltGDA and SimGDA on random matrix game instances. The payoff matrices are generated from six distributions: uniform over $[0, 1]$, uniform over integers in $[-10, 10]$, binary $\{0, 1\}$ with $P(0) = 0.8$, standard normal, standard lognormal, and exponential with location 0 and scale 1. For each distribution, we generate instances of sizes $10 \times 20$, $30 \times 60$, and $60 \times 120$. All algorithms are implemented with stepsize $\eta = 0.01$ and run for $T = 10^6$ iterations. We repeat each experiment ten times, and we initialize the starting point randomly. We report the mean and standard deviation across repeats at every iteration. Results on the $10 \times 20$ instances are shown in Fig. 4, while the remaining figures are provided in Appendix F.

---

[5]Again, we pick any NE with the maximum support if there are multiple.

The experimental results show that AltGDA achieves an $O(1/T)$ convergence rate numerically, and this rate is robust to the choice of the initial point. As consistently observed, the convergence is slow in the early phase, which can be explained by the "energy decay" introduced in Section 5. In contrast, SimGDA fails to converge under a constant stepsize that is independent of the time horizon. In Appendix F.2, we test AltGDA with different stepsizes, demonstrating that the empirical convergence rate scales linearly with $1/\eta$, which is roughly in agreement with Theorems 1 and 2.

## 8 CONCLUSION

We establish the first result demonstrating AltGDA achieves faster convergence than its simultaneous counterpart in constrained minimax problems. In particular, we prove an $O(1/T)$ convergence rate of AltGDA in bilinear games with an interior NE, along with a local $O(1/T)$ convergence rate for arbitrary bilinear games. Moreover, we develop a PEP framework that simultaneously optimizes the performance measure(s) and stepsizes, and we show that AltGDA achieves an $O(1/T)$ convergence rate for any bilinear minimax problem over convex compact sets when the total number of iterations is moderately small.

### ACKNOWLEDGMENTS

This research was supported by the Office of Naval Research awards N00014-22-1-2530 and N00014-23-1-2374, and the National Science Foundation awards IIS-2147361 and IIS-2238960. S. Das Gupta acknowledges support from AFOSR Grant Number FA9550-25-1-0183.

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

# APPENDIX

## A  ADDITIONAL DETAILS ON FIG. 2 AND FIG. 3

Since the behavior of AltGDA can differ depending on whether an interior NE exists, we examine the behavior of AltGDA on two instances. In the rock-paper-scissors game which admits an interior NE, we show the trajectory of AltGDA starting from the initial points $x_0 = (1, 0, 0)$ and $y_0 = (0, 1, 0)$. For the game without interior NE, we generate a $3 \times 3$ matrix game whose payoff matrix is sampled from the standard normal distribution with random seed 1. This matrix has a non-interior NE: $x^* = (0, 0.56, 0.44), y^* = (0.37, 0.63, 0)$. We initialize AltGDA from $x_0 = y_0 = (1/3, 1/3, 1/3)$.

In both instances, we use a stepsize of $\eta = 0.01$, and we plot the evolution of the duality gap and the energy function as defined in Eqs. (Duality Gap) and (Energy).

See Fig. 10 and Fig. 11 for the evolutions of Fig. 2 and Fig. 3, respectively.

## B  OMITTED PROOFS IN SECTION 5

We start by summarizing the notations used in Appendices B and D in Table 1.

Table 1: Notation table

| NOTATION | EXPRESSION |
|---|---|
| $\mathbf{0}_n$ | $n$-dimensional all-zero vector |
| $\mathbf{1}_n$ | $n$-dimensional all-one vector |
| $\Delta_n, \Delta_m$ | Probability simplices for $x$-player and $y$-player |
| $\bar{\Delta}_n, \bar{\Delta}_m$ | $\{\boldsymbol{x} \in \mathbb{R}^n \mid \sum_{i=1}^n x_i = 1\}, \{\boldsymbol{y} \in \mathbb{R}^m \mid \sum_{j=1}^m y_j = 1\}$ |
| $(\boldsymbol{x}, \boldsymbol{y})$ | An arbitrary pair of strategies in $\Delta_n \times \Delta_m$ |
| $(\boldsymbol{x}^*, \boldsymbol{y}^*)$ | An arbitrary NE of the maximum support |
| $(\boldsymbol{x}^t, \boldsymbol{y}^t), \ \forall t \geq 0$ | A pair of iterates at the $t$-th iteration |
| $\phi_t(\boldsymbol{x}, \boldsymbol{y}), \ \forall t \geq 0$ | $\frac{1}{2}\|\boldsymbol{x}^t - \boldsymbol{x}\|_2^2 + \frac{1}{2}\|\boldsymbol{y}^t - \boldsymbol{y}\|_2^2 + \eta(\boldsymbol{y}^t)^\top A \boldsymbol{x}$ |
| $\psi_t(\boldsymbol{x}, \boldsymbol{y}), \ \forall t \geq 1$ | $\frac{1}{2}\|\boldsymbol{x}^t - \boldsymbol{x}\|_2^2 + \frac{1}{2}\|\boldsymbol{y}^{t-1} - \boldsymbol{y}\|_2^2 - \frac{1}{2}\|\boldsymbol{y}^t - \boldsymbol{y}^{t-1}\|_2^2$ |
| $I^*$ | $\{i \in [n] \mid x_i^* > 0\}$ |
| $J^*$ | $\{j \in [m] \mid y_j^* > 0\}$ |
| $I^t, \ \forall t \geq 0$ | $\{i \in [n] \mid x_i^t > 0\}$ |
| $J^t, \ \forall t \geq 0$ | $\{j \in [m] \mid y_j^t > 0\}$ |
| $\mathcal{E}(\boldsymbol{x}, \boldsymbol{y})$ | $\|\boldsymbol{x} - \boldsymbol{x}^*\|_2^2 + \|\boldsymbol{y} - \boldsymbol{y}^*\|_2^2 - \eta \boldsymbol{y}^\top A \boldsymbol{x}^t$ |
| $\mathcal{V}(\boldsymbol{x}, \boldsymbol{y})$ | $\|\boldsymbol{x} - \boldsymbol{x}^*\|_2^2 + \|\boldsymbol{y} - \boldsymbol{y}^*\|_2^2 - \eta(\boldsymbol{y} - \boldsymbol{y}^*)^\top A(\boldsymbol{x} - \boldsymbol{x}^*)$ |
| $\mathcal{V}_t, \ \forall t \geq 0$ | $\mathcal{V}(\boldsymbol{x}^t, \boldsymbol{y}^t)$ |
| $\boldsymbol{v}^t, \ \forall t \geq 0$ | $-A^\top \boldsymbol{y}^t + \frac{\sum_{\ell=1}^n (A^\top \boldsymbol{y}^t)_\ell}{n} \mathbf{1}_n$ |
| $\boldsymbol{u}^t, \ \forall t \geq 0$ | $A \boldsymbol{x}^t - \frac{\sum_{\ell=1}^m (A \boldsymbol{x}^t)_\ell}{m} \mathbf{1}_m$ |
| $\boldsymbol{\gamma}^t, \ \forall t \geq 0$ | $\frac{\Pi_{\bar{\Delta}_n}(\boldsymbol{x}^t - \eta A^\top \boldsymbol{y}^t) - \boldsymbol{x}^{t+1}}{\eta} = \frac{\boldsymbol{x}^t + \eta \boldsymbol{v}^t - \boldsymbol{x}^{t+1}}{\eta}$ |
| $\boldsymbol{\lambda}^t, \ \forall t \geq 0$ | $\frac{\Pi_{\bar{\Delta}_m}(\boldsymbol{y}^t + \eta A \boldsymbol{x}^{t+1}) - \boldsymbol{y}^{t+1}}{\eta} = \frac{\boldsymbol{y}^t + \eta \boldsymbol{u}^{t+1} - \boldsymbol{y}^{t+1}}{\eta}$ |
| $\bar{\gamma}^t, \ \forall t \geq 0$ | $\max_{i \in [n]} \gamma_i$ |
| $\bar{\lambda}^t, \ \forall t \geq 0$ | $\max_{j \in [m]} \lambda_j$ |

Before the proof, we first show the following elementary inequalities that will be used later.

**Lemma 5.** *For any $\boldsymbol{x}, \boldsymbol{x}' \in \Delta_n, \boldsymbol{y}, \boldsymbol{y}' \in \Delta_m$, we have*

1. $\|\boldsymbol{x} - \boldsymbol{x}'\|_2 \leq 2, \ \|\boldsymbol{y} - \boldsymbol{y}'\|_2 \leq 2$,
2. $(\boldsymbol{y} - \boldsymbol{y}')^\top A(\boldsymbol{x} - \boldsymbol{x}') \leq \|A\|_2 \|\boldsymbol{x} - \boldsymbol{x}'\|_2 \|\boldsymbol{y} - \boldsymbol{y}'\|_2 \leq 4\|A\|_2$,
3. $\boldsymbol{y}^\top A \boldsymbol{x} \leq \|A\|_2$,
4. $\|A^\top \boldsymbol{y}\|_2 \leq \|A\|_2$ *and* $\|A \boldsymbol{x}\|_2 \leq \|A\|_2$.

*Proof.* The first item can be shown by $\|\boldsymbol{x} - \boldsymbol{x}'\|_2 \le \|\boldsymbol{x}\|_2 + \|\boldsymbol{x}'\|_2 \le \|\boldsymbol{x}\|_1 + \|\boldsymbol{x}'\|_1 = 2$, where the last equality follows by $\boldsymbol{x}, \boldsymbol{x}' \in \Delta_n$; the $\boldsymbol{y}$ part can be done in the same way.

The second item follows from Cauchy-Schwarz inequality and the fact that because the vector norm $\|\cdot\|_2$ is compatible with the matrix norm $\|\cdot\|_2$ (Horn & Johnson, 2012, Theorem 5.6.2): $(\boldsymbol{y} - \boldsymbol{y}')^\top A(\boldsymbol{x} - \boldsymbol{x}') \le \|\boldsymbol{y} - \boldsymbol{y}'\|_2 \|A(\boldsymbol{x} - \boldsymbol{x}')\|_2 \le \|A\|_2 \|\boldsymbol{x} - \boldsymbol{x}'\|_2 \|\boldsymbol{y} - \boldsymbol{y}'\|_2$. Then, the first item implies the second one.

For the third item, for any $\boldsymbol{x} \in \Delta_n, \boldsymbol{y} \in \Delta_m$, we have

$$\boldsymbol{y}^\top A \boldsymbol{x} \overset{(a)}{\le} \|\boldsymbol{x}\|_2 \|A^\top \boldsymbol{y}\|_2 \le \|\boldsymbol{x}\|_1 \|A^\top \boldsymbol{y}\|_2 = \|A^\top \boldsymbol{y}\|_2 \overset{(b)}{\le} \|A\|_2 \|\boldsymbol{y}\|_2 \le \|A\|_2 \|\boldsymbol{y}\|_1 = \|A\|_2,$$

where $(a)$ follows from Cauchy-Schwarz inequality, $(b)$ follows because the vector norm $\|\cdot\|_2$ is compatible with the matrix norm $\|\cdot\|_2$ (Horn & Johnson, 2012, Theorem 5.6.2), and the two inequalities hold because $\boldsymbol{x} \in \Delta_n$ and $\boldsymbol{y} \in \Delta_m$.

The proof of the forth item is analogous to that of the second one: for any $\boldsymbol{y} \in \Delta_m$, we have $\|A^\top \boldsymbol{y}\|_2 \le \|A\|_2 \|\boldsymbol{y}\|_2 \le \|A\|_2 \|\boldsymbol{y}\|_1 = \|A\|_2$, where the first inequality follows by Horn & Johnson (2012, Theorem 5.6.2) and the last inequality holds because $\boldsymbol{y} \in \Delta_m$. Similarly, for any $\boldsymbol{x} \in \Delta_n$, we have $\|A\boldsymbol{x}\|_2 \le \|A\|_2 \|\boldsymbol{x}\|_2 \le \|A\|_2 \|\boldsymbol{x}\|_1 = \|A\|_2$. $\square$

We start with the proof of Lemma 1.

**Lemma 1.** Let $\{(\boldsymbol{x}^t, \boldsymbol{y}^t)\}_{t=0,1,\dots}$ be a sequence of iterates generated by Algorithm 1 with $\eta > 0$. Then, for any $(\boldsymbol{x}, \boldsymbol{y}) \in \Delta_n \times \Delta_m$, we have

$$\eta \left( \boldsymbol{y}^\top A \boldsymbol{x}^t - (\boldsymbol{y}^t)^\top A \boldsymbol{x} \right)$$
$$\le \psi_t(\boldsymbol{x}, \boldsymbol{y}) - \psi_{t+1}(\boldsymbol{x}, \boldsymbol{y}) + \eta \langle -A^\top \boldsymbol{y}^t, \boldsymbol{x}^{t+1} - \boldsymbol{x} \rangle - \frac{1}{2} \|\boldsymbol{x}^{t+1} - \boldsymbol{x}^t\|_2^2 - \frac{1}{2} \|\boldsymbol{y}^{t+1} - \boldsymbol{y}^t\|_2^2,$$
$$\forall t \ge 1 \quad (8)$$

$$\eta \left( \boldsymbol{y}^\top A \boldsymbol{x}^{t+1} - (\boldsymbol{y}^{t+1})^\top A \boldsymbol{x} \right)$$
$$\le \phi_t(\boldsymbol{x}, \boldsymbol{y}) - \phi_{t+1}(\boldsymbol{x}, \boldsymbol{y}) + \eta \langle A \boldsymbol{x}^{t+1}, \boldsymbol{y}^{t+1} - \boldsymbol{y}^t \rangle - \frac{1}{2} \|\boldsymbol{x}^{t+1} - \boldsymbol{x}^t\|_2^2 - \frac{1}{2} \|\boldsymbol{y}^{t+1} - \boldsymbol{y}^t\|_2^2,$$
$$\forall t \ge 0 \quad (9)$$

where $\phi_t(\boldsymbol{x}, \boldsymbol{y}) := \frac{1}{2} \|\boldsymbol{x}^t - \boldsymbol{x}\|_2^2 + \frac{1}{2} \|\boldsymbol{y}^t - \boldsymbol{y}\|_2^2 + \eta (\boldsymbol{y}^t)^\top A \boldsymbol{x}$ and $\psi_t(\boldsymbol{x}, \boldsymbol{y}) := \frac{1}{2} \|\boldsymbol{x}^t - \boldsymbol{x}\|_2^2 + \frac{1}{2} \|\boldsymbol{y}^{t-1} - \boldsymbol{y}\|_2^2 - \frac{1}{2} \|\boldsymbol{y}^t - \boldsymbol{y}^{t-1}\|_2^2$.

*Proof of Lemma 1.* Consider any $\boldsymbol{x} \in \mathcal{X}$ and $\boldsymbol{y} \in \mathcal{Y}$. By the property of the projection operators, we have

$$\langle \boldsymbol{x}^t - \eta A^\top \boldsymbol{y}^t - \boldsymbol{x}^{t+1}, \boldsymbol{x}^{t+1} - \boldsymbol{x} \rangle \ge 0, \ \forall t \ge 0$$
$$\langle \boldsymbol{y}^t + \eta A \boldsymbol{x}^{t+1} - \boldsymbol{y}^{t+1}, \boldsymbol{y}^{t+1} - \boldsymbol{y} \rangle \ge 0, \ \forall t \ge 0. \quad (10)$$

Thus, we have

$$\langle \boldsymbol{x}^t - \boldsymbol{x}^{t+1}, \boldsymbol{x}^{t+1} - \boldsymbol{x} \rangle \ge \eta \langle A^\top \boldsymbol{y}^t, \boldsymbol{x}^{t+1} - \boldsymbol{x} \rangle$$
$$= \eta \langle A^\top \boldsymbol{y}^{t+1}, \boldsymbol{x}^{t+1} - \boldsymbol{x} \rangle + \eta \langle A^\top \boldsymbol{y}^t - A^\top \boldsymbol{y}^{t+1}, \boldsymbol{x}^{t+1} - \boldsymbol{x} \rangle, \quad (11)$$
$$\langle \boldsymbol{y}^t - \boldsymbol{y}^{t+1}, \boldsymbol{y}^{t+1} - \boldsymbol{y} \rangle \ge -\eta \langle A \boldsymbol{x}^{t+1}, \boldsymbol{y}^{t+1} - \boldsymbol{y} \rangle. \quad (12)$$

Note that

$$2 \langle \boldsymbol{x}^t - \boldsymbol{x}^{t+1}, \boldsymbol{x}^{t+1} - \boldsymbol{x} \rangle = \|\boldsymbol{x}^t - \boldsymbol{x}\|_2^2 - \|\boldsymbol{x}^t - \boldsymbol{x}^{t+1}\|_2^2 - \|\boldsymbol{x}^{t+1} - \boldsymbol{x}\|_2^2$$
$$2 \langle \boldsymbol{y}^t - \boldsymbol{y}^{t+1}, \boldsymbol{y}^{t+1} - \boldsymbol{y} \rangle = \|\boldsymbol{y}^t - \boldsymbol{y}\|_2^2 - \|\boldsymbol{y}^t - \boldsymbol{y}^{t+1}\|_2^2 - \|\boldsymbol{y}^{t+1} - \boldsymbol{y}\|_2^2$$

and

$$\langle A^\top \boldsymbol{y}^{t+1}, \boldsymbol{x}^{t+1} - \boldsymbol{x} \rangle - \langle A \boldsymbol{x}^{t+1}, \boldsymbol{y}^{t+1} - \boldsymbol{y} \rangle = \boldsymbol{y}^\top A \boldsymbol{x}^{t+1} - (\boldsymbol{y}^{t+1})^\top A \boldsymbol{x}.$$

Denote $\phi_t(\boldsymbol{x}, \boldsymbol{y}) = \frac{1}{2}\|\boldsymbol{x}^t - \boldsymbol{x}\|_2^2 + \frac{1}{2}\|\boldsymbol{y}^t - \boldsymbol{y}\|_2^2 + \eta\langle A^\top\boldsymbol{y}^t, \boldsymbol{x}\rangle$. Combining the above inequalities and identities, we obtain Eq. (4).

Similar to Eq. (10), we have

$$\langle \boldsymbol{x}^t - \eta A^\top\boldsymbol{y}^t - \boldsymbol{x}^{t+1}, \boldsymbol{x}^{t+1} - \boldsymbol{x}\rangle \geq 0, \ \forall t \geq 0$$
$$\langle \boldsymbol{y}^{t-1} + \eta A\boldsymbol{x}^t - \boldsymbol{y}^t, \boldsymbol{y}^t - \boldsymbol{y}\rangle \geq 0, \ \forall t \geq 1.$$

Thus, we have

$$\langle \boldsymbol{x}^t - \boldsymbol{x}^{t+1}, \boldsymbol{x}^{t+1} - \boldsymbol{x}\rangle \geq \eta\langle A^\top\boldsymbol{y}^t, \boldsymbol{x}^{t+1} - \boldsymbol{x}\rangle = \eta\langle A^\top\boldsymbol{y}^t, \boldsymbol{x}^t - \boldsymbol{x}\rangle + \eta\langle A^\top\boldsymbol{y}^t, \boldsymbol{x}^{t+1} - \boldsymbol{x}^t\rangle,$$
$$\langle \boldsymbol{y}^{t-1} - \boldsymbol{y}^t, \boldsymbol{y}^t - \boldsymbol{y}\rangle \geq -\eta\langle A\boldsymbol{x}^t, \boldsymbol{y}^t - \boldsymbol{y}\rangle.$$

Denote $\psi_t(\boldsymbol{x}, \boldsymbol{y}) = \frac{1}{2}\|\boldsymbol{x}^t - \boldsymbol{x}\|_2^2 + \frac{1}{2}\|\boldsymbol{y}^{t-1} - \boldsymbol{y}\|_2^2 - \frac{1}{2}\|\boldsymbol{y}^t - \boldsymbol{y}^{t-1}\|_2^2$. Combining the above two inequalities, we obtain Eq. (3). □

Next, we proceed with proving Lemma 2. Before that, we present a few lemmas.

For any positive integer $d$, we denote $\bar{\Delta}_d = \{\boldsymbol{x} \in \mathbb{R}^d \mid \sum_{i=1}^d x_i = 1\}$, which is the affine hull of the probability simplex $\Delta_d$. The following lemma connects the projection onto a simplex $\Delta_d$ with the projection onto its affine hull.

**Lemma 6.** *For any $\boldsymbol{y} \in \mathbb{R}^d$, we have $\Pi_{\Delta_d}(\boldsymbol{y}) = \Pi_{\Delta_d}(\Pi_{\bar{\Delta}_d}(\boldsymbol{y}))$. Furthermore, for any $\boldsymbol{x} \in \Delta_d$, we have $\langle \boldsymbol{\gamma}, \Pi_{\Delta_d}(\boldsymbol{y}) - \boldsymbol{x}\rangle \geq 0$ where $\boldsymbol{\gamma} := \Pi_{\bar{\Delta}_d}(\boldsymbol{y}) - \Pi_{\Delta_d}(\boldsymbol{y})$.*

*Proof.* Using the properties of projection onto a closed affine set (Bauschke & Combettes, 2017, Corollary 3.22), we have $\|\boldsymbol{x} - \boldsymbol{y}\|_2^2 = \|\boldsymbol{x} - \Pi_{\bar{\Delta}_d}(\boldsymbol{y})\|_2^2 + \|\Pi_{\bar{\Delta}_d}(\boldsymbol{y}) - \boldsymbol{y}\|_2^2$ for any $\boldsymbol{x} \in \bar{\Delta}_d$. Hence, using the definition of projection,

$$\Pi_{\Delta_d}(\boldsymbol{y}) = \operatorname*{argmin}_{\boldsymbol{x}\in\Delta_d}\|\boldsymbol{x} - \boldsymbol{y}\|_2^2 = \operatorname*{argmin}_{\boldsymbol{x}\in\Delta_d}\|\boldsymbol{x} - \Pi_{\bar{\Delta}_d}(\boldsymbol{y})\|_2^2 = \Pi_{\Delta_d}(\Pi_{\bar{\Delta}_d}(\boldsymbol{y})).$$

Then, using the properties of projection onto a closed convex set again, we have $\langle \Pi_{\bar{\Delta}_d}(\boldsymbol{y}) - \Pi_{\Delta_d}(\boldsymbol{y}), \Pi_{\Delta_d}(\boldsymbol{y}) - \boldsymbol{x}\rangle \geq 0$ for any $\boldsymbol{x} \in \Delta_d$. □

Denote

$$\boldsymbol{\gamma}^t := \frac{\Pi_{\bar{\Delta}_n}(\boldsymbol{x}^t - \eta A^\top\boldsymbol{y}^t) - \boldsymbol{x}^{t+1}}{\eta} \tag{13}$$

and

$$\boldsymbol{\lambda}^t := \frac{\Pi_{\bar{\Delta}_m}(\boldsymbol{y}^t + \eta A\boldsymbol{x}^{t+1}) - \boldsymbol{y}^{t+1}}{\eta}. \tag{14}$$

The following lemma provides two useful inequalities involving $\boldsymbol{\gamma}^t$ and $\boldsymbol{\lambda}^t$.

**Lemma 7.** *Assume that the bilinear game admits an interior NE. Let $\{(\boldsymbol{x}^t, \boldsymbol{y}^t)\}_{t=0,1,\dots}$ be a sequence of iterates generated by Algorithm 1 with $\eta \leq \frac{1}{\|A\|_2}\min\{\min_{i\in[n]} x_i^*, \min_{j\in[m]} y_j^*\}$. Then, the iterates of AltGDA satisfy*

1. *$\langle \boldsymbol{\gamma}^t, \boldsymbol{x}^{t+1} - \boldsymbol{x}\rangle \geq 0, \ \forall \boldsymbol{x} \in \Delta_n$ and $\langle \boldsymbol{\lambda}^t, \boldsymbol{y}^{t+1} - \boldsymbol{y}\rangle \geq 0, \ \forall \boldsymbol{y} \in \Delta_m$,*
2. *$\langle \boldsymbol{\gamma}^t, \boldsymbol{x}^t - \boldsymbol{x}^*\rangle \geq 0$ and $\langle \boldsymbol{\lambda}^t, \boldsymbol{y}^t - \boldsymbol{y}^*\rangle \geq 0$.*

*Proof.* The first item directly follows from Lemma 6.

For the second item, we have

$$\|\boldsymbol{x}^{t+1} - \boldsymbol{x}^t\|_2 = \|\Pi_{\Delta_n}(\boldsymbol{x}^t - \eta A^\top\boldsymbol{y}^t) - \Pi_{\Delta_n}(\boldsymbol{x}^t)\|_2 \leq \|\boldsymbol{x}^t - \eta A^\top\boldsymbol{y}^t - \boldsymbol{x}^t\|_2 \leq \eta\|A\|_2, \tag{15}$$

where the first inequality is by the nonexpansiveness of the projection operator $\Pi_{\Delta_n}$ and the last inequality follows by Lemma 5. As a result, $\boldsymbol{x}^{t+1} - \boldsymbol{x}^t \in \mathcal{B}(\boldsymbol{0}_n, \eta\|A\|_2)$. Then, we have

$$
\begin{aligned}
&\left\langle \boldsymbol{\gamma}^t,\, \boldsymbol{x}^t - \boldsymbol{x}^* \right\rangle \\
={}&\left\langle \boldsymbol{\gamma}^t,\, \boldsymbol{x}^{t+1} - \boldsymbol{x}^* \right\rangle + \left\langle \boldsymbol{\gamma}^t,\, \boldsymbol{x}^t - \boldsymbol{x}^{t+1} \right\rangle \\
\geq{}&\left\langle \boldsymbol{\gamma}^t,\, \boldsymbol{x}^{t+1} - \boldsymbol{x}^* \right\rangle + \left\langle \boldsymbol{\gamma}^t,\, -\eta\|A\|_2 \frac{\boldsymbol{\gamma}^t}{\|\boldsymbol{\gamma}^t\|_2} \right\rangle &&\text{(by } \boldsymbol{x}^{t+1} - \boldsymbol{x}^t \in \mathcal{B}(\boldsymbol{0}_n, \eta\|A\|_2)) \\
={}&\left\langle \boldsymbol{\gamma}^t,\, \boldsymbol{x}^{t+1} - \eta\|A\|_2 \frac{\boldsymbol{\gamma}^t}{\|\boldsymbol{\gamma}^t\|_2} - \boldsymbol{x}^* \right\rangle \geq 0,
\end{aligned}
\tag{16}
$$

where the last inequality follows from the first item and

$$
\boldsymbol{x}^* + \eta\|A\|_2 \frac{\boldsymbol{\gamma}^t}{\|\boldsymbol{\gamma}^t\|_2} \in \mathcal{B}\left( \boldsymbol{x}^*, \min\left\{ \min_{i\in[n]} x_i^*, \min_{j\in[m]} y_j^* \right\} \right) \bigcap \bar{\Delta}_n \subset \Delta_n.
$$

Here, $\boldsymbol{x}^* + \eta\|A\|_2 \frac{\boldsymbol{\gamma}^t}{\|\boldsymbol{\gamma}^t\|_2} \in \bar{\Delta}_n$ is because $\sum_{i\in[n]} \gamma_i^t = 0$. Similarly, we can prove that $\left\langle \boldsymbol{\lambda}^t,\, \boldsymbol{y}^t - \boldsymbol{y}^* \right\rangle \geq 0$. $\qquad\square$

Recall that the energy function $\mathcal{E} : \Delta_n \times \Delta_m \to \mathbb{R}$ is defined as

$$
\mathcal{E}\left( \boldsymbol{x}^t, \boldsymbol{y}^t \right) = \left\| \boldsymbol{x}^t - \boldsymbol{x}^* \right\|_2^2 + \left\| \boldsymbol{y}^t - \boldsymbol{y}^* \right\|_2^2 - \eta(\boldsymbol{y}^t)^\top A \boldsymbol{x}^t,
$$

where $(\boldsymbol{x}^*, \boldsymbol{y}^*)$ is any Nash equilibrium with full support. We now show this energy function is non-increasing in $t$ in the following lemma.

**Lemma 8.** *Assume that the bilinear game admits an interior NE. Let $\{(\boldsymbol{x}^t, \boldsymbol{y}^t)\}_{t=0,1,\dots}$ be a sequence of iterates generated by Algorithm 1 with $\eta \leq \frac{1}{\|A\|_2} \min\{\min_{i\in[n]} x_i^*, \min_{j\in[m]} y_j^*\}$. Then, we have $\mathcal{E}\left( \boldsymbol{x}^{t+1}, \boldsymbol{y}^{t+1} \right) \leq \mathcal{E}\left( \boldsymbol{x}^t, \boldsymbol{y}^t \right)$ for all $t \geq 0$. In particular, we have for all $t \geq 0$*

$$
\mathcal{E}\left( \boldsymbol{x}^t, \boldsymbol{y}^t \right) - \mathcal{E}\left( \boldsymbol{x}^{t+1}, \boldsymbol{y}^{t+1} \right) = \eta\left\langle \boldsymbol{\gamma}^t,\, \boldsymbol{x}^{t+1} + \boldsymbol{x}^t - 2\boldsymbol{x}^* \right\rangle + \eta\left\langle \boldsymbol{\lambda}^t,\, \boldsymbol{y}^{t+1} + \boldsymbol{y}^t - 2\boldsymbol{y}^* \right\rangle \geq 0. \tag{17}
$$

*Proof.* Because $\Pi_{\bar{\Delta}_d}\left( \boldsymbol{u} + \boldsymbol{g} \right) = \boldsymbol{u} + \boldsymbol{g} - \frac{1}{d}\left( \boldsymbol{1}_d^\top \boldsymbol{g} \right) \boldsymbol{1}_d$ for any $\boldsymbol{u} \in \bar{\Delta}_d$ and $\boldsymbol{g} \in \mathbb{R}^d$ (Beck, 2017, Lemma 6.26), we have

$$
\begin{aligned}
\boldsymbol{x}^{t+1} &= \boldsymbol{x}^t - \eta A^\top \boldsymbol{y}^t + \frac{\eta}{n} \sum_{i=1}^n \left( A^\top \boldsymbol{y}^t \right)_i \boldsymbol{1}_n - \eta\boldsymbol{\gamma}^t \\
\boldsymbol{y}^{t+1} &= \boldsymbol{y}^t + \eta A \boldsymbol{x}^{t+1} - \frac{\eta}{m} \sum_{j=1}^m \left( A \boldsymbol{x}^{t+1} \right)_j \boldsymbol{1}_m - \eta\boldsymbol{\lambda}^t.
\end{aligned}
\tag{18}
$$

Hence, we have

$$
\begin{aligned}
\left\langle \boldsymbol{x}^{t+1} - \boldsymbol{x}^t + \eta A^\top \boldsymbol{y}^t - \frac{\eta}{n} \sum_{i=1}^n (A^\top \boldsymbol{y}^t)_i \cdot \boldsymbol{1}_n + \eta\boldsymbol{\gamma}^t,\, \boldsymbol{x}^{t+1} + \boldsymbol{x}^t - 2\boldsymbol{x}^* \right\rangle &= 0 \\
\left\langle \boldsymbol{y}^{t+1} - \boldsymbol{y}^t - \eta A \boldsymbol{x}^{t+1} + \frac{\eta}{m} \sum_{j=1}^m (A \boldsymbol{x}^{t+1})_j \cdot \boldsymbol{1}_m + \eta\boldsymbol{\lambda}^t,\, \boldsymbol{y}^{t+1} + \boldsymbol{y}^t - 2\boldsymbol{y}^* \right\rangle &= 0.
\end{aligned}
\tag{19}
$$

Because $\left\langle \boldsymbol{1}_n,\, \boldsymbol{x}^{t+1} + \boldsymbol{x}^t - 2\boldsymbol{x}^* \right\rangle = \left\langle \boldsymbol{1}_m,\, \boldsymbol{y}^{t+1} + \boldsymbol{y}^t - 2\boldsymbol{y}^* \right\rangle = 0$, and $\left\langle \boldsymbol{a} - \boldsymbol{b},\, \boldsymbol{a} + \boldsymbol{b} \right\rangle = \|\boldsymbol{a}\|_2^2 - \|\boldsymbol{b}\|_2^2$ for any vectors $\boldsymbol{a}, \boldsymbol{b}$, the above inequalities are equivalent to

$$
\begin{aligned}
\left\| \boldsymbol{x}^{t+1} - \boldsymbol{x}^* \right\|_2^2 - \left\| \boldsymbol{x}^t - \boldsymbol{x}^* \right\|_2^2 + \eta\left\langle A^\top \boldsymbol{y}^t,\, \boldsymbol{x}^{t+1} + \boldsymbol{x}^t - 2\boldsymbol{x}^* \right\rangle + \eta\left\langle \boldsymbol{\gamma}^t,\, \boldsymbol{x}^{t+1} + \boldsymbol{x}^t - 2\boldsymbol{x}^* \right\rangle &= 0 \\
\left\| \boldsymbol{y}^{t+1} - \boldsymbol{y}^* \right\|_2^2 - \left\| \boldsymbol{y}^t - \boldsymbol{y}^* \right\|_2^2 - \eta\left\langle A \boldsymbol{x}^{t+1},\, \boldsymbol{y}^{t+1} + \boldsymbol{y}^t - 2\boldsymbol{y}^* \right\rangle + \eta\left\langle \boldsymbol{\lambda}^t,\, \boldsymbol{y}^{t+1} + \boldsymbol{y}^t - 2\boldsymbol{y}^* \right\rangle &= 0.
\end{aligned}
\tag{20}
$$

Summing up the above two inequalities and plugging in the definition of energy function $\mathcal{E}$, we have

$$
\begin{aligned}
\mathcal{E}\left( \boldsymbol{x}^{t+1}, \boldsymbol{y}^{t+1} \right) - \mathcal{E}\left( \boldsymbol{x}^t, \boldsymbol{y}^t \right) - 2\eta\left\langle A \boldsymbol{x}^*,\, \boldsymbol{y}^t \right\rangle + 2\eta\left\langle A^\top \boldsymbol{y}^*,\, \boldsymbol{x}^{t+1} \right\rangle \\
+ \eta\left\langle \boldsymbol{\gamma}^t,\, \boldsymbol{x}^{t+1} + \boldsymbol{x}^t - 2\boldsymbol{x}^* \right\rangle + \eta\left\langle \boldsymbol{\lambda}^t,\, \boldsymbol{y}^{t+1} + \boldsymbol{y}^t - 2\boldsymbol{y}^* \right\rangle = 0.
\end{aligned}
\tag{21}
$$

Equivalently,

$$\mathcal{E}\left(\boldsymbol{x}^{t+1}, \boldsymbol{y}^{t+1}\right) - \mathcal{E}\left(\boldsymbol{x}^{t}, \boldsymbol{y}^{t}\right) + 2\eta\langle\nu^{*}\mathbf{1}_{m} - A\boldsymbol{x}^{*}, \boldsymbol{y}^{t}\rangle + 2\eta\langle A^{\top}\boldsymbol{y}^{*} - \nu^{*}\mathbf{1}_{n}, \boldsymbol{x}^{t+1}\rangle$$
$$+ \eta\langle\boldsymbol{\gamma}^{t}, \boldsymbol{x}^{t+1} + \boldsymbol{x}^{t} - 2\boldsymbol{x}^{*}\rangle + \eta\langle\boldsymbol{\lambda}^{t}, \boldsymbol{y}^{t+1} + \boldsymbol{y}^{t} - 2\boldsymbol{y}^{*}\rangle = 0. \tag{22}$$

Note that $\boldsymbol{y}^{\top}A\boldsymbol{x}^{*} \le \nu^{*} = (\boldsymbol{y}^{*})^{\top}A\boldsymbol{x}^{*} \le (\boldsymbol{y}^{*})^{\top}A\boldsymbol{x} \ \forall \ \boldsymbol{x} \in \Delta_{n}, \boldsymbol{y} \in \Delta_{m}$ implies that $A\boldsymbol{x}^{*} \le \nu^{*}\mathbf{1}_{m}$ and $A^{\top}\boldsymbol{y}^{*} \ge \nu^{*}\mathbf{1}_{n}$. Therefore, we have

$$\mathcal{E}\left(\boldsymbol{x}^{t+1}, \boldsymbol{y}^{t+1}\right) - \mathcal{E}\left(\boldsymbol{x}^{t}, \boldsymbol{y}^{t}\right) + \eta\langle\boldsymbol{\gamma}^{t}, \boldsymbol{x}^{t+1} + \boldsymbol{x}^{t} - 2\boldsymbol{x}^{*}\rangle + \eta\langle\boldsymbol{\lambda}^{t}, \boldsymbol{y}^{t+1} + \boldsymbol{y}^{t} - 2\boldsymbol{y}^{*}\rangle \le 0.$$

That is,

$$\mathcal{E}\left(\boldsymbol{x}^{t}, \boldsymbol{y}^{t}\right) - \mathcal{E}\left(\boldsymbol{x}^{t+1}, \boldsymbol{y}^{t+1}\right) \ge \eta\langle\boldsymbol{\gamma}^{t}, \boldsymbol{x}^{t+1} + \boldsymbol{x}^{t} - 2\boldsymbol{x}^{*}\rangle + \eta\langle\boldsymbol{\lambda}^{t}, \boldsymbol{y}^{t+1} + \boldsymbol{y}^{t} - 2\boldsymbol{y}^{*}\rangle.$$

If additionally the game admits an interior NE, we have $A\boldsymbol{x}^{*} = \nu^{*}\mathbf{1}_{m}$ and $A^{\top}\boldsymbol{y}^{*} = \nu^{*}\mathbf{1}_{n}$. Therefore,

$$\mathcal{E}\left(\boldsymbol{x}^{t}, \boldsymbol{y}^{t}\right) - \mathcal{E}\left(\boldsymbol{x}^{t+1}, \boldsymbol{y}^{t+1}\right) = \eta\langle\boldsymbol{\gamma}^{t}, \boldsymbol{x}^{t+1} + \boldsymbol{x}^{t} - 2\boldsymbol{x}^{*}\rangle + \eta\langle\boldsymbol{\lambda}^{t}, \boldsymbol{y}^{t+1} + \boldsymbol{y}^{t} - 2\boldsymbol{y}^{*}\rangle. \tag{23}$$

Combining Eq. (23) with Lemma 7 (whose second item requires the presence of an interior NE) completes this lemma. $\square$

Now, we are ready to prove Lemma 2. Recall that

$$r_{t} = \eta\langle-A^{\top}\boldsymbol{y}^{t}, \boldsymbol{x}^{t+1} - \boldsymbol{x}^{t}\rangle + \eta\langle A\boldsymbol{x}^{t+1}, \boldsymbol{y}^{t+1} - \boldsymbol{y}^{t}\rangle - \|\boldsymbol{x}^{t+1} - \boldsymbol{x}^{t}\|_{2}^{2} - \|\boldsymbol{y}^{t+1} - \boldsymbol{y}^{t}\|_{2}^{2}$$
$$= \langle-\eta A^{\top}\boldsymbol{y}^{t} - \boldsymbol{x}^{t+1} + \boldsymbol{x}^{t}, \boldsymbol{x}^{t+1} - \boldsymbol{x}^{t}\rangle + \langle\eta A\boldsymbol{x}^{t+1} - \boldsymbol{y}^{t+1} + \boldsymbol{y}^{t}, \boldsymbol{y}^{t+1} - \boldsymbol{y}^{t}\rangle.$$

**Lemma 2.** Assume that the bilinear game admits an interior NE. Let $\{(\boldsymbol{x}^{t}, \boldsymbol{y}^{t})\}_{t=0,1,\dots}$ be a sequence of iterates generated by Algorithm 1 with $\eta \le \frac{1}{\|A\|_{2}} \min\{\min_{i\in[n]} x_{i}^{*}, \min_{j\in[m]} y_{j}^{*}\}$. Then, we have

$$0 \le r_{t} \le \mathcal{E}(\boldsymbol{x}^{t}, \boldsymbol{y}^{t}) - \mathcal{E}(\boldsymbol{x}^{t+1}, \boldsymbol{y}^{t+1}), \quad \forall t \ge 0.$$

*Proof of Lemma 2.* By Eq. (18) and $\langle\mathbf{1}_{n}, \boldsymbol{x}^{t+1} - \boldsymbol{x}^{t}\rangle = \langle\mathbf{1}_{m}, \boldsymbol{y}^{t+1} - \boldsymbol{y}^{t}\rangle = 0$, we have

$$\eta\langle-A^{\top}\boldsymbol{y}^{t}, \boldsymbol{x}^{t+1} - \boldsymbol{x}^{t}\rangle - \|\boldsymbol{x}^{t+1} - \boldsymbol{x}^{t}\|_{2}^{2} = \langle\eta\boldsymbol{\gamma}^{t}, \boldsymbol{x}^{t+1} - \boldsymbol{x}^{t}\rangle$$
$$\eta\langle A\boldsymbol{x}^{t+1}, \boldsymbol{y}^{t+1} - \boldsymbol{y}^{t}\rangle - \|\boldsymbol{y}^{t+1} - \boldsymbol{y}^{t}\|_{2}^{2} = \langle\eta\boldsymbol{\lambda}^{t}, \boldsymbol{y}^{t+1} - \boldsymbol{y}^{t}\rangle,$$

On the other hand, we have

$$\langle\eta\boldsymbol{\gamma}^{t}, \boldsymbol{x}^{t+1} + \boldsymbol{x}^{t} - 2\boldsymbol{x}^{*}\rangle - \langle\eta\boldsymbol{\gamma}^{t}, \boldsymbol{x}^{t+1} - \boldsymbol{x}^{t}\rangle = 2\langle\eta\boldsymbol{\gamma}^{t}, \boldsymbol{x}^{t} - \boldsymbol{x}^{*}\rangle \ge 0$$
$$\langle\eta\boldsymbol{\lambda}^{t}, \boldsymbol{y}^{t+1} + \boldsymbol{y}^{t} - 2\boldsymbol{y}^{*}\rangle - \langle\eta\boldsymbol{\lambda}^{t}, \boldsymbol{y}^{t+1} - \boldsymbol{y}^{t}\rangle = 2\langle\eta\boldsymbol{\lambda}^{t}, \boldsymbol{y}^{t} - \boldsymbol{y}^{*}\rangle \ge 0,$$

where the inequalities follow from the second item in Lemma 7. Combining the above equalities and inequalities yields

$$\eta\langle-A^{\top}\boldsymbol{y}^{t}, \boldsymbol{x}^{t+1} - \boldsymbol{x}^{t}\rangle - \|\boldsymbol{x}^{t+1} - \boldsymbol{x}^{t}\|_{2}^{2} \le \langle\eta\boldsymbol{\gamma}^{t}, \boldsymbol{x}^{t+1} + \boldsymbol{x}^{t} - 2\boldsymbol{x}^{*}\rangle$$
$$\eta\langle A\boldsymbol{x}^{t+1}, \boldsymbol{y}^{t+1} - \boldsymbol{y}^{t}\rangle - \|\boldsymbol{y}^{t+1} - \boldsymbol{y}^{t}\|_{2}^{2} \le \langle\eta\boldsymbol{\lambda}^{t}, \boldsymbol{y}^{t+1} + \boldsymbol{y}^{t} - 2\boldsymbol{y}^{*}\rangle. \tag{24}$$

Summing up the two inequalities in Eq. (24), by Lemma 8, we obtain Lemma 2. $\square$

Then, we arrive at the $O(1/T)$ convergence rate.

**Theorem 1.** Assume that the bilinear game admits an interior NE. Let $\{(\boldsymbol{x}^{t}, \boldsymbol{y}^{t})\}_{t=0,1,\dots}$ be a sequence of iterates generated by Algorithm 1 with $\eta \le \frac{1}{\|A\|_{2}} \min\{\min_{i\in[n]} x_{i}^{*}, \min_{j\in[m]} y_{j}^{*}\}$. Then, we have

$$\mathtt{DualityGap}\left(\frac{1}{T}\sum_{t=1}^{T}\boldsymbol{x}^{t}, \frac{1}{T}\sum_{t=1}^{T}\boldsymbol{y}^{t}\right) \le \frac{9 + 4\eta\|A\|_{2}}{\eta T}. \tag{25}$$

*Proof of Theorem 1.* Summing up Eqs. (3) and (4), we have

$$\eta \left(\boldsymbol{y}^\top A \boldsymbol{x}^{t+1} - (\boldsymbol{y}^{t+1})^\top A \boldsymbol{x}\right) + \eta \left(\boldsymbol{y}^\top A \boldsymbol{x}^t - (\boldsymbol{y}^t)^\top A \boldsymbol{x}\right)$$

$$\leq \phi_t(\boldsymbol{x}, \boldsymbol{y}) - \phi_{t+1}(\boldsymbol{x}, \boldsymbol{y}) + \psi_t(\boldsymbol{x}, \boldsymbol{y}) - \psi_{t+1}(\boldsymbol{x}, \boldsymbol{y})$$

$$+ \eta \langle A \boldsymbol{x}^{t+1}, \, \boldsymbol{y}^{t+1} - \boldsymbol{y}^t \rangle - \eta \langle A^\top \boldsymbol{y}^t, \, \boldsymbol{x}^{t+1} - \boldsymbol{x}^t \rangle - \left\| \boldsymbol{x}^{t+1} - \boldsymbol{x}^t \right\|_2^2 - \left\| \boldsymbol{y}^{t+1} - \boldsymbol{y}^t \right\|_2^2.$$

By Lemma 2, we obtain that

$$\eta \left(\boldsymbol{y}^\top A \boldsymbol{x}^{t+1} - (\boldsymbol{y}^{t+1})^\top A \boldsymbol{x}\right) + \eta \left(\boldsymbol{y}^\top A \boldsymbol{x}^t - (\boldsymbol{y}^t)^\top A \boldsymbol{x}\right) \leq$$

$$\phi_t(\boldsymbol{x}, \boldsymbol{y}) - \phi_{t+1}(\boldsymbol{x}, \boldsymbol{y}) + \psi_t(\boldsymbol{x}, \boldsymbol{y}) - \psi_{t+1}(\boldsymbol{x}, \boldsymbol{y}) + \mathcal{E}\left(\boldsymbol{x}^t, \boldsymbol{y}^t\right) - \mathcal{E}\left(\boldsymbol{x}^{t+1}, \boldsymbol{y}^{t+1}\right). \quad (26)$$

Summing up Eq. (26) over $t = 1, \ldots, T$ plus Eq. (4) for $t = 0$, we have

$$2\eta \sum_{t=1}^{T} \left(\boldsymbol{y}^\top A \boldsymbol{x}^t - (\boldsymbol{y}^t)^\top A \boldsymbol{x}\right) + \eta \left(\boldsymbol{y}^\top A \boldsymbol{x}^{T+1} - (\boldsymbol{y}^{T+1})^\top A \boldsymbol{x}\right)$$

$$\leq \phi_1(\boldsymbol{x}, \boldsymbol{y}) - \phi_{T+1}(\boldsymbol{x}, \boldsymbol{y}) + \psi_1(\boldsymbol{x}, \boldsymbol{y}) - \psi_{T+1}(\boldsymbol{x}, \boldsymbol{y}) + \mathcal{E}\left(\boldsymbol{x}^1, \boldsymbol{y}^1\right) - \mathcal{E}\left(\boldsymbol{x}^{T+1}, \boldsymbol{y}^{T+1}\right)$$

$$+ \phi_0(\boldsymbol{x}, \boldsymbol{y}) - \phi_1(\boldsymbol{x}, \boldsymbol{y}) + \eta \langle A \boldsymbol{x}^1, \, \boldsymbol{y}^1 - \boldsymbol{y}^0 \rangle - \frac{1}{2} \left\| \boldsymbol{x}^1 - \boldsymbol{x}^0 \right\|_2^2 - \frac{1}{2} \left\| \boldsymbol{y}^1 - \boldsymbol{y}^0 \right\|_2^2$$

$$\leq \phi_0(\boldsymbol{x}, \boldsymbol{y}) - \phi_{T+1}(\boldsymbol{x}, \boldsymbol{y}) + \psi_1(\boldsymbol{x}, \boldsymbol{y}) - \psi_{T+1}(\boldsymbol{x}, \boldsymbol{y}) + \mathcal{E}\left(\boldsymbol{x}^1, \boldsymbol{y}^1\right) - \mathcal{E}\left(\boldsymbol{x}^{T+1}, \boldsymbol{y}^{T+1}\right)$$

$$+ \eta \langle A \boldsymbol{x}^1, \, \boldsymbol{y}^1 - \boldsymbol{y}^0 \rangle.$$

This inequality gives the following upper bound:

$$\boldsymbol{y}^\top A \left(\frac{1}{T} \sum_{t=1}^{T} \boldsymbol{x}^t\right) - \left(\frac{1}{T} \sum_{t=1}^{T} \boldsymbol{y}^t\right)^\top A \boldsymbol{x} = \frac{1}{T} \sum_{t=1}^{T} \left(\boldsymbol{y}^\top A \boldsymbol{x}^t - (\boldsymbol{y}^t)^\top A \boldsymbol{x}\right) \leq \frac{C(\boldsymbol{x}, \boldsymbol{y})}{2\eta T}, \quad (27)$$

where

$$C(\boldsymbol{x}, \boldsymbol{y}) = \phi_0(\boldsymbol{x}, \boldsymbol{y}) - \phi_{T+1}(\boldsymbol{x}, \boldsymbol{y}) + \psi_1(\boldsymbol{x}, \boldsymbol{y}) - \psi_{T+1}(\boldsymbol{x}, \boldsymbol{y}) + \mathcal{E}\left(\boldsymbol{x}^0, \boldsymbol{y}^0\right) - \mathcal{E}\left(\boldsymbol{x}^{T+1}, \boldsymbol{y}^{T+1}\right)$$

$$- \eta \langle A \boldsymbol{x}^1, \, \boldsymbol{y}^1 - \boldsymbol{y}^0 \rangle - \eta \left(\boldsymbol{y}^\top A \boldsymbol{x}^{T+1} - (\boldsymbol{y}^{T+1})^\top A \boldsymbol{x}\right)$$

$$\forall \, \boldsymbol{x}, \boldsymbol{y} \in \Delta_m \times \Delta_n.$$

For any $\boldsymbol{x} \in \Delta_n, \boldsymbol{y} \in \Delta_m$, we can bound each term in $C(\boldsymbol{x}, \boldsymbol{y})$ as follows:

$$\phi_0(\boldsymbol{x}, \boldsymbol{y}) = \frac{1}{2} \left\| \boldsymbol{x}^0 - \boldsymbol{x} \right\|_2^2 + \frac{1}{2} \left\| \boldsymbol{y}^0 - \boldsymbol{y} \right\|_2^2 + \eta (\boldsymbol{y}^0)^\top A \boldsymbol{x} \leq 4 + \eta \|A\|_2,$$

$$-\phi_{T+1}(\boldsymbol{x}, \boldsymbol{y}) = -\frac{1}{2} \left\| \boldsymbol{x}^{T+1} - \boldsymbol{x} \right\|_2^2 - \frac{1}{2} \left\| \boldsymbol{y}^{T+1} - \boldsymbol{y} \right\|_2^2 - \eta (\boldsymbol{y}^{T+1})^\top A \boldsymbol{x} \leq \eta \|A\|_2,$$

$$\psi_1(\boldsymbol{x}, \boldsymbol{y}) = \frac{1}{2} \left\| \boldsymbol{x}^1 - \boldsymbol{x} \right\|_2^2 + \frac{1}{2} \left\| \boldsymbol{y}^0 - \boldsymbol{y} \right\|_2^2 - \frac{1}{2} \left\| \boldsymbol{y}^1 - \boldsymbol{y}^0 \right\|_2^2 \leq 4, \quad (28)$$

$$-\psi_{T+1}(\boldsymbol{x}, \boldsymbol{y}) = -\frac{1}{2} \left\| \boldsymbol{x}^{T+1} - \boldsymbol{x} \right\|_2^2 - \frac{1}{2} \left\| \boldsymbol{y}^T + \boldsymbol{y} \right\|_2^2 + \frac{1}{2} \left\| \boldsymbol{y}^{T+1} - \boldsymbol{y}^T \right\|_2^2 \leq 2,$$

$$\mathcal{E}\left(\boldsymbol{x}^0, \boldsymbol{y}^0\right) = \left\| \boldsymbol{x}^0 - \boldsymbol{x}^* \right\|_2^2 + \left\| \boldsymbol{y}^0 - \boldsymbol{y}^* \right\|_2^2 - \eta (\boldsymbol{y}^0)^\top A \boldsymbol{x}^0 \leq 8 + \eta \|A\|_2,$$

$$-\mathcal{E}\left(\boldsymbol{x}^{T+1}, \boldsymbol{y}^{T+1}\right) = -\left\| \boldsymbol{x}^{T+1} - \boldsymbol{x}^* \right\|_2^2 - \left\| \boldsymbol{y}^{T+1} - \boldsymbol{y}^* \right\|_2^2 + \eta (\boldsymbol{y}^{T+1})^\top A \boldsymbol{x}^{T+1} \leq \eta \|A\|_2,$$

and $-\eta \langle A \boldsymbol{x}^1, \, \boldsymbol{y}^1 - \boldsymbol{y}^0 \rangle - \eta \left(\boldsymbol{y}^\top A \boldsymbol{x}^{T+1} - (\boldsymbol{y}^{T+1})^\top A \boldsymbol{x}\right) \leq 4\eta \|A\|_2$, where all the inequalities follow by Lemma 5. Therefore, we can bound $C(\boldsymbol{x}, \boldsymbol{y})$ by $18 + 8\eta \|A\|_2$. By taking the maximum on the both sides of Eq. (27), we complete the proof. $\square$

## C  ADDITIONAL RESULTS IN SECTION 5

In this section, we provide several additional results implied by our main results in Section 5.

First, we notice that in harmonic games (Candogan et al., 2011), the uniformly mixed strategy profile is always a Nash equilibrium (Candogan et al., 2011, Theorem 5.5.). Therefore, as a corollary of Theorem 1, we have

**Corollary 1.** *In harmonic games, let $\{(\boldsymbol{x}^t, \boldsymbol{y}^t)\}_{t=0,1,\ldots}$ be a sequence of iterates generated by Algorithm 1 with $\eta \le \frac{1}{\|A\|_2} \min\{\frac{1}{n}, \frac{1}{m}\}$, starting from any initial point within $\Delta_n \times \Delta_m$. Then, the duality gap of the averaged iterate converges with a rate of $O(1/T)$.*

## C.1 "INTERIOR CYCLIC TRAJECTORIES" INDICATE INTERIOR NE

As an additional result, we show that we can detect the presence of interior NE via observing the trajectories of AltGDA. In particular, the presence of "interior cyclic trajectories" indicates that the game cannot admit non-degenerate non-interior NE.

Here, we adopt a strict definition of "interior cyclic trajectories." Specifically, we call a trajectory an *interior cyclic trajectory* if (1) it evolves along a periodic orbit, and (2) it remains strictly in the interior of the simplices. Formally, we say AltGDA eventually exhibits an interior cyclic trajectory if there exist $T, s \ge 1$ such that

- $(\boldsymbol{x}^t, \boldsymbol{y}^t) = (\boldsymbol{x}^{t+s}, \boldsymbol{y}^{t+s})$ for all $t > T$;
- $\boldsymbol{\gamma}^t = \boldsymbol{0}_n$ and $\boldsymbol{\lambda}^t = \boldsymbol{0}_m$ for all $t > T$;
- for any $i \in [n]$, there exists $t_i > T$ such that $x_i^{t_i} > 0$, and for any $j \in [m]$, there exists $t_j > T$ such that $y_j^{t_j} > 0$.

A Nash equilibrium is *non-degenerate* if $(A\boldsymbol{x}^*)_j < \nu^*$ whenever $\boldsymbol{y}_j^* = 0$ and $(A^\top \boldsymbol{y}^*)_i > \nu^*$ whenever $\boldsymbol{x}_i^* = 0$. With these notions, we present the following lemma.

**Lemma 9.** *If AltGDA eventually exhibits an interior cyclic trajectory in a bilinear game, then there cannot be any non-degenerate non-interior NE in the game.*

*Proof.* We prove this lemma by contradiction. Suppose that there exists a non-degenerate non-interior NE $(\boldsymbol{x}', \boldsymbol{y}')$. Let $\mathcal{E}'(\boldsymbol{x}^t, \boldsymbol{y}^t)$ be the energy function defined w.r.t. $(\boldsymbol{x}', \boldsymbol{y}')$ as in Eq. (Energy). Recall that, without assuming an interior NE, we have Eq. (22), i.e.,

$$\mathcal{E}'\left(\boldsymbol{x}^{t+1}, \boldsymbol{y}^{t+1}\right) - \mathcal{E}'\left(\boldsymbol{x}^t, \boldsymbol{y}^t\right) + 2\eta\langle\nu^*\boldsymbol{1}_m - A\boldsymbol{x}', \boldsymbol{y}^t\rangle + 2\eta\langle A^\top\boldsymbol{y}' - \nu^*\boldsymbol{1}_n, \boldsymbol{x}^{t+1}\rangle$$
$$+ \eta\langle\boldsymbol{\gamma}^t, \boldsymbol{x}^{t+1} + \boldsymbol{x}^t - 2\boldsymbol{x}'\rangle + \eta\langle\boldsymbol{\lambda}^t, \boldsymbol{y}^{t+1} + \boldsymbol{y}^t - 2\boldsymbol{y}'\rangle = 0. \tag{29}$$

Given that $\boldsymbol{\gamma}^t = \boldsymbol{0}_n, \boldsymbol{\lambda}^t = \boldsymbol{0}_m$ for all $t > T$, and $A\boldsymbol{x}' \le \nu^*\boldsymbol{1}_m$ and $A^\top\boldsymbol{y}' \ge \nu^*\boldsymbol{1}_n$, we have
$$\mathcal{E}'\left(\boldsymbol{x}^{t+1}, \boldsymbol{y}^{t+1}\right) \le \mathcal{E}'\left(\boldsymbol{x}^t, \boldsymbol{y}^t\right) \quad \forall t > T.$$

Further, with a non-degenerate non-interior NE there has to be at least an $i'$ or $j'$ such that $(A^\top\boldsymbol{y}')_{i'} > \nu^*$ or $(A\boldsymbol{x}')_{j'} < \nu^*$. Because of the third items in the definition of "interior cyclic trajectory", there has to be at least one iteration per period in which
$$\eta\langle\nu^*\boldsymbol{1}_m - A\boldsymbol{x}', \boldsymbol{y}^t\rangle + \eta\langle A^\top\boldsymbol{y}' - \nu^*\boldsymbol{1}_n, \boldsymbol{x}^{t+1}\rangle > 0.$$
Then, by the second item in the definition of interior cyclic trajectory, we obtain that
$$\mathcal{E}\left(\boldsymbol{x}^t, \boldsymbol{y}^t\right) < \mathcal{E}\left(\boldsymbol{x}^{t+s}, \boldsymbol{y}^{t+s}\right) \quad \forall t > T,$$
which contradicts the first item in the definition of interior cyclic trajectory. $\square$

## C.2 AN ADAPTIVE STEPSIZE RULE WITHOUT KNOWING THE NE

In this subsection, we design an adaptive stepsize rule that searches for an admissible stepsize. With this rule, even without knowing any interior NE, we can still set up the AltGDA algorithm and achieves $O(1/T)$ convergence rate. Additionally, this rule allows us to start with an initial stepsize that is potentially much larger than the theoretical one, therefore might lead to better performances in practice. We present the AltGDA algorithm equipped with this adaptive stepsize rule in Algorithm 2.

**Theorem 3.** *Assume that the bilinear game admits an interior NE. Let $\{(\boldsymbol{x}^t, \boldsymbol{y}^t)\}_{t=0,1,\ldots}$ be a sequence of iterates generated by Algorithm 2 with an initial stepsize $\eta^0 \le \frac{1}{\|A\|_2}$. Then, we have*

$$\texttt{DualityGap}\left(\frac{1}{T}\sum_{t=1}^T \boldsymbol{x}^t, \frac{1}{T}\sum_{t=1}^T \boldsymbol{y}^t\right) \le \frac{C}{\eta^* T}, \tag{30}$$

*where $\eta^* = \frac{1}{\|A\|_2} \min\left\{\min_{i\in[n]} x_i^*, \min_{j\in[m]} y_j^*\right\}$, $(\boldsymbol{x}^*, \boldsymbol{y}^*)$ is any interior NE, and $C = \lceil\log_2\left(\eta^0/\eta^*\right)\rceil\left(9 + 5\eta^0\|A\|_2\right) + \left(18 + 8\eta^0\|A\|_2\right)$.*

---

**Algorithm 2** AltGDA with an adaptive stepsize rule

---

**input:** number of iterations $T$, initial step size $\eta^0 > 0$
**initialize:** $(\boldsymbol{x}^0, \boldsymbol{y}^0) \in \mathcal{X} \times \mathcal{Y}, \eta^t = \eta^0 \leq \frac{1}{\|A\|_2}, r\text{sum} = 0$
**for** $t = 0, \ldots, T - 1$ **do**
    $\boldsymbol{x}^{t+1} = \Pi_{\mathcal{X}}(\boldsymbol{x}^t - \eta^t A^\top \boldsymbol{y}^t)$
    $\boldsymbol{y}^{t+1} = \Pi_{\mathcal{Y}}(\boldsymbol{y}^t + \eta^t A \boldsymbol{x}^{t+1})$
    Compute $r_t$ via Eq. (5)
    $r\text{sum} = r\text{sum} + r_t$
    **if** $r\text{sum} > 8 + 2\eta^0 \|A\|_2$ **then**
        $\eta^{t+1} = \eta^t/2$
        $r\text{sum} = 0$
    **else**
        $\eta^{t+1} = \eta^t$
    **end if**
**end for**
**output:** $(\frac{1}{T} \sum_{t=1}^T \boldsymbol{x}^t, \frac{1}{T} \sum_{t=1}^T \boldsymbol{y}^t) \in \mathcal{X} \times \mathcal{Y}$

---

*Proof.* First, note that the stepsizes are monotonically non-increasing, thereby $\eta^t \leq \eta^0$ for all $t \geq 0$. Second, if we are using an admissible stepsize, i.e., a stepsize $\eta$ satisfying $\eta \leq \eta^*$. Then, by Lemma 2, the residual terms $r_t$ (defined in Eq. (5)) is summable and (by the same bounds as in Eq. (28))

$$\sum_{t=0}^T r_t \leq \mathcal{E}(\boldsymbol{x}^0, \boldsymbol{y}^0) - \mathcal{E}(\boldsymbol{x}^{T+1}, \boldsymbol{y}^{T+1}) \leq 8 + 2\eta^0 \|A\|_2.$$

Next, we denote $t_k$ as the $k$-th time point at which the stepsize shrinks (by $1/2$). We call the iterations from 0 to $t_1 - 1$ (inclusively) as the 0-th epoch, and the iterations from $t_k$ to $t_{k+1} - 1$ (inclusively) as the $k$-th epoch for $k = 1, 2, \ldots$. Let $\eta^k$ denote the stepsize across the $k$-th epoch. Let $K$ denote the total number of stepsize shrinkage before we find an admissible stepsize, i.e., $\eta^K = \hat{\eta} \leq \eta^*$. It then follows that $K \leq \lceil \log_2 (\eta^0/\eta^*) \rceil$ and the final stepsize $\hat{\eta} \geq \eta^*/2$. Then, we consider the following two cases:

(the first case) For all $k \leq K - 1$, i.e., for all $k$ such that $\eta^k > \eta^*$, we have

$$2 \sum_{t=t_k}^{t_{k+1}-1} \eta^k \left(\boldsymbol{y}^\top A \boldsymbol{x}^t - (\boldsymbol{y}^t)^\top A \boldsymbol{x}\right)$$

$$- \eta^k \left(\boldsymbol{y}^\top A \boldsymbol{x}^{t_k} - (\boldsymbol{y}^{t_k})^\top A \boldsymbol{x}\right) + \eta^k \left(\boldsymbol{y}^\top A \boldsymbol{x}^{t_{k+1}} - (\boldsymbol{y}^{t_{k+1}})^\top A \boldsymbol{x}\right)$$

$$= \sum_{t=t_k}^{t_{k+1}-1} \eta^k \left(\boldsymbol{y}^\top A \boldsymbol{x}^t - (\boldsymbol{y}^t)^\top A \boldsymbol{x}\right) + \eta^k \left(\boldsymbol{y}^\top A \boldsymbol{x}^{t+1} - (\boldsymbol{y}^{t+1})^\top A \boldsymbol{x}\right)$$

$$\leq \psi_{t_k}(\boldsymbol{x}, \boldsymbol{y}) - \psi_{t_{k+1}}(\boldsymbol{x}, \boldsymbol{y}) + \phi_{t_k}(\boldsymbol{x}, \boldsymbol{y}) - \phi_{t_{k+1}}(\boldsymbol{x}, \boldsymbol{y}) + \sum_{t=t_k}^{t_{k+1}-1} r_t \quad \text{(by Lemma 1)}$$

$$\leq \psi_{t_k}(\boldsymbol{x}, \boldsymbol{y}) - \psi_{t_{k+1}}(\boldsymbol{x}, \boldsymbol{y}) + \phi_{t_k}(\boldsymbol{x}, \boldsymbol{y}) - \phi_{t_{k+1}}(\boldsymbol{x}, \boldsymbol{y}) + (8 + 2\eta^0 \|A\|_2) + 2(\eta^0)^2 \|A\|_2^2$$

$$\leq 10 + 2\eta^k \|A\|_2 + (8 + 2\eta^0 \|A\|_2) + 2(\eta^0)^2 \|A\|_2^2, \tag{31}$$

where the last inequality follows by the similar bounds as in Eq. (28). To show the second to last inequality, first by the stepsize shrinkage condition, we have the accumulated residual terms are at most $8 + 2\eta^0 \|A\|_2 + \bar{r}$ where $r_t \leq \bar{r}$ for all $t$. Then, we obtain $\bar{r}$ as follows:

$$r_t = \eta^t \left\langle -A^\top \boldsymbol{y}^t, \boldsymbol{x}^{t+1} - \boldsymbol{x}^t \right\rangle + \eta^t \left\langle A \boldsymbol{x}^{t+1}, \boldsymbol{y}^{t+1} - \boldsymbol{y}^t \right\rangle - \left\|\boldsymbol{x}^{t+1} - \boldsymbol{x}^t\right\|_2^2 - \left\|\boldsymbol{y}^{t+1} - \boldsymbol{y}^t\right\|_2^2$$

$$\leq \eta^t \|A^\top \boldsymbol{y}^t\|_2 \|\boldsymbol{x}^{t+1} - \boldsymbol{x}^t\|_2 + \eta^t \|A \boldsymbol{x}^{t+1}\|_2 \|\boldsymbol{y}^{t+1} - \boldsymbol{y}^t\|_2$$

$$\leq \eta^t \|A^\top \boldsymbol{y}^t\|_2 \|\boldsymbol{x}^t - \eta^t A^\top \boldsymbol{y}^t - \boldsymbol{x}^t\|_2 + \eta^t \|A \boldsymbol{x}^{t+1}\|_2 \|\boldsymbol{y}^t + \eta^t A \boldsymbol{x}^{t+1} - \boldsymbol{y}^t\|_2$$

$$\leq 2(\eta^t)^2 \|A\|_2^2$$

$$\leq 2(\eta^0)^2 \|A\|_2^2,$$

where the last three inequalities follow by non-expansiveness, the forth item in Lemma 5, and $\eta^t \leq \eta^0$ for all $t \geq 0$. It then follows by Eq. (31) and the third item in Lemma 5 that

$$\sum_{t=t_k}^{t_{k+1}-1} \left( \boldsymbol{y}^\top A \boldsymbol{x}^t - (\boldsymbol{y}^t)^\top A \boldsymbol{x} \right) \leq \frac{C_k}{2\eta^k} \leq \frac{C_k}{2\eta^*},$$

where $C_k = 10 + 2\eta^k \|A\|_2 + (8 + 2\eta^0 \|A\|_2) + 2(\eta^0)^2 \|A\|_2^2 + 4\eta^k \|A\|_2$.

(the second case) For $k = K$, i.e., for all $t \geq t_K$ and $\eta^t = \hat{\eta} \leq \eta^*$, the regret will remain finitely bounded by the same proof as in the proof of Theorem 1. Specifically, we have

$$\sum_{t=t_K}^{T} \left( \boldsymbol{y}^\top A \boldsymbol{x}^t - (\boldsymbol{y}^t)^\top A \boldsymbol{x} \right) \leq \frac{C(\boldsymbol{x}, \boldsymbol{y})}{2\hat{\eta}} \leq \frac{C(\boldsymbol{x}, \boldsymbol{y})}{\eta^*} \leq \frac{18 + 8\eta^0 \|A\|_2}{\eta^*}.$$

Let $\bar{C} = \max_{k=0,1,\ldots,K} C_k \leq 18 + 10\eta^0 \|A\|_2$. Combining the two cases, we have

$$\sum_{t=0}^{T} \left( \boldsymbol{y}^\top A \boldsymbol{x}^t - (\boldsymbol{y}^t)^\top A \boldsymbol{x} \right) \leq \left\lceil \log_2 \left( \frac{\eta^0}{\eta^*} \right) \right\rceil \frac{18 + 10\eta^0 \|A\|_2}{2\eta^*} + \frac{18 + 8\eta^0 \|A\|_2}{\eta^*}. \tag{32}$$

We completes the proof by dividing $T$ and taking maximum over $(\boldsymbol{x}, \boldsymbol{y}) \in \Delta_n \times \Delta_m$ on the both sides of Eq. (32). $\qquad \square$

## D  OMITTED PROOFS IN SECTION 6

In this section, we introduce some additional notations to facilitate the proof. We already define $I^* = \{i \in [n] \mid x_i^* > 0\}$ and $J^* = \{j \in [m] \mid y_j^* > 0\}$ in Section 5. Analogously, we denote $I^t = \{i \in [n] \mid x_i^t > 0\}$ and $J^t = \{j \in [m] \mid y_j^t > 0\}$ for all $t \geq 0$. For conciseness, for any $t \geq 0$, we introduce the following vectors to denote the "projected" gradients for a pair of $(\boldsymbol{x}^t, \boldsymbol{y}^t)$:

$$\begin{aligned}
\boldsymbol{v}^t &:= -A^\top \boldsymbol{y}^t + \frac{\sum_{\ell=1}^{n} (A^\top \boldsymbol{y}^t)_\ell}{n} \mathbf{1}_n, \\
\boldsymbol{u}^t &:= A \boldsymbol{x}^t - \frac{\sum_{\ell=1}^{m} (A \boldsymbol{x}^t)_\ell}{m} \mathbf{1}_m.
\end{aligned} \tag{33}$$

Note that $\sum_{i \in [n]} v_i^t = \sum_{j \in [m]} u_j^t = 0$. Recall that $\Pi_{\bar{\Delta}_d} (\boldsymbol{u} + \boldsymbol{g}) = \boldsymbol{u} + \boldsymbol{g} - \frac{1}{d} \left( \mathbf{1}_d^\top \boldsymbol{g} \right) \mathbf{1}_d$ for any $\boldsymbol{u} \in \bar{\Delta}_d$ and $\boldsymbol{g} \in \mathbb{R}^d$ (Beck, 2017, Lemma 6.26), thereby we have $\Pi_{\bar{\Delta}_n} \left( \boldsymbol{x}^t - \eta A^\top \boldsymbol{y}^t \right) = \boldsymbol{x}^t + \eta \boldsymbol{v}^t$ and $\Pi_{\bar{\Delta}_m} \left( \boldsymbol{y}^t + \eta A \boldsymbol{x}^{t+1} \right) = \boldsymbol{y}^t + \eta \boldsymbol{u}^{t+1}$. With $\boldsymbol{v}^t$ and $\boldsymbol{u}^t$, we can also write the nonsmooth parts of the iterate updates $\boldsymbol{\gamma}^t$ and $\boldsymbol{\lambda}^t$ defined in Eqs. (13) and (14) as follows:

$$\begin{aligned}
\boldsymbol{\gamma}^t &= \frac{\boldsymbol{x}^t + \eta \boldsymbol{v}^t - \boldsymbol{x}^{t+1}}{\eta} \\
\boldsymbol{\lambda}^t &= \frac{\boldsymbol{y}^t + \eta \boldsymbol{u}^{t+1} - \boldsymbol{y}^{t+1}}{\eta}.
\end{aligned} \tag{34}$$

Additionally, we define

$$\bar{\gamma}^t = \max_{i \in [n]} \gamma_i^t \quad \text{and} \quad \bar{\lambda}^t = \max_{j \in [m]} \lambda_j^t. \tag{35}$$

In this convention, the update rule of Algorithm 1 can be expressed as

$$\begin{aligned}
\boldsymbol{x}^{t+1} &= \boldsymbol{x}^t + \eta \boldsymbol{v}^t - \eta \boldsymbol{\gamma}^t \\
\boldsymbol{y}^{t+1} &= \boldsymbol{y}^t + \eta \boldsymbol{u}^{t+1} - \eta \boldsymbol{\lambda}^t.
\end{aligned} \tag{36}$$

We start the proof of the $O(1/T)$ local convergence rate with the following lemma. This lemma captures useful properties of $\boldsymbol{\gamma}^t$ and $\boldsymbol{\lambda}^t$.

**Lemma 10.** *For any $t \geq 0$, we have $\gamma_i^t = \bar{\gamma}^t \geq 0$ for all $i \in I^{t+1}$ and $\lambda_j^t = \bar{\lambda}^t \geq 0$ for all $j \in J^{t+1}$. Furthermore, if $\gamma_i^t \leq 0$ for some $i$ then $|\gamma_i^t| \leq |v_i^t|$, similarly, if $\lambda_j^t \leq 0$ for some $j$ then $|\lambda_j^t| \leq |u_j^{t+1}|$.*

*Proof.* Note that $\eta\boldsymbol{\gamma}^t = \boldsymbol{x}^t + \eta\boldsymbol{v}^t - \Pi_{\Delta_n}(\boldsymbol{x}^t + \eta\boldsymbol{v}^t)$. By the first-order optimality of the minimization problem corresponding to $\Pi_{\Delta_n}$, there exists a unique $\tau$ such that $x_i^{t+1} = \max\{x_i^t + \eta v_i^t - \tau, 0\}$ for all $i \in [n]$ (See, e.g., Page 77 in Held et al. (1974)). Note that $\tau \geq 0$ because

$$1 = \sum_{i\in[n]} x_i^{t+1} = \sum_{i\in[n]} \max\{x_i^t + \eta v_i^t - \tau, 0\} \geq \sum_{i\in[n]} (x_i^t + \eta v_i^t - \tau) = 1 - n\tau,$$

where we have used $\sum_{i\in[n]} v_i^t = 0$. It follows that $\eta\gamma_i^t = x_i^t + \eta v_i^t - \max\{x_i^t + \eta v_i^t - \tau, 0\} \leq x_i^t + \eta v_i^t - (x_i^t + \eta v_i^t - \tau) = \tau$ for all $i \in [n]$. Moreover, if $x_i^{t+1} > 0$ (i.e., $i \in I^{t+1}$), we have $x_i^{t+1} = x_i^t + \eta v_i^t - \tau$ thus $\eta\gamma_i^t = \tau$. As $\eta\gamma_i^t \leq \tau$ for all $i \in [n]$ and $\eta\gamma_i^t = \tau$ for all $i \in I^{t+1}$, we have $\tau = \eta\bar{\gamma}^t$. Symmetrically, we can show $\lambda_j^t = \bar{\lambda}^t$ for all $j \in J^{t+1}$.

To show the second part of this lemma, we consider two cases: $\bar{\gamma}^t > 0$ and $\bar{\gamma}^t = 0$. We first assume $\bar{\gamma}^t > 0$. If $\gamma_i^t \leq 0$ for some $i$, then we have that $x_i^{t+1} = x_i^t + \eta v_i^t - \eta\gamma_i^t \geq x_i^t + \eta v_i^t$. On the other hand, because $x_i^{t+1} = \max\{x_i^t + \eta v_i^t - \eta\bar{\gamma}^t, 0\}$ and $x_i^t + \eta v_i^t - \eta\bar{\gamma}^t < x_i^t + \eta v_i^t$, we have $x_i^{t+1} = 0 = x_i^t + \eta v_i^t - \eta\gamma_i^t$ and therefore $x_i^t + \eta v_i^t \leq 0$. Since $x_i^t \geq 0$, we have $v_i^t \leq 0$. Also, it holds that $\eta\gamma_i^t = x_i^t + \eta v_i^t \geq \eta v_i^t$. This implies $|\gamma_i^t| \leq |v_i^t|$ as $\gamma_i^t \leq 0$ and $v_i^t \leq 0$. For the other case in which $\bar{\gamma}^t = 0$, by the definition of $\bar{\gamma}^t$ we have $\gamma_i^t \leq 0$ for all $i$. Then, we have $x_i^t + \eta v_i^t - \eta\gamma_i^t \geq x_i^t + \eta v_i^t$ for each $i \in [n]$ and

$$1 = \sum_{i\in[n]} x_i^{t+1} = \sum_{i\in[n]} x_i^t + \eta v_i^t - \eta\gamma_i^t \geq \sum_{i\in[n]} x_i^t + \eta v_i^t = 1,$$

which implies $\sum_{i\in[n]} \gamma_i^t = 0$. Because $\gamma_i^t \leq 0$ for all $i \in [n]$ when $\bar{\gamma}^t = 0$, we must have $\gamma_i^t = 0$ for every $i \in [n]$. Therefore, $|\gamma_i^t| \leq |v_i^t|$ holds trivially. Symmetrically, we can show $|\lambda_j^t| \leq |u_j^{t+1}|$ if $\lambda_j^t \leq 0$. $\qquad\square$

Recall that, the value of the game is denoted as $\nu^* = \min_i (A^\top\boldsymbol{y}^*)_i = \max_j (A\boldsymbol{x}^*)_j$ and the game-specific parameter is defined as

$$\delta = \min\left\{\min_{i\notin I^*} \frac{(A^\top\boldsymbol{y}^*)_i - \nu^*}{\|A\|_2}, \min_{j\notin J^*} \frac{\nu^* - (A\boldsymbol{x}^*)_j}{\|A\|_2}, \min_{i\in I^*} x_i^*, \min_{j\in J^*} y_i^*\right\}. \tag{37}$$

This parameter measures the gap between the suboptimal payoffs to the optimal payoff for the both players. In particular,

$$\begin{aligned}(A^\top\boldsymbol{y}^*)_i &\geq \nu^* + \delta\|A\|_2 && \forall\, i \notin I^*, \\ (A\boldsymbol{x}^*)_j &\leq \nu^* - \delta\|A\|_2 && \forall\, j \notin J^*.\end{aligned} \tag{38}$$

Now, we present the proof of Lemma 3. In words, this lemma says that if the current iterate $(\boldsymbol{x}, \boldsymbol{y})$ is in $S$, then the components $x_i$ and $y_j$ corresponding to $I^*$ and $J^*$ are kept bounded away from zero; and other components monotonically decrease and approach zero. In a high level, this lemma provides the monotonicity we need to finish the proof.

**Lemma 3.** If the current iterate $(\boldsymbol{x}, \boldsymbol{y}) \in S$, and the next iterate $(\boldsymbol{x}^+, \boldsymbol{y}^+)$ is generated by Algorithm 1 with the stepsize $\eta \leq \frac{1}{2\|A\|_2}$, then we have

1. $x_i^+, x_i \geq \frac{\delta}{2}$ for all $i \in I^*$ and $y_j^+, y_j \geq \frac{\delta}{2}$ for all $j \in J^*$;
2. $x_i^+ \leq x_i$ for all $i \notin I^*$ and $y_j^+ \leq y_j$ for all $j \notin J^*$.

*Proof of Lemma 3.* To keep the presentation concise, we only prove the "$\boldsymbol{x}$" part; the "$\boldsymbol{y}$" part can be done symmetrically. Because $\|\boldsymbol{y} - \boldsymbol{y}^*\|_2 \leq \frac{\delta}{4}$, for all $i \in [n]$, we have

$$\left|-(A^\top\boldsymbol{y})_i + (A^\top\boldsymbol{y}^*)_i\right| \leq \left\|A^\top\boldsymbol{y}^* - A^\top\boldsymbol{y}\right\|_2 \leq \frac{\delta}{4}\|A\|_2. \tag{39}$$

As a result, for any $i, i' \in I^*$, we have $\nu^* = (A^\top\boldsymbol{y}^*)_i = (A^\top\boldsymbol{y}^*)_{i'}$, therefore

$$\left|-(A^\top\boldsymbol{y})_i + (A^\top\boldsymbol{y})_{i'}\right| \leq \left|-(A^\top\boldsymbol{y})_i + (A^\top\boldsymbol{y}^*)_i\right| + \left|-(A^\top\boldsymbol{y}^*)_{i'} + (A^\top\boldsymbol{y})_{i'}\right| \leq \frac{\delta}{2}\|A\|_2. \tag{40}$$

This further implies that

$$v_{i'} \le v_i + \frac{\delta}{2}\|A\|_2 \quad \forall\, i, i' \in I^*. \tag{41}$$

Moreover, for all $i \in I^*$ and $i' \notin I^*$,

$$(A^\top \boldsymbol{y})_i \overset{(39)}{\le} (A^\top \boldsymbol{y}^*)_i + \frac{\delta}{4}\|A\|_2 \overset{(38)}{\le} (A^\top \boldsymbol{y}^*)_{i'} - \delta\|A\|_2 + \frac{\delta}{4}\|A\|_2 \overset{(39)}{\le} (A^\top \boldsymbol{y})_{i'} - \delta\|A\|_2 + \frac{\delta}{2}\|A\|_2$$

$$= (A^\top \boldsymbol{y})_{i'} - \frac{\delta}{2}\|A\|_2.$$

Equivalently, $-(A^\top \boldsymbol{y})_{i'} \le -(A^\top \boldsymbol{y})_i - \frac{\delta}{2}\|A\|_2$ and therefore

$$v_{i'} \le v_i - \frac{\delta}{2}\|A\|_2 \le v_i \qquad \forall\, i \in I^*, i' \notin I^*. \tag{42}$$

Next, we show that $v_i - \gamma_i \ge -\frac{\delta}{2}\|A\|_2$ for all $i \in I^*$ by contradiction. Suppose otherwise, we have $v_i - \bar{\gamma} \le v_i - \gamma_i < -\frac{\delta}{2}\|A\|_2$ for some $i \in I^*$. Fix this $i \in I^*$. Then, for all $\ell \in [n]$ such that $x_\ell^+ > 0$, we have

$$x_\ell^+ = x_\ell + \eta v_\ell - \eta\bar{\gamma} \le x_\ell + \eta v_i + \frac{\delta}{2}\eta\|A\|_2 - \eta\bar{\gamma} < x_\ell,$$

where the first equality follows by Lemma 10, and the first inequality is implied by Eqs. (41) and (42). This leads to $\sum_{\ell \in [n]} x_\ell^+ = \sum_{\ell:x_\ell^+>0} x_\ell^+ < \sum_{\ell:x_\ell^+>0} x_\ell \le \sum_{\ell \in [n]} x_\ell = 1$ and thus a contradiction.

Then, we can prove the first part of the lemma. For each $i \in I^*$, since $v_i - \gamma_i \ge -\frac{\delta}{2}\|A\|_2$ and $0 < \eta \le \frac{1}{2\|A\|_2}$, we have $\eta v_i - \eta\gamma_i \ge -\frac{\delta}{4}$. On the other hand, since $|x_i - x_i^*| \le \|\boldsymbol{x} - \boldsymbol{x}^*\|_2 \le \frac{\delta}{4}$, we have $x_i \ge x_i^* - \frac{\delta}{4} \ge \delta - \frac{\delta}{4} = \frac{3\delta}{4}$ where the second inequality follows by the definition of $\delta$. Thus, $x_i^+ = x_i + \eta v_i - \eta\gamma_i \ge \frac{\delta}{2} > 0$ for all $i \in I^*$. By Lemma 10, $\gamma_i = \bar{\gamma}$ for all $i \in I^*$.

To show the second part, we first provide a lower bound for $\bar{\gamma}$. Observe that

$$0 = \sum_{i \in I^*} (x_i^+ - x_i) + \sum_{i \notin I^*} (x_i^+ - x_i) \ge \sum_{i \in I^*} (\eta v_i - \eta\bar{\gamma}) - (n - |I^*|)\frac{\eta\|A\|_2}{2}\frac{|I^*|}{n - |I^*|}\delta, \tag{43}$$

where the inequality follows by $\boldsymbol{x}^+ \ge 0$ and $\max_{i \notin I^*} x_i \le \frac{\eta\|A\|_2}{2}\frac{|I^*|}{n-|I^*|}\delta$. By rearranging terms, Eq. (43) yields

$$\bar{\gamma} \ge \frac{1}{|I^*|} \sum_{i \in I^*} v_i - \frac{\|A\|_2}{2}\delta \ge \min_{i \in I^*} v_i - \frac{\|A\|_2}{2}\delta \overset{(42)}{\ge} v_{i'} \qquad \forall\, i' \notin I^*.$$

Then, $x_i^+ \le x_i$ for each $i \notin I^*$ can be shown by contradiction: suppose that $x_i^+ > x_i \ge 0$, then by Lemma 10 we have $\gamma_i = \bar{\gamma}$ and therefore $x_i^+ = x_i + \eta v_i - \eta\bar{\gamma} \le x_i$, which is a contradiction. $\quad\square$

Recall that, for ease of presentation, we define a variant of the energy function, $\mathcal{V}: \Delta_n \times \Delta_m \to \mathbb{R}$, as follows:

$$\mathcal{V}(\boldsymbol{x}, \boldsymbol{y}) = \|\boldsymbol{x} - \boldsymbol{x}^*\|_2^2 + \|\boldsymbol{y} - \boldsymbol{y}^*\|_2^2 - \eta(\boldsymbol{y} - \boldsymbol{y}^*)^\top A(\boldsymbol{x} - \boldsymbol{x}^*).$$

For simplicity, we also use a shorthand notation

$$\mathcal{V}_t := \mathcal{V}(\boldsymbol{x}^t, \boldsymbol{y}^t). \tag{44}$$

Then, we derive an upper bound for the difference of this variant of the energy function.

**Lemma 11.** *Let $\mathcal{V}_t$ be defined as in Eq. (44), where $\{(\boldsymbol{x}^t, \boldsymbol{y}^t)\}_{t\ge0}$ be a sequence of iterates generated by Algorithm 1 with stepsize $\eta > 0$, then the change of the energy function per iteration satisfies that*

$$\Delta\mathcal{V}_t := \mathcal{V}_{t+1} - \mathcal{V}_t \le -\eta\langle \boldsymbol{\gamma}^t, \boldsymbol{x}^{t+1} + \boldsymbol{x}^t - 2\boldsymbol{x}^* \rangle - \eta\langle \boldsymbol{\lambda}^t, \boldsymbol{y}^{t+1} + \boldsymbol{y}^t - 2\boldsymbol{y}^* \rangle$$

$$\le -\eta\langle \boldsymbol{\gamma}^t, \boldsymbol{x}^t - \boldsymbol{x}^* \rangle - \eta\langle \boldsymbol{\lambda}^t, \boldsymbol{y}^t - \boldsymbol{y}^* \rangle. \tag{45}$$

*Proof.* By Eq. (36), we have

$$\langle \boldsymbol{x}^{t+1} - \boldsymbol{x}^t - \eta \boldsymbol{v}^t + \eta \boldsymbol{\gamma}^t \,, \, \boldsymbol{x}^{t+1} + \boldsymbol{x}^t - 2\boldsymbol{x}^* \rangle = 0$$
$$\langle \boldsymbol{y}^{t+1} - \boldsymbol{y}^t - \eta \boldsymbol{u}^{t+1} + \eta \boldsymbol{\lambda}^t \,, \, \boldsymbol{y}^{t+1} + \boldsymbol{y}^t - 2\boldsymbol{y}^* \rangle = 0. \tag{46}$$

By Eq. (33) and the fact that $\boldsymbol{x}^{t+1}, \boldsymbol{x}^t, \boldsymbol{x}^* \in \Delta_n$ and $\boldsymbol{y}^{t+1}, \boldsymbol{y}^t, \boldsymbol{y}^* \in \Delta_m$, we have $\langle \mathbf{1}_n \,, \, \boldsymbol{x}^{t+1} + \boldsymbol{x}^t - \boldsymbol{x}^* \rangle = \langle \mathbf{1}_m \,, \, \boldsymbol{y}^{t+1} + \boldsymbol{y}^t - 2\boldsymbol{y}^* \rangle = 0$, which leads to

$$\langle \boldsymbol{v}^t \,, \, \boldsymbol{x}^{t+1} + \boldsymbol{x}^t - 2\boldsymbol{x}^* \rangle = -\langle A^\top \boldsymbol{y}^t \,, \, \boldsymbol{x}^{t+1} + \boldsymbol{x}^t - 2\boldsymbol{x}^* \rangle$$
$$\langle \boldsymbol{u}^{t+1} \,, \, \boldsymbol{y}^{t+1} + \boldsymbol{y}^t - 2\boldsymbol{y}^* \rangle = \langle A\boldsymbol{x}^{t+1} \,, \, \boldsymbol{y}^{t+1} + \boldsymbol{y}^t - 2\boldsymbol{y}^* \rangle. \tag{47}$$

By using Eq. (47) and $\langle \boldsymbol{a} - \boldsymbol{b} \,, \, \boldsymbol{a} + \boldsymbol{b} \rangle = \|\boldsymbol{a}\|_2^2 - \|\boldsymbol{b}\|_2^2$ for any vectors $\boldsymbol{a}, \boldsymbol{b}$, one can see that Eq. (46) is equivalent to

$$\|\boldsymbol{x}^{t+1} - \boldsymbol{x}^*\|_2^2 - \|\boldsymbol{x}^t - \boldsymbol{x}^*\|_2^2 + \eta \langle A^\top \boldsymbol{y}^t \,, \, \boldsymbol{x}^{t+1} + \boldsymbol{x}^t - 2\boldsymbol{x}^* \rangle + \eta \langle \boldsymbol{\gamma}^t \,, \, \boldsymbol{x}^{t+1} + \boldsymbol{x}^t - 2\boldsymbol{x}^* \rangle = 0 \tag{48}$$

$$\|\boldsymbol{y}^{t+1} - \boldsymbol{y}^*\|_2^2 - \|\boldsymbol{y}^t - \boldsymbol{y}^*\|_2^2 - \eta \langle A\boldsymbol{x}^{t+1} \,, \, \boldsymbol{y}^{t+1} + \boldsymbol{y}^t - 2\boldsymbol{y}^* \rangle + \eta \langle \boldsymbol{\lambda}^t \,, \, \boldsymbol{y}^{t+1} + \boldsymbol{y}^t - 2\boldsymbol{y}^* \rangle = 0. \tag{49}$$

To derive the energy change between two consecutive iterates, we notice that

$$\eta \langle A^\top \boldsymbol{y}^t \,, \, \boldsymbol{x}^{t+1} + \boldsymbol{x}^t - 2\boldsymbol{x}^* \rangle - \eta \langle A\boldsymbol{x}^{t+1} \,, \, \boldsymbol{y}^{t+1} + \boldsymbol{y}^t - 2\boldsymbol{y}^* \rangle$$
$$= \eta \langle \boldsymbol{y}^t \,, \, A\boldsymbol{x}^t \rangle - 2\eta \langle \boldsymbol{y}^t \,, \, A\boldsymbol{x}^* \rangle - \eta \langle \boldsymbol{y}^{t+1} \,, \, A\boldsymbol{x}^{t+1} \rangle + 2\eta \langle \boldsymbol{y}^* \,, \, A\boldsymbol{x}^{t+1} \rangle$$
$$= \eta (\boldsymbol{y}^t - \boldsymbol{y}^*)^\top A(\boldsymbol{x}^t - \boldsymbol{x}^*) + \eta \langle \boldsymbol{y}^* \,, \, A\boldsymbol{x}^t \rangle - \eta \langle \boldsymbol{y}^* \,, \, A\boldsymbol{x}^* \rangle - \eta \langle \boldsymbol{y}^t \,, \, A\boldsymbol{x}^* \rangle$$
$$\qquad - \eta (\boldsymbol{y}^{t+1} - \boldsymbol{y}^*)^\top A(\boldsymbol{x}^{t+1} - \boldsymbol{x}^*) - \eta \langle \boldsymbol{y}^{t+1} \,, \, A\boldsymbol{x}^* \rangle + \eta \langle \boldsymbol{y}^* \,, \, A\boldsymbol{x}^* \rangle + \eta \langle \boldsymbol{y}^* \,, \, A\boldsymbol{x}^{t+1} \rangle$$
$$= \eta (\boldsymbol{y}^t - \boldsymbol{y}^*)^\top A(\boldsymbol{x}^t - \boldsymbol{x}^*) - \eta (\boldsymbol{y}^{t+1} - \boldsymbol{y}^*)^\top A(\boldsymbol{x}^{t+1} - \boldsymbol{x}^*)$$
$$\qquad\qquad + \eta \langle A^\top \boldsymbol{y}^* \,, \, \boldsymbol{x}^t + \boldsymbol{x}^{t+1} \rangle - \eta \langle A\boldsymbol{x}^* \,, \, \boldsymbol{y}^t + \boldsymbol{y}^{t+1} \rangle \tag{50}$$

and

$$\eta \langle A^\top \boldsymbol{y}^* \,, \, \boldsymbol{x}^t + \boldsymbol{x}^{t+1} \rangle - \eta \langle A\boldsymbol{x}^* \,, \, \boldsymbol{y}^t + \boldsymbol{y}^{t+1} \rangle$$
$$= \eta \langle A^\top \boldsymbol{y}^* - \nu^* \mathbf{1}_n \,, \, \boldsymbol{x}^t + \boldsymbol{x}^{t+1} \rangle + \eta \langle \nu^* \mathbf{1}_m - A\boldsymbol{x}^* \,, \, \boldsymbol{y}^t + \boldsymbol{y}^{t+1} \rangle \geq 0. \tag{51}$$

Summing up Eqs. (50) and (51), we have

$$\eta \langle A^\top \boldsymbol{y}^t \,, \, \boldsymbol{x}^{t+1} + \boldsymbol{x}^t - 2\boldsymbol{x}^* \rangle - \eta \langle A\boldsymbol{x}^{t+1} \,, \, \boldsymbol{y}^{t+1} + \boldsymbol{y}^t - 2\boldsymbol{y}^* \rangle$$
$$\geq \eta (\boldsymbol{y}^t - \boldsymbol{y}^*)^\top A(\boldsymbol{x}^t - \boldsymbol{x}^*) - \eta (\boldsymbol{y}^{t+1} - \boldsymbol{y}^*)^\top A(\boldsymbol{x}^{t+1} - \boldsymbol{x}^*). \tag{52}$$

Combining Eqs. (48), (49) and (52), and the definition of energy function $\mathcal{V}_t, \mathcal{V}_{t+1}$, we have

$$\mathcal{V}_{t+1} \leq \mathcal{V}_t - \eta \langle \boldsymbol{\gamma}^t \,, \, \boldsymbol{x}^{t+1} + \boldsymbol{x}^t - 2\boldsymbol{x}^* \rangle - \eta \langle \boldsymbol{\lambda}^t \,, \, \boldsymbol{y}^{t+1} + \boldsymbol{y}^t - 2\boldsymbol{y}^* \rangle.$$

Additionally, by Lemma 7, we further have Eq. (45). □

By leveraging Lemma 10, we can derive the following identities regarding the right-hand side of Eq. (45).

**Lemma 12.** *Let $\boldsymbol{\gamma}, \boldsymbol{\lambda}, \bar{\gamma}, \bar{\lambda}$ defined as in Eqs.* (34) *and* (35). *Then, we have*

$$\langle \boldsymbol{\gamma}^t \,, \, \boldsymbol{x}^t - \boldsymbol{x}^* \rangle = \sum_{i \notin I^{t+1}} (\gamma_i^t - \bar{\gamma}^t)(x_i^t - x_i^*)$$
$$\langle \boldsymbol{\lambda}^t \,, \, \boldsymbol{y}^t - \boldsymbol{y}^* \rangle = \sum_{j \notin J^{t+1}} (\lambda_j^t - \bar{\lambda}^t)(y_j^t - y_j^*). \tag{53}$$

*Proof.*

$$
\begin{aligned}
\langle \boldsymbol{\gamma}^t ,\, \boldsymbol{x}^t - \boldsymbol{x}^* \rangle =& \langle \boldsymbol{\gamma}^t ,\, \boldsymbol{x}^t - \boldsymbol{x}^{t+1} \rangle + \langle \boldsymbol{\gamma}^t ,\, \boldsymbol{x}^{t+1} - \boldsymbol{x}^* \rangle \\
=& \sum_{i \notin I^{t+1}} \gamma_i^t x_i^t + \sum_{i \in I^{t+1}} \gamma_i^t (x_i^t - x_i^{t+1}) + \langle \boldsymbol{\gamma}^t ,\, \boldsymbol{x}^{t+1} - \boldsymbol{x}^* \rangle \\
=& \sum_{i \notin I^{t+1}} \gamma_i^t x_i^t + \bar{\gamma}^t \sum_{i \in I^{t+1}} (x_i^t - x_i^{t+1}) + \langle \boldsymbol{\gamma}^t ,\, \boldsymbol{x}^{t+1} - \boldsymbol{x}^* \rangle \quad \text{(by Lemma 10)} \\
=& \sum_{i \notin I^{t+1}} \gamma_i^t x_i^t - \bar{\gamma}^t (1 - \sum_{i \in I^{t+1}} x_i^t) + \langle \boldsymbol{\gamma}^t ,\, \boldsymbol{x}^{t+1} - \boldsymbol{x}^* \rangle \\
& \hspace{4cm} \text{(by the definition of } I^{t+1}\text{)} \\
=& \sum_{i \notin I^{t+1}} \gamma_i^t x_i^t - \bar{\gamma}^t \sum_{i \notin I^{t+1}} x_i^t + \langle \boldsymbol{\gamma}^t ,\, \boldsymbol{x}^{t+1} - \boldsymbol{x}^* \rangle \\
=& \sum_{i \notin I^{t+1}} (\gamma_i^t - \bar{\gamma}^t) x_i^t + \langle \boldsymbol{\gamma}^t - \bar{\gamma}^t \mathbf{1}_n ,\, \boldsymbol{x}^{t+1} - \boldsymbol{x}^* \rangle \quad \text{(by } \langle \mathbf{1}_n ,\, \boldsymbol{x}^{t+1} - \boldsymbol{x}^* \rangle = 0\text{)} \\
=& \sum_{i \notin I^{t+1}} (\gamma_i^t - \bar{\gamma}^t) x_i^t - \sum_{i \notin I^{t+1}} (\gamma_i^t - \bar{\gamma}^t) x_i^* \\
& \hspace{4cm} \text{(by Lemma 10 and the definition of } I^{t+1}\text{)} \\
=& \sum_{i \notin I^{t+1}} (\gamma_i^t - \bar{\gamma}^t)(x_i^t - x_i^*). \quad\quad\quad\quad\quad\quad\quad\quad\quad\quad\quad (54)
\end{aligned}
$$

Symmetrically, we have $\langle \boldsymbol{\lambda}^t ,\, \boldsymbol{y}^t - \boldsymbol{y}^* \rangle = \sum_{j \notin J^{t+1}} (\lambda_j^t - \bar{\lambda}^t)(y_j^t - y_j^*)$. $\qquad\square$

If the game does not have an interior NE, then the right-hand side of Eq. (45) can be positive for some iterations. That said, the energy function is not monotonically decreasing. Even though, as shown below, by exploiting the local property we can derive that the sum of the energy increase has an upper bound, and hence we still obtains an $O(1/T)$ convergence rate.

In the rest of the proof, we provides the proof of Lemma 4 to formalize this idea, and then conclude the $O(1/T)$ convergence rate by an analogous argument as in the proof of Theorem 1.

Recall that

$$
S_0 = \left\{ (\boldsymbol{x}, \boldsymbol{y}) \,\middle|\, \|\boldsymbol{x} - \boldsymbol{x}^*\|_2 \le \frac{\delta}{8}, \ \|\boldsymbol{y} - \boldsymbol{y}^*\|_2 \le \frac{\delta}{8}, \ \max_{i \notin I^*} x_i \le \frac{c}{2} r_x \delta, \ \max_{j \notin J^*} y_j \le \frac{c}{2} r_y \delta \right\} \subset S
$$

where $c = \min\{\eta\|A\|_2, \frac{\delta}{192|I^*|}, \frac{\delta}{192|J^*|}\}$ always stay in $S$.

**Lemma 4.** Let $\{(\boldsymbol{x}^t, \boldsymbol{y}^t)\}_{t \ge 0}$ be a sequence of iterates generated by Algorithm 1 with stepsize $\eta \le \frac{1}{2\|A\|_2}$ and an initial point $(\boldsymbol{x}^0, \boldsymbol{y}^0) \in S_0$. Then, the iterates $\{(\boldsymbol{x}^t, \boldsymbol{y}^t)\}_{t \ge 0}$ stay within the local region $S$. Furthermore, for any $T > 0$, we have

$$
-\eta \sum_{t=0}^{T} \left( \langle \boldsymbol{\gamma}^t ,\, \boldsymbol{x}^t - \boldsymbol{x}^* \rangle + \langle \boldsymbol{\lambda}^t ,\, \boldsymbol{y}^t - \boldsymbol{y}^* \rangle \right) \le \frac{1}{128} \delta^2.
$$

*Proof of Lemma 4.* We prove the first part of this lemma by contradiction. Since $(\boldsymbol{x}^0, \boldsymbol{y}^0) \in S_0 \subset S$, by Lemma 3, as long as $(\boldsymbol{x}^{t'}, \boldsymbol{y}^{t'}) \in S$ for all $t' < t$, we have that

$$
x_i^t \le x_i^{t-1} \le x_i^0 \le \frac{\eta\|A\|}{2} r_x \delta, \quad \forall\, i \notin I^*,
$$

$$
y_j^t \le y_j^{t-1} \le y_i^0 \le \frac{\eta\|A\|}{2} r_y \delta, \quad \forall\, j \notin J^*.
$$

Suppose, to the contrary, that there exists a time point $t \ge 0$ such that $(\boldsymbol{x}^t, \boldsymbol{y}^t)$ leaves the region $S$ for the first time. Then, the above observation implies that at least one of $\|\boldsymbol{x}^t - \boldsymbol{x}^*\|_2 > \frac{\delta}{4}$ and

$\|\boldsymbol{y}^t - \boldsymbol{y}^*\|_2 > \frac{\delta}{4}$ happens. Therefore, the energy at the $t$-th iteration has the following lower bound:

$$
\begin{aligned}
\mathcal{V}_t &= \|\boldsymbol{x}^t - \boldsymbol{x}^*\|_2^2 + \|\boldsymbol{y}^t - \boldsymbol{y}^*\|_2^2 - \eta(\boldsymbol{y}^t - \boldsymbol{y}^*)^\top A(\boldsymbol{x}^t - \boldsymbol{x}^*) \\
&\geq \|\boldsymbol{x}^t - \boldsymbol{x}^*\|_2^2 + \|\boldsymbol{y}^t - \boldsymbol{y}^*\|_2^2 - \eta\|A\|_2\|\boldsymbol{x}^t - \boldsymbol{x}^*\|_2\|\boldsymbol{y}^t - \boldsymbol{y}^*\|_2 \\
&\geq \|\boldsymbol{x}^t - \boldsymbol{x}^*\|_2^2 + \|\boldsymbol{y}^t - \boldsymbol{y}^*\|_2^2 - \frac{\eta\|A\|_2}{2}\left(\|\boldsymbol{x}^t - \boldsymbol{x}^*\|_2^2 + \|\boldsymbol{y}^t - \boldsymbol{y}^*\|_2^2\right) \\
&\geq \frac{3}{4}\|\boldsymbol{x}^t - \boldsymbol{x}^*\|_2^2 + \frac{3}{4}\|\boldsymbol{y}^t - \boldsymbol{y}^*\|_2^2 \\
&> \frac{3}{4}\left(\frac{\delta}{4}\right)^2 = \frac{3}{64}\delta^2.
\end{aligned}
\tag{55}
$$

On the other hand, the initial energy is guaranteed to be sufficiently small. Let $\mathcal{V}_0$ be the initial energy corresponding to $(\boldsymbol{x}^0, \boldsymbol{y}^0)$. By definition, we have

$$
\begin{aligned}
\mathcal{V}_0 &= \|\boldsymbol{x}^0 - \boldsymbol{x}^*\|_2^2 + \|\boldsymbol{y}^0 - \boldsymbol{y}^*\|_2^2 - \eta(\boldsymbol{y}^0 - \boldsymbol{y}^*)^\top A(\boldsymbol{x}^0 - \boldsymbol{x}^*) \\
&\leq \|\boldsymbol{x}^0 - \boldsymbol{x}^*\|_2^2 + \|\boldsymbol{y}^0 - \boldsymbol{y}^*\|_2^2 + \eta\|A\|_2\|\boldsymbol{y}^0 - \boldsymbol{y}^*\|_2\|\boldsymbol{x}^0 - \boldsymbol{x}^*\|_2 \\
&\leq \left(\frac{\delta}{8}\right)^2 + \left(\frac{\delta}{8}\right)^2 + \eta\|A\|_2\left(\frac{\delta}{8}\right)^2 = \frac{2 + \eta\|A\|_2}{64}\delta^2 \leq \frac{5}{128}\delta^2.
\end{aligned}
\tag{56}
$$

By Lemma 11, we know the change of the energy function $\Delta\mathcal{V}_k$ is upper bounded by $-\eta\langle\boldsymbol{\gamma}^k, \boldsymbol{x}^k - \boldsymbol{x}^*\rangle - \eta\langle\boldsymbol{\lambda}^k, \boldsymbol{y}^k - \boldsymbol{y}^*\rangle$ for all $k \geq 0$. As $t$ denotes the first time at which the iterate leaves the local region $S$, for each $k = 0, \ldots, t-1$, we can further bound $\Delta\mathcal{V}_k$ as

$$
\begin{aligned}
\Delta\mathcal{V}_k &\leq -\eta\langle\boldsymbol{\gamma}^k, \boldsymbol{x}^k - \boldsymbol{x}^*\rangle - \eta\langle\boldsymbol{\lambda}^k, \boldsymbol{y}^k - \boldsymbol{y}^*\rangle && \text{(by Lemma 11)} \\
&= -\eta\sum_{i\notin I^{k+1}}(\gamma_i^k - \bar{\gamma}^k)(x_i^k - x_i^*) - \eta\sum_{j\notin J^{k+1}}(\lambda_j^k - \bar{\lambda}^k)(y_j^k - y_j^*) && \text{(by Lemma 12)} \\
&= -\eta\sum_{i:i\notin I^{k+1}, i\notin I^*}(\gamma_i^k - \bar{\gamma}^k)x_i^k - \eta\sum_{j:j\notin J^{k+1}, j\notin J^*}(\lambda_j^k - \bar{\lambda}^k)y_j^k && \text{(by Lemma 3)} \\
&= \eta\sum_{i:i\notin I^{k+1}, i\notin I^*}(\bar{\gamma}^k - \gamma_i^k)x_i^k + \eta\sum_{j:j\notin J^{k+1}, j\notin J^*}(\bar{\lambda}^k - \lambda_j^k)y_j^k
\end{aligned}
\tag{57}
$$

The first term in the right-hand side of Eq. (57) can be bounded as follows:

$$
\begin{aligned}
&\eta\sum_{i:i\notin I^{k+1}, i\notin I^*}(\bar{\gamma}^k - \gamma_i^k)x_i^k \\
&= \eta\sum_{i:i\notin I^{k+1}, i\notin I^*, \gamma_i^k>0}(\bar{\gamma}^k - \gamma_i^k)x_i^k + \eta\sum_{i:i\notin I^{k+1}, i\notin I^*, \gamma_i^k\leq 0}(\bar{\gamma}^k - \gamma_i^k)x_i^k \\
&\leq \eta\sum_{i:i\notin I^{k+1}, i\notin I^*, \gamma_i^k>0}\bar{\gamma}^k x_i^k + \eta\sum_{i:i\notin I^{k+1}, i\notin I^*, \gamma_i^k\leq 0}\left(\bar{\gamma}^k + |\gamma_i^k|\right)x_i^k.
\end{aligned}
\tag{58}
$$

To derive an upper bound for $\bar{\gamma}^k$, we observe that $(\boldsymbol{x}^k, \boldsymbol{y}^k) \in S$ for all $k \in [0, t-1]$. Thereby, Lemma 3 implies that $x_i^k > 0$ for all $i \in I^*$. Then, we have $x_i^{k+1} = x_i^k + \eta v_i^k - \eta\bar{\gamma}^k$, $\forall i \in I^*$. Summing up this equation over $i \in I^*$, we have

$$
\begin{aligned}
|I^*|\eta\bar{\gamma}^k &= \sum_{i\in I^*}(x_i^k - x_i^{k+1}) + \eta\sum_{i\in I^*}v_i^k \\
&\leq \sum_{i\in I^*}|x_i^k - x_i^{k+1}| + \eta\sum_{i\in I^*}|v_i^k| \\
&\leq |I^*|\|\boldsymbol{x}^{k+1} - \boldsymbol{x}^k\|_2 + \eta|I^*|\|\boldsymbol{v}^k\|_2 \\
&\leq 2|I^*|\eta\|A\|_2,
\end{aligned}
$$

where the last inequality holds because

$$
\begin{aligned}
\|\boldsymbol{x}^{k+1} - \boldsymbol{x}^k\|_2 &\overset{(a)}{\leq} \|\boldsymbol{x}^k - \eta A^\top\boldsymbol{y}^k - \boldsymbol{x}^k\|_2 \leq \eta\|A\|_2 \\
\|\boldsymbol{v}^k\|_2 &= \left\|-(A^\top\boldsymbol{y}^k) + \frac{1}{n}\sum_{\ell=1}^n (A^\top\boldsymbol{y}^k)_\ell\right\|_2 \leq \|A^\top\boldsymbol{y}^k\|_2 \leq \|A\|_2,
\end{aligned}
$$

where $(a)$ follows from nonexpansiveness of projection onto a closed convex set. Therefore, we obtain $\bar{\gamma}^k \leq 2\|A\|_2$. On the other hand, by Lemma 10, $|\gamma_i^k| \leq |v_i^k| \leq \|\boldsymbol{v}^k\|_2 \leq \|A\|_2$ for each $i$ such that $\gamma_i^t \leq 0$. Combining the above results, we have

$$\sum_{i:i\notin I^{k+1},i\notin I^*} (\bar{\gamma}^k - \gamma_i^k)x_i^k \leq 3\|A\|_2 \sum_{i:i\notin I^{k+1},i\notin I^*} x_i^k = 3\|A\|_2 \sum_{i:i\in I^k,i\notin I^{k+1},i\notin I^*} x_i^k.$$

Notice that, by Lemma 3, $x_i^{k+1} \leq x_i^k$ for all $i \notin I^*$ and $k \in [0, t-1]$. Hence, there is at most one $k \in [0, t-1]$ satisfying $i \in I^k, i \notin I^{k+1}$ for each $i \notin I^*$. Also, $x_i^k \leq \frac{c}{2}r_x\delta \leq \frac{r_x}{384|I^*|}\delta^2$ for all $i \notin I^*$ and $k \in [0, t-1]$. This translate to

$$\eta \sum_{k=0}^{t-1} \sum_{i:i\notin I^{k+1},i\notin I^*} (\bar{\gamma}^k - \gamma_i^k)x_i^k \leq \eta \sum_{k=0}^{t-1} \sum_{i:i\in I^k,i\notin I^{k+1},i\notin I^*} 3\|A\|_2 x_i^k$$

$$\leq \frac{1}{2\|A\|_2}(n - |I^*|)3\|A\|_2 \frac{r_x}{|I^*|}\frac{1}{384}\delta^2 = \frac{1}{256}\delta^2.$$

A symmetrical analysis gives us that

$$\eta \sum_{k=0}^{t-1} \sum_{j:j\notin J^{k+1},j\notin J^*} (\bar{\lambda}^k - \lambda_j^k)y_j^k \leq \frac{1}{2\|A\|_2}(m - |J^*|)3\|A\|_2 \frac{r_y}{|J^*|}\frac{1}{384}\delta^2 = \frac{1}{256}\delta^2.$$

Therefore, the change of energy up to $t$ is at most

$$\mathcal{V}_t - \mathcal{V}_0 = \sum_{k=0}^{t-1} \Delta\mathcal{V}_k \leq \eta \sum_{k=0}^{t-1} \left( \sum_{i:i\notin I^{k+1},i\notin I^*} (\bar{\gamma}^k - \gamma_i^k)x_i^k + \sum_{j:j\notin J^{k+1},j\notin J^*} (\bar{\lambda}^k - \lambda_j^k)y_j^k \right) \leq \frac{\delta^2}{128}.$$

$$(59)$$

This contradicts Eqs. (55) and (56).

Because $(\boldsymbol{x}^t, \boldsymbol{y}^t)$ for all $t \geq 0$, i.e., the condition in Lemma 3 is satisfied by all iterates generated by Algorithm 1 with stepsize $\eta \leq \frac{1}{2\|A\|_2}$ and an initial point $(\boldsymbol{x}^0, \boldsymbol{y}^0) \in S_0$, one can then verify that the upper bound in Eq. (59) still holds for an arbitrary $t \geq 0$ by the same derivation as above. In this way, the second part of this lemma follows. $\square$

**Theorem 2.** Let $\{(\boldsymbol{x}^t, \boldsymbol{y}^t)\}_{t\geq 0}$ be a sequence of iterates generated by Algorithm 1 with stepsize $\eta \leq \frac{1}{2\|A\|_2}$ and an initial point $(\boldsymbol{x}^0, \boldsymbol{y}^0) \in S_0$, where $S_0$ is defined in Eq. (7). Then, we have that

$$\texttt{DualityGap}\left(\frac{1}{T}\sum_{t=1}^{T}\boldsymbol{x}^t, \frac{1}{T}\sum_{t=1}^{T}\boldsymbol{y}^t\right) \leq \frac{9 + 7\eta\|A\|_2 + (\delta^2/128)}{\eta T}, \tag{60}$$

where $\delta$ is defined in Eq. (6).

*Proof of Theorem 2.* By Eq. (36), we have

$$\eta\langle -A^\top\boldsymbol{y}^t, \boldsymbol{x}^{t+1} - \boldsymbol{x}^t\rangle - \|\boldsymbol{x}^{t+1} - \boldsymbol{x}^t\|_2^2 = \langle -\eta A^\top\boldsymbol{y}^t - \boldsymbol{x}^{t+1} + \boldsymbol{x}^t, \boldsymbol{x}^{t+1} - \boldsymbol{x}^t\rangle$$

$$= \eta\langle \boldsymbol{\gamma}^t, \boldsymbol{x}^{t+1} - \boldsymbol{x}^t\rangle$$

$$= \eta\langle \boldsymbol{\gamma}^t, \boldsymbol{x}^{t+1} + \boldsymbol{x}^t - 2\boldsymbol{x}^*\rangle - 2\eta\langle \boldsymbol{\gamma}^t, \boldsymbol{x}^t - \boldsymbol{x}^*\rangle \quad (61)$$

$$\eta\langle A\boldsymbol{x}^{t+1}, \boldsymbol{y}^{t+1} - \boldsymbol{y}^t\rangle - \|\boldsymbol{y}^{t+1} - \boldsymbol{y}^t\|_2^2 = \langle \eta A\boldsymbol{x}^{t+1} - \boldsymbol{y}^{t+1} + \boldsymbol{y}^t, \boldsymbol{y}^{t+1} - \boldsymbol{y}^t\rangle$$

$$= \eta\langle \boldsymbol{\lambda}^t, \boldsymbol{y}^{t+1} - \boldsymbol{y}^t\rangle$$

$$= \eta\langle \boldsymbol{\lambda}^t, \boldsymbol{y}^{t+1} + \boldsymbol{y}^t - 2\boldsymbol{y}^*\rangle - 2\eta\langle \boldsymbol{\lambda}^t, \boldsymbol{y}^t - \boldsymbol{y}^*\rangle. \quad (62)$$

By Lemma 1 and Eqs. (61) and (62), we have

$$\eta\left(\boldsymbol{y}^\top A\boldsymbol{x}^{t+1} - (\boldsymbol{y}^{t+1})^\top A\boldsymbol{x}\right) + \eta\left(\boldsymbol{y}^\top A\boldsymbol{x}^t - (\boldsymbol{y}^t)^\top A\boldsymbol{x}\right)$$

$$\leq -\phi_{t+1}(\boldsymbol{x}, \boldsymbol{y}) + \phi_t(\boldsymbol{x}, \boldsymbol{y}) - \psi_{t+1}(\boldsymbol{x}, \boldsymbol{y}) + \psi_t(\boldsymbol{x}, \boldsymbol{y})$$

$$+ \eta\langle \boldsymbol{\gamma}^t, \boldsymbol{x}^{t+1} + \boldsymbol{x}^t - 2\boldsymbol{x}^*\rangle - 2\eta\langle \boldsymbol{\gamma}^t, \boldsymbol{x}^t - \boldsymbol{x}^*\rangle + \eta\langle \boldsymbol{\lambda}^t, \boldsymbol{y}^{t+1} + \boldsymbol{y}^t - 2\boldsymbol{y}^*\rangle - 2\eta\langle \boldsymbol{\lambda}^t, \boldsymbol{y}^t - \boldsymbol{y}^*\rangle.$$

By Lemma 11, for any $\boldsymbol{x}, \boldsymbol{y} \in \Delta_n \times \Delta_m$ and $t \geq 0$, we have:

$$\eta\left(\boldsymbol{y}^\top A \boldsymbol{x}^{t+1} - (\boldsymbol{y}^{t+1})^\top A \boldsymbol{x}\right) + \eta\left(\boldsymbol{y}^\top A \boldsymbol{x}^t - (\boldsymbol{y}^t)^\top A \boldsymbol{x}\right)$$
$$\leq -\phi_{t+1}(\boldsymbol{x}, \boldsymbol{y}) + \phi_t(\boldsymbol{x}, \boldsymbol{y}) - \psi_{t+1}(\boldsymbol{x}, \boldsymbol{y}) + \psi_t(\boldsymbol{x}, \boldsymbol{y}) + \mathcal{V}_t - \mathcal{V}_{t+1}$$
$$- 2\eta\langle \boldsymbol{\gamma}^t, \boldsymbol{x}^t - \boldsymbol{x}^*\rangle - 2\eta\langle \boldsymbol{\lambda}^t, \boldsymbol{y}^t - \boldsymbol{y}^*\rangle. \tag{63}$$

Recall that $\phi_t(\boldsymbol{x}, \boldsymbol{y}) := \frac{1}{2}\|\boldsymbol{x}^t - \boldsymbol{x}\|_2^2 + \frac{1}{2}\|\boldsymbol{y}^t - \boldsymbol{y}\|_2^2 + \eta(\boldsymbol{y}^t)^\top A \boldsymbol{x}$ and $\psi_t(\boldsymbol{x}, \boldsymbol{y}) := \frac{1}{2}\|\boldsymbol{x}^t - \boldsymbol{x}\|_2^2 + \frac{1}{2}\|\boldsymbol{y}^{t-1} - \boldsymbol{y}\|_2^2 - \frac{1}{2}\|\boldsymbol{y}^t - \boldsymbol{y}^{t-1}\|_2^2.$

Summing up Eq. (63) over $t = 1, \ldots, T$ plus Eq. (4) for $t = 0$, we have

$$2\eta \sum_{t=1}^T \left(\boldsymbol{y}^\top A \boldsymbol{x}^t - (\boldsymbol{y}^t)^\top A \boldsymbol{x}\right) + \eta\left(\boldsymbol{y}^\top A \boldsymbol{x}^{T+1} - (\boldsymbol{y}^{T+1})^\top A \boldsymbol{x}\right)$$

$$\leq \phi_1(\boldsymbol{x}, \boldsymbol{y}) - \phi_{T+1}(\boldsymbol{x}, \boldsymbol{y}) + \psi_1(\boldsymbol{x}, \boldsymbol{y}) - \psi_{T+1}(\boldsymbol{x}, \boldsymbol{y}) + \mathcal{V}_1 - \mathcal{V}_{T+1} + \frac{1}{64}\delta^2$$

$$+ \phi_0(\boldsymbol{x}, \boldsymbol{y}) - \phi_1(\boldsymbol{x}, \boldsymbol{y}) + \eta\langle A \boldsymbol{x}^1, \boldsymbol{y}^1 - \boldsymbol{y}^0\rangle - \frac{1}{2}\|\boldsymbol{x}^1 - \boldsymbol{x}^0\|_2^2 - \frac{1}{2}\|\boldsymbol{y}^1 - \boldsymbol{y}^0\|_2^2$$

$$\leq \phi_0(\boldsymbol{x}, \boldsymbol{y}) - \phi_{T+1}(\boldsymbol{x}, \boldsymbol{y}) + \psi_1(\boldsymbol{x}, \boldsymbol{y}) - \psi_{T+1}(\boldsymbol{x}, \boldsymbol{y}) + \mathcal{V}_1 - \mathcal{V}_{T+1} + \eta\langle A \boldsymbol{x}^1, \boldsymbol{y}^1 - \boldsymbol{y}^0\rangle + \frac{1}{64}\delta^2.$$

This inequality gives the following upper bound:

$$\boldsymbol{y}^\top A \left(\frac{1}{T}\sum_{t=1}^T \boldsymbol{x}^t\right) - \left(\frac{1}{T}\sum_{t=1}^T \boldsymbol{y}^t\right)^\top A \boldsymbol{x} = \frac{1}{T}\sum_{t=1}^T \left(\boldsymbol{y}^\top A \boldsymbol{x}^t - (\boldsymbol{y}^t)^\top A \boldsymbol{x}\right) \leq \frac{C(\boldsymbol{x}, \boldsymbol{y})}{2\eta T}, \tag{64}$$

where

$$C(\boldsymbol{x}, \boldsymbol{y}) = \phi_0(\boldsymbol{x}, \boldsymbol{y}) - \phi_{T+1}(\boldsymbol{x}, \boldsymbol{y}) + \psi_1(\boldsymbol{x}, \boldsymbol{y}) - \psi_{T+1}(\boldsymbol{x}, \boldsymbol{y}) + \mathcal{V}_1 - \mathcal{V}_{T+1} + \frac{1}{64}\delta^2$$
$$+ \eta\langle A \boldsymbol{x}^1, \boldsymbol{y}^1 - \boldsymbol{y}^0\rangle - \eta\left(\boldsymbol{y}^\top A \boldsymbol{x}^{T+1} - (\boldsymbol{y}^{T+1})^\top A \boldsymbol{x}\right)$$
$$\forall\, \boldsymbol{x}, \boldsymbol{y} \in \Delta_m \times \Delta_n.$$

For any $\boldsymbol{x} \in \Delta_n, \boldsymbol{y} \in \Delta_m$, we can bound each term in $C(\boldsymbol{x}, \boldsymbol{y})$ as follows:

$$\phi_0(\boldsymbol{x}, \boldsymbol{y}) = \frac{1}{2}\|\boldsymbol{x}^0 - \boldsymbol{x}\|_2^2 + \frac{1}{2}\|\boldsymbol{y}^0 - \boldsymbol{y}\|_2^2 + \eta(\boldsymbol{y}^0)^\top A \boldsymbol{x} \leq 4 + \eta\|A\|_2,$$

$$-\phi_{T+1}(\boldsymbol{x}, \boldsymbol{y}) = -\frac{1}{2}\|\boldsymbol{x}^{T+1} - \boldsymbol{x}\|_2^2 - \frac{1}{2}\|\boldsymbol{y}^{T+1} - \boldsymbol{y}\|_2^2 - \eta(\boldsymbol{y}^{T+1})^\top A \boldsymbol{x} \leq \eta\|A\|_2,$$

$$\psi_1(\boldsymbol{x}, \boldsymbol{y}) = \frac{1}{2}\|\boldsymbol{x}^1 - \boldsymbol{x}\|_2^2 + \frac{1}{2}\|\boldsymbol{y}^0 - \boldsymbol{y}\|_2^2 - \frac{1}{2}\|\boldsymbol{y}^1 - \boldsymbol{y}^0\|_2^2 \leq 4,$$

$$-\psi_{T+1}(\boldsymbol{x}, \boldsymbol{y}) = -\frac{1}{2}\|\boldsymbol{x}^{T+1} - \boldsymbol{x}\|_2^2 - \frac{1}{2}\|\boldsymbol{y}^T + \boldsymbol{y}\|_2^2 + \frac{1}{2}\|\boldsymbol{y}^{T+1} - \boldsymbol{y}^T\|_2^2 \leq 2,$$

$$\mathcal{V}_0 = \|\boldsymbol{x}^0 - \boldsymbol{x}^*\|_2^2 + \|\boldsymbol{y}^0 - \boldsymbol{y}^*\|_2^2 - \eta(\boldsymbol{y}^0 - \boldsymbol{y}^*)^\top A(\boldsymbol{x}^0 - \boldsymbol{x}^*) \leq 8 + 4\eta\|A\|_2,$$

$$-\mathcal{V}_{T+1} = -\|\boldsymbol{x}^{T+1} - \boldsymbol{x}^*\|_2^2 - \|\boldsymbol{y}^{T+1} - \boldsymbol{y}^*\|_2^2 + \eta(\boldsymbol{y}^{T+1} - \boldsymbol{y}^*)^\top A(\boldsymbol{x}^0 - \boldsymbol{x}^*)$$
$$\leq 4\eta\|A\|_2,$$

and $-\eta\langle A \boldsymbol{x}^1, \boldsymbol{y}^1 - \boldsymbol{y}^0\rangle - \eta\left(\boldsymbol{y}^\top A \boldsymbol{x}^{T+1} - (\boldsymbol{y}^{T+1})^\top A \boldsymbol{x}\right) \leq 4\eta\|A\|_2$, where all the inequalities follow by Lemma 5. Therefore, we can bound $C(\boldsymbol{x}, \boldsymbol{y})$ by $18 + 14\eta\|A\|_2 + \delta^2/64$. By taking the maximum on the both sides of Eq. (64), we complete the proof. $\qquad\square$

## E  SDP FORMULATION OF (INNER)

In this section, we reformulate the inner problem (INNER) as a convex SDP by using results from (Taylor et al., 2017a; Bousselmi et al., 2024). We use the following notation: write $\odot(\boldsymbol{x}, \boldsymbol{y}) = (\boldsymbol{x}\boldsymbol{y}^\top + \boldsymbol{y}\boldsymbol{x}^\top)/2$ to denote the symmetric outer product between the vectors $\boldsymbol{x}, \boldsymbol{y} \in \mathbb{R}^d$. For a symmetric matrix $M \succeq 0$ means that $M$ is positive semidefinite.

**Span based form of AltGDA.** First, we present an equivalent form of AltGDA, which we will use in our transformation to keep the resultant formulation in a compact form by decoupling the iterates and their interaction with $A$. To that goal, we first recall the following definition.

**Definition 1** (Indicator function and normal cone of a set.). *For any set $\mathcal{S} \subseteq \mathbb{R}^n$, its indicator function $\delta_{\mathcal{S}}(\boldsymbol{x})$ is 0 if $\boldsymbol{x} \in \mathcal{S}$ and is $\infty$ if $\boldsymbol{x} \notin \mathcal{S}$. For a closed convex set $\mathcal{C} \subseteq \mathbb{R}^n$, the subdifferential of its indicator function (also called normal cone), denoted by $\partial\delta_{\mathcal{C}}$, satisfies:*

$$\partial\delta_{\mathcal{C}}(\boldsymbol{x}) = \begin{cases} \emptyset & \text{if } \boldsymbol{x} \notin \mathcal{C} \\ \{\boldsymbol{y} \mid \boldsymbol{y}^{\top}(\boldsymbol{z} - \boldsymbol{x}) \leq 0 \text{ for all } \boldsymbol{z} \in \mathcal{C}\} & \text{if } \boldsymbol{x} \in \mathcal{C}. \end{cases}$$

*Define an arbitrary element of $\partial\delta_{\mathcal{C}}(\boldsymbol{x})$ by $\delta'_{\mathcal{C}}(\boldsymbol{x})$.*

**Lemma 13** (Equivalent representation of AltGDA). *Algorithm 1 can be written equivalently as:*

$$\begin{aligned}
\boldsymbol{x}^t &= \boldsymbol{x}^0 - \sum_{j=1}^{t} \delta'_{\mathcal{X}}(\boldsymbol{x}^j) - \eta \sum_{j=0}^{t-1} \boldsymbol{q}^j, \quad t \in \{1, 2, \ldots, T\} \\
\boldsymbol{y}^t &= \boldsymbol{y}^0 - \sum_{j=1}^{t} \delta'_{\mathcal{Y}}(\boldsymbol{y}^j) + \eta \sum_{j=1}^{t} \boldsymbol{p}^j, \quad t \in \{1, 2, \ldots, T\} \\
\boldsymbol{p}^t &= A\boldsymbol{x}^t, \quad t \in \{1, 2, \ldots, T\} \\
\boldsymbol{q}^t &= A^{\top}\boldsymbol{y}^t, \quad t \in \{1, 2, \ldots, T\}.
\end{aligned} \quad (65)$$

*Proof.* Recall that for any closed convex set $\mathcal{C}$, we have $\boldsymbol{p} = \Pi_{\mathcal{C}}(\boldsymbol{x})$ if and only if $\boldsymbol{x} - \boldsymbol{p} = \delta'_{\mathcal{C}}(\boldsymbol{p})$ for some $\delta'_{\mathcal{C}}(\boldsymbol{p}) \in \partial\delta_{\mathcal{C}}(\boldsymbol{p})$ (Bauschke & Combettes, 2017, Proposition 6.47). Using this, we can write the $\boldsymbol{x}$-iterates of AltGDA as

$$\begin{aligned}
\boldsymbol{x}^{t+1} &= \Pi_{\mathcal{X}}\left(\boldsymbol{x}^t - \eta A^{\top}\boldsymbol{y}^t\right) \\
\Leftrightarrow \boldsymbol{x}^{t+1} &= \boldsymbol{x}^t - \delta'_{\mathcal{X}}(\boldsymbol{x}^{t+1}) - \eta A^{\top}\boldsymbol{y}^t \text{ for some } \delta'_{\mathcal{X}}(\boldsymbol{x}^{t+1}) \in \partial\delta_{\mathcal{X}}(\boldsymbol{x}^{t+1})
\end{aligned}$$

which can be expanded to

$$\boldsymbol{x}^t = \boldsymbol{x}^0 - \sum_{j=1}^{t} \delta'_{\mathcal{X}}(\boldsymbol{x}^j) - \eta \sum_{j=0}^{t-1} A^{\top}\boldsymbol{y}^j, \quad t \in \{1, 2, \ldots, T\}. \quad (66)$$

Similarly, we can write the $\boldsymbol{y}$-iterates of AltGDA as

$$\begin{aligned}
\boldsymbol{y}^{t+1} &= \Pi_{\mathcal{Y}}\left(\boldsymbol{y}^t + \eta A\boldsymbol{x}^{t+1}\right) \\
\Leftrightarrow \boldsymbol{y}^{t+1} &= \boldsymbol{y}^t - \delta'_{\mathcal{Y}}(\boldsymbol{y}^{t+1}) + \eta A\boldsymbol{x}^{t+1}, \text{ where } \delta'_{\mathcal{Y}}(\boldsymbol{y}^{t+1}) \in \partial\delta_{\mathcal{Y}}(\boldsymbol{y}^{t+1})
\end{aligned}$$

leading to:

$$\boldsymbol{y}^t = \boldsymbol{y}^0 - \sum_{j=1}^{t} \delta'_{\mathcal{Y}}(\boldsymbol{y}^j) + \eta \sum_{j=1}^{t} A\boldsymbol{x}^j \quad t \in \{1, 2, \ldots, T\}. \quad (67)$$

Finally, setting

$$\begin{aligned}
\boldsymbol{p}^t &= A\boldsymbol{x}^t, \quad t \in \{1, 2, \ldots, T\} \\
\boldsymbol{q}^t &= A^{\top}\boldsymbol{y}^t, \quad t \in \{0, 1, 2, \ldots, T\}
\end{aligned}$$

in (66) and (67), we arrive at (65). $\qquad\square$

**Infinite-dimensional inner maximization problem.** For notational convenience of indexing the variables, first we write $\boldsymbol{x} := \boldsymbol{x}^{\diamond}$, $\boldsymbol{y} := \boldsymbol{y}^{\diamond}$ and merely rewrite (INNER) as follows:

$$\mathcal{P}_T(\eta) = \begin{pmatrix} \underset{\substack{\mathcal{X}\subseteq\mathbb{R}^n,\mathcal{Y}\subseteq\mathbb{R}^m,A\in\mathbb{R}^{m\times n},\\ \{\boldsymbol{x}^t\}_{t\in\{\diamond,0,1,\ldots,T\}}\subseteq\mathbb{R}^n,\\ \{\boldsymbol{y}^t\}_{t\in\{\diamond,0,1,\ldots,T\}}\subseteq\mathbb{R}^m,\\ m,n\in\mathbb{N}.}}{\text{maximize}} \quad \frac{1}{T}\sum_{t=1}^T \left((\boldsymbol{y}^\diamond)^\top A\boldsymbol{x}^t - (\boldsymbol{y}^t)^\top A\boldsymbol{x}^\diamond\right) \\ \text{subject to} \\ \mathcal{X} \text{ is a convex compact set in } \mathbb{R}^n \text{ with radius } 1, \\ \mathcal{Y} \text{ is convex compact set in } \mathbb{R}^m \text{ with radius } 1, \\ A \in \mathbb{R}^{m\times n} \text{ has maximum singular value } 1, \\ \{(\boldsymbol{x}^t,\boldsymbol{y}^t)\}_{t\in\{1,2,\ldots,T\}} \text{ are generated by AltGDA with stepsize } \eta \\ \qquad\qquad\qquad \text{from initial point } (\boldsymbol{x}^0,\boldsymbol{y}^0) \in \mathcal{X}\times\mathcal{Y}, \\ (\boldsymbol{x}^\diamond,\boldsymbol{y}^\diamond) \in \mathcal{X}\times\mathcal{Y}. \end{pmatrix} \quad \text{(INNER)}$$

Using Lemma 13 and by denoting $\boldsymbol{p}^\diamond = A\boldsymbol{x}^\diamond$ and $\boldsymbol{q}^\diamond = A^\top\boldsymbol{y}^\diamond$, we can write (INNER) in the following infinite-dimensional form:

$$\mathcal{P}_T(\eta) = \begin{pmatrix} \underset{\substack{\mathcal{X}\subseteq\mathbb{R}^n,\mathcal{Y}\subseteq\mathbb{R}^m,A\in\mathbb{R}^{m\times n},\\ \{\boldsymbol{x}^t\}_{t\in\{\diamond,0,1,\ldots,T\}}\subseteq\mathbb{R}^n,\\ \{\boldsymbol{y}^t\}_{t\in\{\diamond,0,1,\ldots,T\}}\subseteq\mathbb{R}^m,\\ m,n\in\mathbb{N}.}}{\text{maximize}} \quad \frac{1}{T}\sum_{t=1}^T \left((\boldsymbol{q}^\diamond)^\top \boldsymbol{x}^t - (\boldsymbol{y}^t)^\top \boldsymbol{p}^\diamond\right) \\ \text{subject to} \\ \mathcal{X} \text{ is a convex compact set in } \mathbb{R}^n \text{ with radius } 1, \\ \mathcal{Y} \text{ is convex compact set in } \mathbb{R}^m \text{ with radius } 1, \\ \boldsymbol{x}^t = \boldsymbol{x}^0 - \sum_{j=1}^t \delta'_{\mathcal{X}}(\boldsymbol{x}^j) - \eta\sum_{j=0}^{t-1}\boldsymbol{q}^j, \quad t\in\{1,2,\ldots,T\} \\ \boldsymbol{y}^t = \boldsymbol{y}^0 - \sum_{j=1}^t \delta'_{\mathcal{Y}}(\boldsymbol{y}^j) + \eta\sum_{j=1}^t \boldsymbol{p}^j, \quad t\in\{1,2,\ldots,T\} \\ A \in \mathbb{R}^{m\times n} \text{ has maximum singular value } 1, \\ \boldsymbol{p}^t = A\boldsymbol{x}^t, \quad t\in\{\diamond,1,2,\ldots,T\} \\ \boldsymbol{q}^t = A^\top\boldsymbol{y}^t, \quad t\in\{\diamond,1,2,\ldots,T\}. \\ (\boldsymbol{x}^\diamond,\boldsymbol{y}^\diamond) \in \mathcal{X}\times\mathcal{Y}. \end{pmatrix} \quad (68)$$

**Interpolation argument.** We next convert the infinite-dimensional inner maximization problem (68) into a finite-dimensional (albeit still intractable) one with the following interpolation results. The core intuition behind these results is that a first-order algorithm such as AltGDA interacts with the infinite-dimensional objects $\mathcal{X}$, $\mathcal{Y}$, or $A$ only through the first-order information it observes at the iterates. Hence, under suitable conditions, it may be possible to reconstruct these objects from the iterates and their associated first-order information in such a way that, based solely on the first-order information, the algorithm cannot distinguish between the original infinite-dimensional object and the reconstructed one. The following lemmas show that such reconstruction is possible in our setup.

**Lemma 14** (Interpolation of a convex compact set with bounded radius.(Taylor et al., 2017a, Theorem 3.6)). *Let $\mathcal{I}$ be an index set and let $\{\boldsymbol{x}^i,\boldsymbol{g}^i\}_{i\in\mathcal{I}} \subseteq \mathbb{R}^d\times\mathbb{R}^d$. Then there exists a compact convex set $\mathcal{C}\subseteq\mathbb{R}^d$ with radius $R$ satisfying $\delta'_{\mathcal{C}}(\boldsymbol{x}^i) = \boldsymbol{g}^i$ for all $i\in\mathcal{I}$ if and only if*

$$(\boldsymbol{g}^j)^\top(\boldsymbol{x}^i - \boldsymbol{x}^j) \leq 0, \quad \forall i,j\in\mathcal{I}$$
$$\|\boldsymbol{x}^i\|_2^2 \leq R^2, \quad \forall i\in\mathcal{I}.$$

**Lemma 15** (Interpolation of a matrix with bounded singular value.(Bousselmi et al., 2024, Theorem 3.1)). *Consider the sets of pairs $\{(\boldsymbol{x}^i,\boldsymbol{p}^i)\}_{i\in\{1,2,\ldots,T_1\}} \subseteq \mathbb{R}^n\times\mathbb{R}^m$ and $\{(\boldsymbol{y}^j,\boldsymbol{q}^j)\}_{j\in\{1,2,\ldots,T_2\}} \subseteq \mathbb{R}^m\times\mathbb{R}^n$, and define the following matrices:*

$$X = [\boldsymbol{x}^1 \mid \boldsymbol{x}^2 \mid \ldots \mid \boldsymbol{x}^{T_1}] \in \mathbb{R}^{n\times T_1},$$
$$P = [\boldsymbol{p}^1 \mid \boldsymbol{p}^2 \mid \ldots \mid \boldsymbol{p}^{T_1}] \in \mathbb{R}^{m\times T_1},$$
$$Y = [\boldsymbol{y}^1 \mid \boldsymbol{y}^2 \mid \ldots \mid \boldsymbol{y}^{T_2}] \in \mathbb{R}^{m\times T_2},$$
$$Q = [\boldsymbol{q}^1 \mid \boldsymbol{q}^2 \mid \ldots \mid \boldsymbol{q}^{T_2}] \in \mathbb{R}^{n\times T_2}.$$

*Then there exists a matrix $A \in \mathbb{R}^{m \times n}$ with maximum singular value $\sigma_{\max}(A) \leq L$ such that $\boldsymbol{p}^i = A\boldsymbol{x}^i$ for all $i \in \{1, 2, \ldots, T_1\}$ and $\boldsymbol{q}^j = A^\top \boldsymbol{y}^j$ for all $j \in \{1, 2, \ldots, T_2\}$ if and only if*

$$
\begin{aligned}
X^\top Q &= P^\top Y, \\
L^2 X^\top X - P^\top P &\succeq 0, \\
L^2 Y^\top Y - Q^\top Q &\succeq 0.
\end{aligned}
$$

In order to apply Lemma 14 and Lemma 15 to (68), define the following for notational convenience:

$$
\begin{aligned}
&\text{index } \diamond \text{ is denoted by } -1, \\
&\mathcal{I}_T = \{-1, 0, 1, \ldots, T\}, \\
&\delta'_{\mathcal{X}}(\boldsymbol{x}^i) = \hat{\boldsymbol{f}}_i, \quad i \in \mathcal{I}_T, \\
&\delta'_{\mathcal{Y}}(\boldsymbol{y}^j) = \hat{\boldsymbol{h}}_i, \quad i \in \mathcal{I}_T, \\
&X = [\boldsymbol{x}^1 \mid \boldsymbol{x}^2 \mid \ldots \mid \boldsymbol{x}^T] \in \mathbb{R}^{n \times T}, \\
&P = [\boldsymbol{p}^1 \mid \boldsymbol{p}^2 \mid \ldots \mid \boldsymbol{p}^T] \in \mathbb{R}^{m \times T}, \\
&Y = [\boldsymbol{y}^1 \mid \boldsymbol{y}^2 \mid \ldots \mid \boldsymbol{y}^T] \in \mathbb{R}^{m \times T}, \\
&Q = [\boldsymbol{q}^1 \mid \boldsymbol{q}^2 \mid \ldots \mid \boldsymbol{q}^T] \in \mathbb{R}^{n \times T}.
\end{aligned}
$$

**Finite-dimensional inner maximization problem.**   Using Lemma 14 and Lemma 15 and the new notation above, we can reformulate (68) as:

$$
\mathcal{P}_T(\eta) = \begin{pmatrix}
\underset{\substack{\{\boldsymbol{x}^i, \hat{\boldsymbol{f}}_i, \boldsymbol{q}^i\}_{i \in \mathcal{I}_T} \subseteq \mathbb{R}^n, \\ \{\boldsymbol{y}^i, \hat{\boldsymbol{h}}_i, \boldsymbol{p}^i\}_{i \in \mathcal{I}_T} \subseteq \mathbb{R}^m, \\ m, n \in \mathbb{N}.}}{\text{maximize}} \quad \frac{1}{T} \sum_{i=1}^{T} \left( (\boldsymbol{q}^{-1})^\top \boldsymbol{x}^i - (\boldsymbol{y}^i)^\top \boldsymbol{p}^{-1} \right) \\
\text{subject to} \\
\hat{\boldsymbol{f}}_j^\top (\boldsymbol{x}^i - \boldsymbol{x}^j) \leq 0, \quad i, j \in \mathcal{I}_T, \\
\|\boldsymbol{x}^i\|_2^2 \leq 1, \quad i \in \mathcal{I}_T, \\
\hat{\boldsymbol{h}}_j^\top (\boldsymbol{y}^i - \boldsymbol{y}^j) \leq 0, \quad i, j \in \mathcal{I}_T, \\
\|\boldsymbol{y}^i\|_2^2 \leq 1, \quad i \in \mathcal{I}_T, \\
\boldsymbol{x}^i = \boldsymbol{x}^0 - \sum_{j=1}^{i} \hat{\boldsymbol{f}}_j - \eta \sum_{j=0}^{i-1} \boldsymbol{q}^j \quad i \in \{1, 2, \ldots, T\} \\
\boldsymbol{y}^i = \boldsymbol{y}^0 - \sum_{j=1}^{i} \hat{\boldsymbol{h}}_j + \eta \sum_{j=1}^{i} \boldsymbol{p}^j \quad i \in \{1, 2, \ldots, T\} \\
(\boldsymbol{x}^i)^\top \boldsymbol{q}^j = (\boldsymbol{p}^i)^\top \boldsymbol{y}^j, \quad i, j \in \mathcal{I}_T \\
X^\top X - P^\top P \succeq 0, \\
Y^\top Y - Q^\top Q \succeq 0.
\end{pmatrix} \tag{69}
$$

Note that the problem does not contain any infinite-dimensional variable anymore, however, it still is nonconvex and intractable due to terms such as $\hat{\boldsymbol{f}}_j^\top (\boldsymbol{x}^i - \boldsymbol{x}^j)$ and $\hat{\boldsymbol{h}}_j^\top (\boldsymbol{y}^i - \boldsymbol{y}^j)$ and presence of dimensions $m$ and $n$ as variables. Next, we show how (69) can be transformed into a semidefinite programming problem that is dimension-free without any loss.

**Grammian formulation.**   Next we formulate (INNER) into a finite-dimensional convex SDP in maximization form. Let

$$
\begin{aligned}
&H_{\boldsymbol{x}, \boldsymbol{q}} = [\boldsymbol{x}^{-1} \mid \boldsymbol{x}^0 \mid \hat{\boldsymbol{f}}_{-1} \mid \hat{\boldsymbol{f}}_0 \mid \hat{\boldsymbol{f}}_1 \mid \ldots \mid \hat{\boldsymbol{f}}_T \mid \boldsymbol{q}^{-1} \mid \boldsymbol{q}^0 \mid \boldsymbol{q}^1 \mid \ldots \mid \boldsymbol{q}^T] \in \mathbb{R}^{n \times (2T+6)}, \\
&G_{\boldsymbol{x}, \boldsymbol{q}} = H_{\boldsymbol{x}, \boldsymbol{q}}^\top H_{\boldsymbol{x}, \boldsymbol{q}} \in \mathbb{S}_+^{(2T+6)}, \\
&H_{\boldsymbol{y}, \boldsymbol{p}} = [\boldsymbol{y}^{-1} \mid \boldsymbol{y}^0 \mid \hat{\boldsymbol{h}}_{-1} \mid \hat{\boldsymbol{h}}_0 \mid \hat{\boldsymbol{h}}_1 \mid \ldots \mid \hat{\boldsymbol{h}}_T \mid \boldsymbol{p}^{-1} \mid \boldsymbol{p}^0 \mid \boldsymbol{p}^1 \mid \ldots \mid \boldsymbol{p}^T] \in \mathbb{R}^{m \times (2T+6)}, \\
&G_{\boldsymbol{y}, \boldsymbol{p}} = H_{\boldsymbol{y}, \boldsymbol{p}}^\top H_{\boldsymbol{y}, \boldsymbol{p}} \in \mathbb{S}_+^{2T+6},
\end{aligned}
$$

where $\mathbf{rank}\, G_{\boldsymbol{x},\boldsymbol{q}} \leq n$ and $\mathbf{rank}\, G_{\boldsymbol{y},\boldsymbol{p}} \leq m$, that becomes void when maximizing over $m, n$ as we do in (69). Next define the following notation to select the columns of $H_{\boldsymbol{x},\boldsymbol{q}}$ and $H_{\boldsymbol{y},\boldsymbol{p}}$:

$$\widetilde{\mathbf{x}}_{-1} = e_1 \in \mathbb{R}^{2T+6}, \widetilde{\mathbf{x}}_0 = e_2 \in \mathbb{R}^{2T+6},$$

$$\hat{\widetilde{\mathbf{f}}}_i = e_{i+4} \in \mathbb{R}^{2T+6} \text{ for } i \in \mathcal{I}_T,$$

$$\widetilde{\mathbf{q}}_i = e_{i+T+6} \in \mathbb{R}^{2T+6} \text{ for } i \in \mathcal{I}_T,$$

$$\widetilde{\mathbf{x}}_i = \widetilde{\mathbf{x}}_0 - \sum_{j=1}^{i} \hat{\widetilde{\mathbf{f}}}_j - \eta \sum_{j=0}^{i-1} \widetilde{\mathbf{q}}_j \in \mathbb{R}^{2T+6} \text{ for } i \in \{1, 2, \ldots, T\},$$

$$\mathbf{X} = [\widetilde{\mathbf{x}}_{-1} \mid \widetilde{\mathbf{x}}_0 \mid \widetilde{\mathbf{x}}_1 \mid \ldots \mid \widetilde{\mathbf{x}}_T] \in \mathbb{R}^{(2T+6)\times(T+2)}$$

$$\widetilde{\mathbf{y}}_{-1} = e_1 \in \mathbb{R}^{2T+6}, \widetilde{\mathbf{y}}_0 = e_2 \in \mathbb{R}^{2T+6},$$

$$\hat{\widetilde{\mathbf{h}}}_i = e_{i+4} \in \mathbb{R}^{2T+6} \text{ for } i \in \mathcal{I}_T,$$

$$\widetilde{\mathbf{p}}_i = e_{i+T+6} \in \mathbb{R}^{2T+6} \text{ for } i \in \mathcal{I}_T,$$

$$\widetilde{\mathbf{y}}_i = \widetilde{\mathbf{y}}_0 - \sum_{j=1}^{i} \hat{\widetilde{\mathbf{h}}}_j + \eta \sum_{j=1}^{i} \widetilde{\mathbf{p}}_j \in \mathbb{R}^{2T+6} \text{ for } i \in \{1, 2, \ldots, T\},$$

$$\mathbf{Y} = [\widetilde{\mathbf{y}}_{-1} \mid \widetilde{\mathbf{y}}_0 \mid \widetilde{\mathbf{y}}_1 \mid \ldots \mid \widetilde{\mathbf{y}}_T] \in \mathbb{R}^{(2T+6)\times(T+2)}.$$

Note that $\widetilde{\mathbf{x}}_i$ and $\widetilde{\mathbf{y}}_i$ depend linearly on the stepsize $\eta$ for $i \in \{1, 2, \ldots, T\}$. The notation above is defined so that for all $i \in \mathcal{I}_T$ we have

$$\boldsymbol{x}^i = H_{\boldsymbol{x},\boldsymbol{q}} \widetilde{\mathbf{x}}_i, \ \hat{\boldsymbol{f}}_i = H_{\boldsymbol{x},\boldsymbol{q}} \hat{\widetilde{\mathbf{f}}}_i, \ \boldsymbol{q}^i = H_{\boldsymbol{x},\boldsymbol{q}} \widetilde{\mathbf{q}}_i,$$

$$\boldsymbol{y}^i = H_{\boldsymbol{y},\boldsymbol{p}} \widetilde{\mathbf{y}}_i, \ \hat{\boldsymbol{h}}_i = H_{\boldsymbol{y},\boldsymbol{p}} \hat{\widetilde{\mathbf{h}}}_i, \ \boldsymbol{p}^i = H_{\boldsymbol{y},\boldsymbol{p}} \boldsymbol{p}^i,$$

leading to the identities:

$$\frac{1}{T} \sum_{i=1}^{T} \left( (\boldsymbol{q}^{-1})^\top \boldsymbol{x}^i - (\boldsymbol{y}^i)^\top \boldsymbol{p}^{-1} \right) = \frac{1}{T} \sum_{i=1}^{T} \left( \mathbf{tr}\, G_{\boldsymbol{x},\boldsymbol{q}} \odot (\widetilde{\mathbf{q}}_{-1}, \widetilde{\mathbf{x}}_i) - \mathbf{tr}\, G_{\boldsymbol{y},\boldsymbol{p}} \odot (\widetilde{\mathbf{y}}_i, \widetilde{\mathbf{p}}_{-1}) \right)$$

$$\hat{\boldsymbol{f}}_j^\top (\boldsymbol{x}^i - \boldsymbol{x}^j) = \mathbf{tr}\, G_{\boldsymbol{x},\boldsymbol{q}} \odot (\hat{\widetilde{\mathbf{f}}}_j, \widetilde{\mathbf{x}}_i - \widetilde{\mathbf{x}}_j), \ \hat{\boldsymbol{h}}_j^\top (\boldsymbol{y}^i - \boldsymbol{y}^j) = \mathbf{tr}\, G_{\boldsymbol{y},\boldsymbol{p}} \odot (\hat{\widetilde{\mathbf{h}}}_j, \widetilde{\mathbf{y}}_i - \widetilde{\mathbf{y}}_j),$$

$$\|\boldsymbol{x}^i\|_2^2 = \mathbf{tr}\, G_{\boldsymbol{x},\boldsymbol{q}} \odot (\widetilde{\mathbf{x}}_i, \widetilde{\mathbf{x}}_i), \ \|\boldsymbol{y}^i\|_2^2 = \mathbf{tr}\, G_{\boldsymbol{y},\boldsymbol{p}} \odot (\widetilde{\mathbf{y}}_i, \widetilde{\mathbf{y}}_i),$$

$$(\boldsymbol{x}^i)^\top \boldsymbol{q}^j - (\boldsymbol{p}^i)^\top \boldsymbol{y}^j = \mathbf{tr}\, G_{\boldsymbol{x},\boldsymbol{q}} \odot (\widetilde{\mathbf{x}}_i \odot \widetilde{\mathbf{q}}_j) - \mathbf{tr}\, G_{\boldsymbol{y},\boldsymbol{p}} \odot (\widetilde{\mathbf{p}}_i, \widetilde{\mathbf{y}}_j)$$

$$X^\top X - P^\top P = \mathbf{X}^\top G_{\boldsymbol{x},\boldsymbol{q}} \mathbf{X} - \mathbf{P}^\top G_{\boldsymbol{y},\boldsymbol{p}} \mathbf{P},$$

$$Y^\top Y - Q^\top Q = \mathbf{Y}^\top G_{\boldsymbol{y},\boldsymbol{p}} \mathbf{Y} - \mathbf{Q}^\top G_{\boldsymbol{x},\boldsymbol{q}} \mathbf{Q}.$$

Using these identities, we can formulate (69) as the following semidefinite optimization problem in maximization form:

$$\mathcal{P}_T(\eta) = \begin{pmatrix} \underset{\substack{G_{\boldsymbol{x},\boldsymbol{q}} \in \mathbb{S}^{2T+6} \\ G_{\boldsymbol{y},\boldsymbol{p}} \in \mathbb{S}^{2T+6}}}{\text{maximize}} \frac{1}{T} \sum_{i=1}^{T} \left( \mathbf{tr}\, G_{\boldsymbol{x},\boldsymbol{q}} \odot (\widetilde{\mathbf{q}}_{-1}, \widetilde{\mathbf{x}}_i) - \mathbf{tr}\, G_{\boldsymbol{y},\boldsymbol{p}} \odot (\widetilde{\mathbf{y}}_i, \widetilde{\mathbf{p}}_{-1}) \right) \\ \text{subject to} \\ \mathbf{tr}\, G_{\boldsymbol{x},\boldsymbol{q}} \odot (\hat{\widetilde{\mathbf{f}}}_j, \widetilde{\mathbf{x}}_i - \widetilde{\mathbf{x}}_j) \leq 0, \quad i, j \in \mathcal{I}_T, \\ \mathbf{tr}\, G_{\boldsymbol{x},\boldsymbol{q}} \odot (\widetilde{\mathbf{x}}_i, \widetilde{\mathbf{x}}_i) - 1 \leq 0, \quad i \in \mathcal{I}_T, \\ \mathbf{tr}\, G_{\boldsymbol{y},\boldsymbol{p}} \odot (\hat{\widetilde{\mathbf{h}}}_j, \widetilde{\mathbf{y}}_i - \widetilde{\mathbf{y}}_j) \leq 0, \quad i, j \in \mathcal{I}_T, \\ \mathbf{tr}\, G_{\boldsymbol{y},\boldsymbol{p}} \odot (\widetilde{\mathbf{y}}_i, \widetilde{\mathbf{y}}_i) - 1 \leq 0, \quad i \in \mathcal{I}_T, \\ \mathbf{tr}\, G_{\boldsymbol{x},\boldsymbol{q}} \odot (\widetilde{\mathbf{x}}_i, \widetilde{\mathbf{q}}_j) - \mathbf{tr}\, G_{\boldsymbol{y},\boldsymbol{p}} \odot (\widetilde{\mathbf{p}}_i, \widetilde{\mathbf{y}}_j) = 0, \quad i, j \in \mathcal{I}_T, \\ \mathbf{X}^\top G_{\boldsymbol{x},\boldsymbol{q}} \mathbf{X} - \mathbf{P}^\top G_{\boldsymbol{y},\boldsymbol{p}} \mathbf{P} \succeq 0, \\ \mathbf{Y}^\top G_{\boldsymbol{y},\boldsymbol{p}} \mathbf{Y} - \mathbf{Q}^\top G_{\boldsymbol{x},\boldsymbol{q}} \mathbf{Q} \succeq 0, \\ G_{\boldsymbol{x},\boldsymbol{q}} \succeq 0, G_{\boldsymbol{y},\boldsymbol{p}} \succeq 0. \end{pmatrix} \tag{70}$$

Note that this formulation does not contain dimensions $m, n$ anymore and is a tractable convex problem that can solved to global optimality to compute the convergence bound of AltGDA numerically for a given $\eta$ and finite $T$.

E.1 DETAILED NUMERICAL RESULTS

See Tables 2 and 3 for the detailed data values for Fig. 1.

Table 2: Optimized stepsizes and duality gaps given a time horizon of $T$ for AltGDA

| $T$ | Optimized $\eta$ | Optimized Duality Gap |
|---|---|---|
| 5 | 1.527 | 0.614 |
| 6 | 1.389 | 0.555 |
| 7 | 1.632 | 0.488 |
| 8 | 1.574 | 0.411 |
| 9 | 1.467 | 0.371 |
| 10 | 1.370 | 0.345 |
| 11 | 1.304 | 0.327 |
| 12 | 1.517 | 0.302 |
| 13 | 1.454 | 0.274 |
| 14 | 1.377 | 0.256 |
| 15 | 1.314 | 0.243 |
| 16 | 1.262 | 0.233 |
| 17 | 1.438 | 0.220 |
| 18 | 1.387 | 0.207 |
| 19 | 1.333 | 0.196 |
| 20 | 1.283 | 0.188 |
| 21 | 1.239 | 0.181 |
| 22 | 1.389 | 0.174 |
| 23 | 1.347 | 0.166 |
| 24 | 1.302 | 0.159 |
| 25 | 1.263 | 0.153 |
| 26 | 1.229 | 0.149 |
| 27 | 1.355 | 0.144 |
| 28 | 1.319 | 0.139 |
| 29 | 1.283 | 0.134 |
| 30 | 1.249 | 0.130 |
| 31 | 1.220 | 0.126 |
| 32 | 1.332 | 0.123 |
| 33 | 1.301 | 0.119 |
| 34 | 1.269 | 0.116 |
| 35 | 1.240 | 0.112 |
| 36 | 1.214 | 0.110 |
| 37 | 1.314 | 0.107 |
| 38 | 1.286 | 0.104 |
| 39 | 1.258 | 0.102 |
| 40 | 1.232 | 0.099 |
| 41 | 1.209 | 0.097 |
| 42 | 1.300 | 0.095 |
| 43 | 1.275 | 0.093 |
| 44 | 1.250 | 0.091 |
| 45 | 1.226 | 0.089 |
| 46 | 1.206 | 0.087 |
| 47 | 1.288 | 0.086 |
| 48 | 1.266 | 0.084 |
| 49 | 1.243 | 0.082 |
| 50 | 1.221 | 0.080 |

Table 3: Optimized stepsizes and duality gaps given a time horizon of $T$ for SimGDA

| $T$ | Optimized $\eta$ | Optimized Duality Gap |
|---|---|---|
| 5  | 1.989 | 1.238 |
| 6  | 1.450 | 1.150 |
| 7  | 1.165 | 1.072 |
| 8  | 1.018 | 1.009 |
| 9  | 0.877 | 0.958 |
| 10 | 0.769 | 0.916 |
| 11 | 0.684 | 0.880 |
| 12 | 0.616 | 0.850 |
| 13 | 0.567 | 0.823 |
| 14 | 0.527 | 0.801 |
| 15 | 0.492 | 0.781 |
| 16 | 0.466 | 0.763 |
| 17 | 0.440 | 0.747 |
| 18 | 0.417 | 0.733 |
| 19 | 0.398 | 0.721 |
| 20 | 0.379 | 0.710 |
| 21 | 0.362 | 0.699 |
| 22 | 0.347 | 0.690 |
| 23 | 0.333 | 0.681 |
| 24 | 0.320 | 0.673 |
| 25 | 0.308 | 0.665 |
| 26 | 0.298 | 0.658 |
| 27 | 0.487 | 0.654 |
| 28 | 0.472 | 0.643 |
| 29 | 0.456 | 0.633 |
| 30 | 0.443 | 0.623 |
| 31 | 0.431 | 0.613 |
| 32 | 0.416 | 0.604 |
| 33 | 0.406 | 0.596 |
| 34 | 0.394 | 0.588 |
| 35 | 0.384 | 0.580 |
| 36 | 0.373 | 0.573 |
| 37 | 0.363 | 0.565 |
| 38 | 0.353 | 0.559 |
| 39 | 0.345 | 0.552 |
| 40 | 0.335 | 0.546 |
| 41 | 0.326 | 0.539 |
| 42 | 0.318 | 0.533 |
| 43 | 0.310 | 0.528 |
| 44 | 0.303 | 0.522 |
| 45 | 0.296 | 0.517 |
| 46 | 0.289 | 0.511 |
| 47 | 0.284 | 0.506 |
| 48 | 0.278 | 0.501 |
| 49 | 0.272 | 0.497 |
| 50 | 0.266 | 0.492 |

# F  ADDITIONAL NUMERICAL EXPERIMENTS

## F.1  NUMERICAL PERFORMANCES: ALTGDA VERSUS SIMGDA

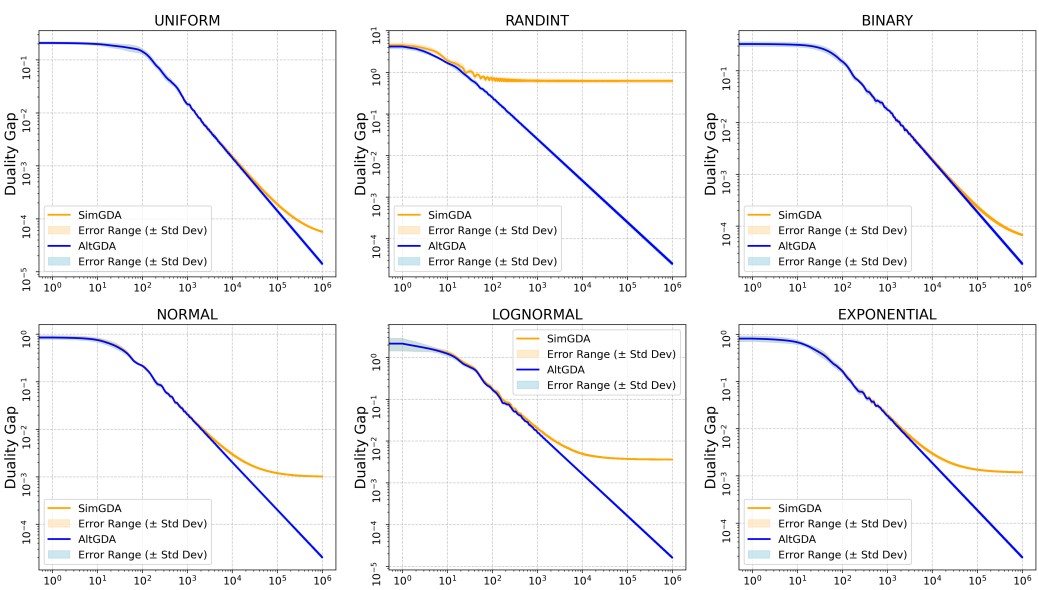

Figure 5: Numerical performances of AltGDA and SimGDA on $30 \times 60$ synthesized matrix games.

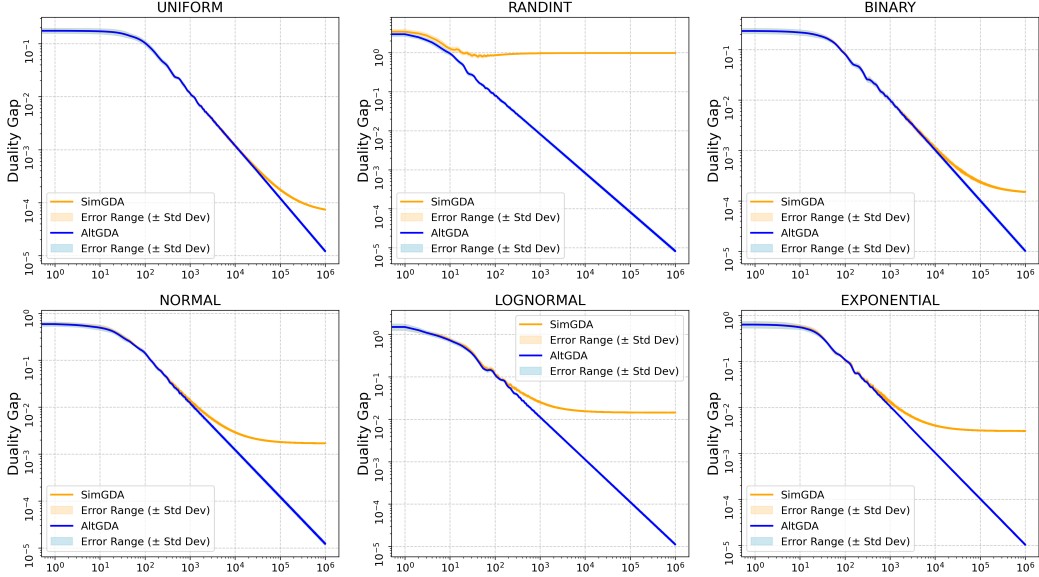

Figure 6: Numerical performances of AltGDA and SimGDA on $60 \times 120$ synthesized matrix games.

## F.2 NUMERICAL PERFORMANCES: ALTGDA WITH DIFFERENT STEPSIZES

We conduct the numerical experiments for AltGDA in the same setup as in the preceding subsection. For each instance, we run AltGDA with three different stepsizes: $\eta = 0.001$, $0.01$, and $0.1$.

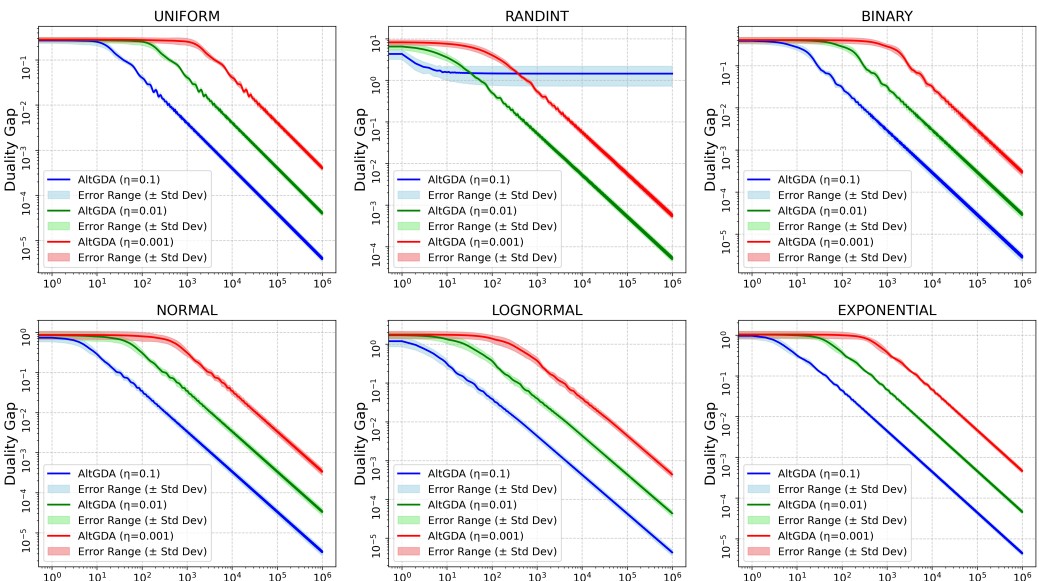

Figure 7: Numerical performances of AltGDA with different stepsizes on $10 \times 20$ synthesized matrix games.

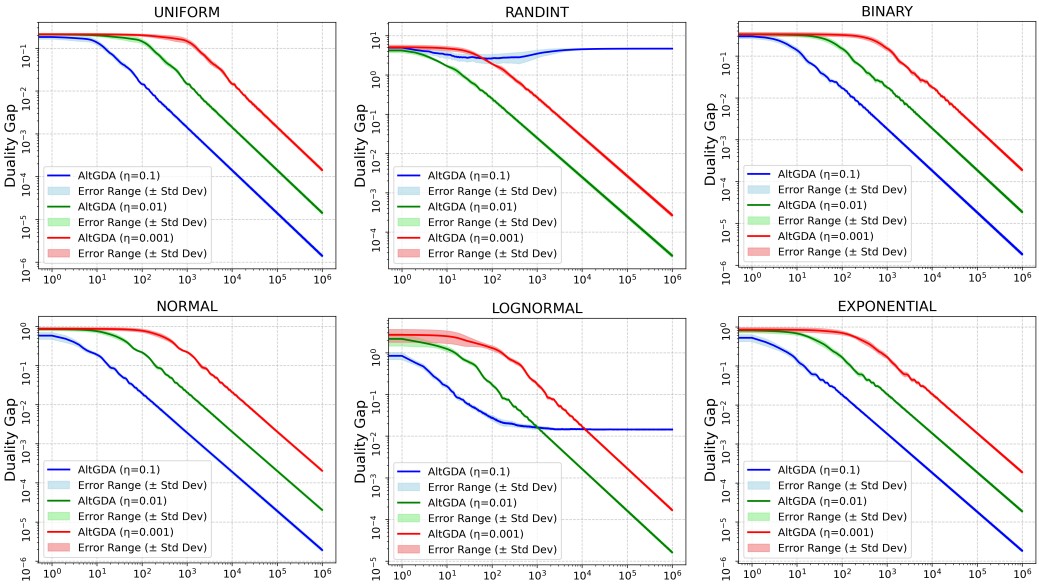

Figure 8: Numerical performances of AltGDA with different stepsizes on $30 \times 60$ synthesized matrix games.

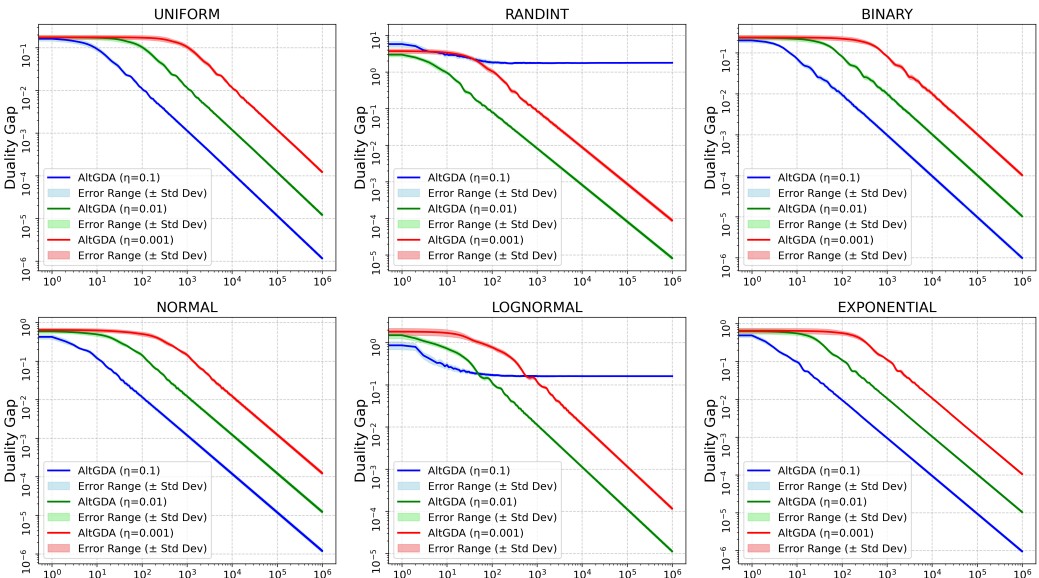

Figure 9: Numerical performances of AltGDA with different stepsizes on $60 \times 120$ synthesized matrix games.

## F.3 EVOLUTIONS OF FIG. 2 AND FIG. 3

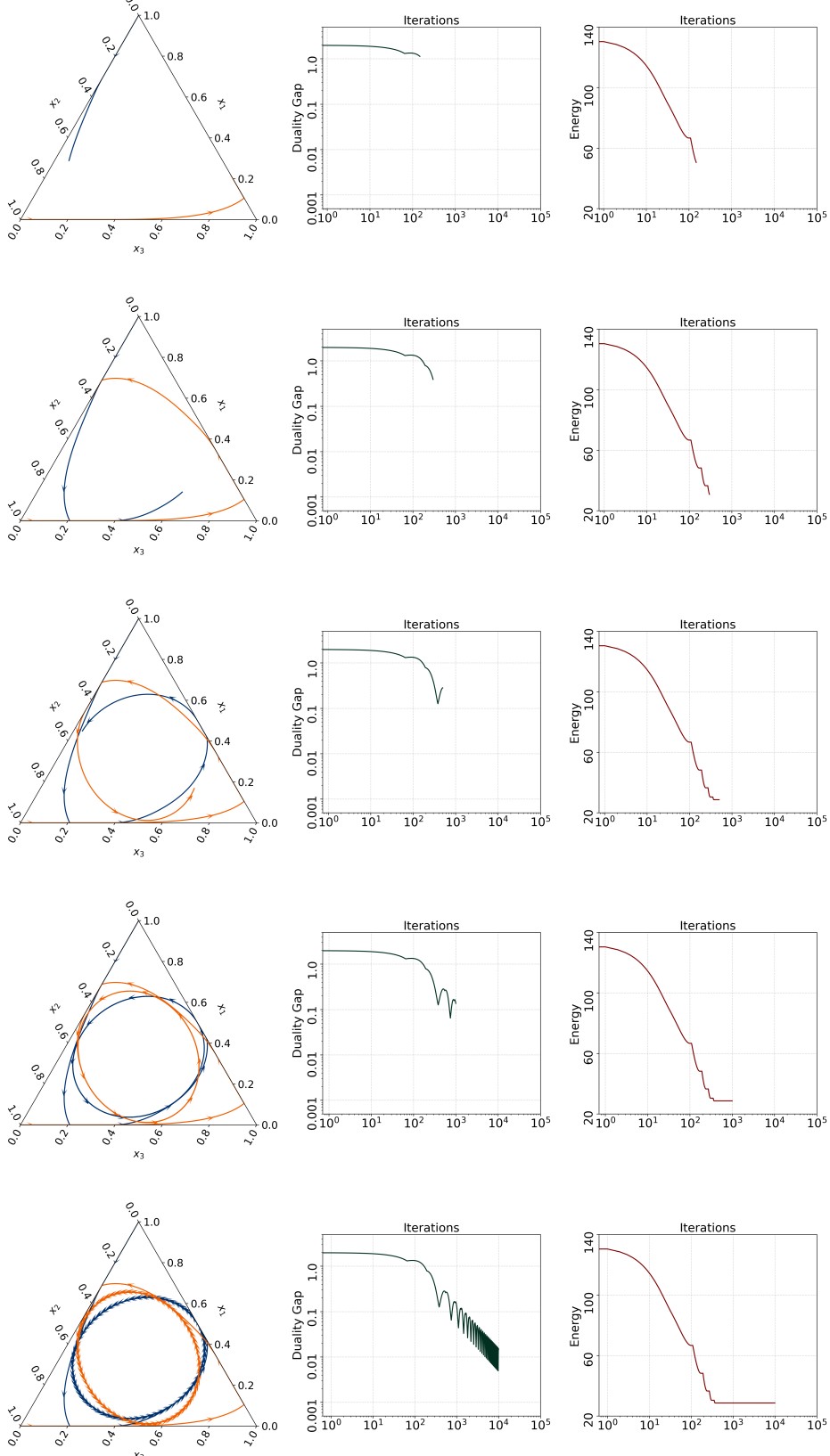

Figure 10: Evolution of Fig. 2

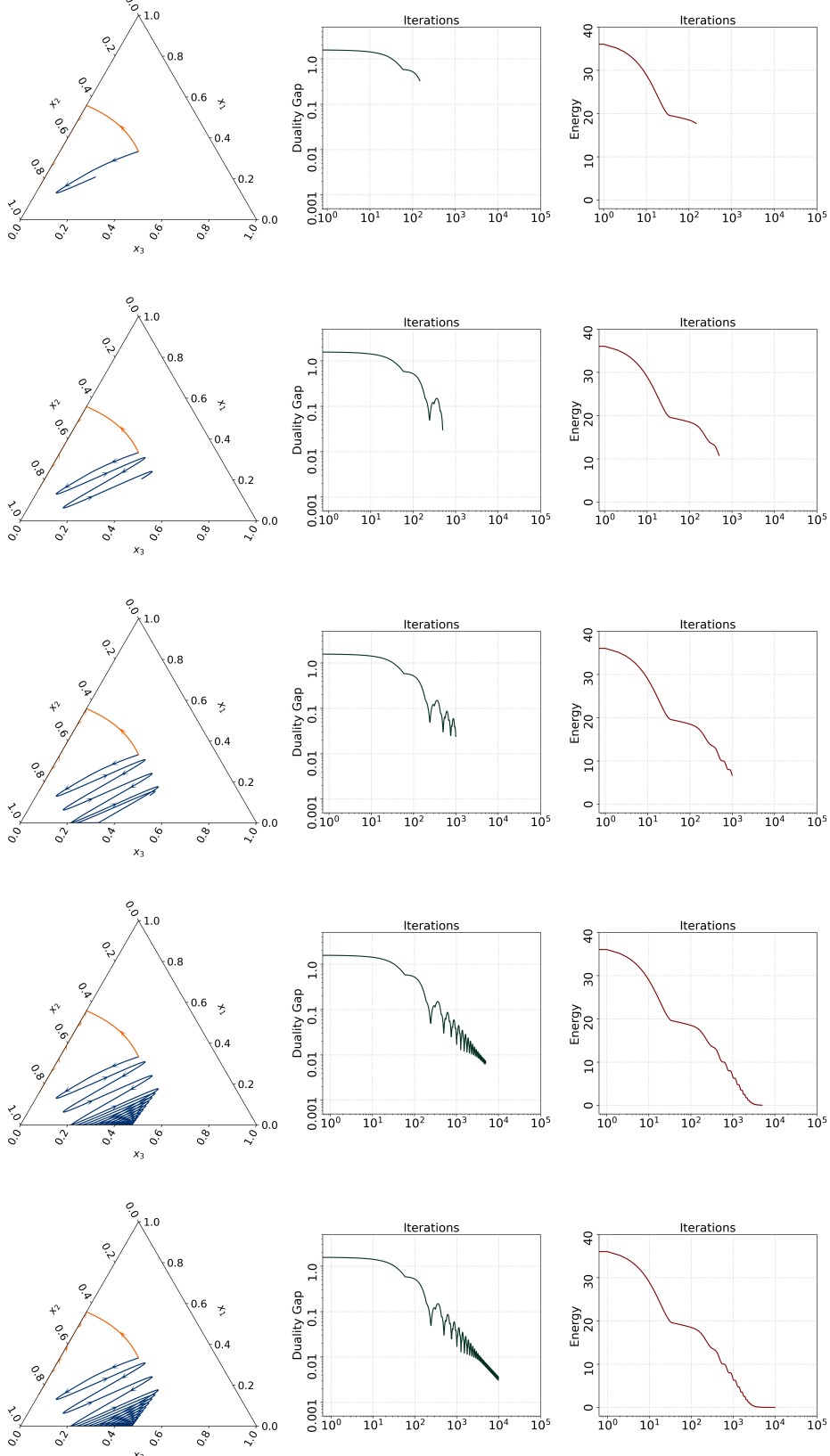

Figure 11: Evolution of Fig. 3

