# OpenReview forum: "On the $O(1/T)$ Convergence of Alternating Gradient Descent–Ascent in Bilinear Games"
_ICLR.cc/2026/Conference — ICLR 2026 Poster_

### Official Review · Reviewer_MTEk · 2025-10-26

**Soundness:** 3
**Presentation:** 3
**Contribution:** 3
**Rating:** 8
**Confidence:** 3

**Summary:**

This paper studies the problem of the average convergence rate of online gradient descent-ascent in bilinear zero-sum games. The main results include:
* A computer-assisted framework to find the optimized step sizes for a given time horizon.
* A proof of the $O(1/T)$ average convergence rate of online gradient descent-ascent in constrained bilinear zero-sum games with interior equilibrium. For games without interior equilibrium, a local result is also provided.

Numerical results are also provided to support the theoretical findings.

**Strengths:**

* The purposed PEP framework provide new tools to proof the convergence rate of learning algorithms in games.
* The result of $O(1/T)$ convergence rate of online gradient descent-ascend for constrained zero-sum games greatly improves the results of (Bailey et al., 2020) in the unconstrained setting.
* The proof techniques, especially the construction of the decayed energy function and its relation to the duality gap, are interesting.
* The paper is well-written and easy to follow, and the motivation and the strategy of proofs are clearly stated.
* Numerical results coincide well with the theoretical findings.

**Weaknesses:**

One weakness I see is that the PEP part (Section 4) is a bit rough. For example, a crucial point of how to reduce the infinite-dimensional nonconvex optimization problem INNER to a solvable finite-dimensional problem is not clearly stated in the main text.

Moreover, I think two recent works of (Feng et al., 2025), which studied the seperation between alternating and simultaneous momentum dynamics in zero-sum games, and (Hait et al., 2025), which studied the regret properties of alternating learning dynamics in convex games, are also relevant to this work. It would be good if the authors could include them in the related work section.

A minor problem: In Figure 1, the label of the x-axis should be T instead of N.

Reference:
1. Feng et al., Continuous-Time Analysis of Heavy Ball Momentum in Min-Max Games . ICML 2025
2. Hait et al, Alternating regret for online convex optimization. COLT 2025

**Questions:**

What are the obstacles to extending the current framework to the analysis of alternating Mirror Descent algorithms in zero-sum games (Wibisono et al., 2022)?

---

> ### Author Response · Authors · 2025-11-20
>
> ### **Responses to weaknesses**
>
> > One weakness I see is that the PEP part (Section 4) is a bit rough. For example, a crucial point of how to reduce the infinite-dimensional nonconvex optimization problem INNER to a solvable finite-dimensional problem is not clearly stated in the main text.
>
> We are improving the presentation of the PEP section. In particular, in the revised PDF we will include a paragraph elaborating on the main reduction and add a pointer to the appendix where the full description can be found.
>
> > Moreover, I think two recent works of (Feng et al., 2025), which studied the seperation between alternating and simultaneous momentum dynamics in zero-sum games, and (Hait et al., 2025), which studied the regret properties of alternating learning dynamics in convex games, are also relevant to this work. It would be good if the authors could include them in the related work section.
>
> Thank you for pointing out these related work.
> We will make sure to include them in the revised version of this work.
>
> > A minor problem: In Figure 1, the label of the x-axis should be T instead of N.
>
> Thanks, we will fix this.
>
> ### **Responses to questions**
>
> > What are the obstacles to extending the current framework to the analysis of alternating Mirror Descent algorithms in zero-sum games (Wibisono et al., 2022)?
>
> Thank you for this question.
> Indeed, one can think of extending our analysis to alternating mirror descent algorithms with a Bregman divergence of a Legendre function.
> However, the analysis is not quite analogous to our $\ell_2$ norm setting.
> The main difference is as follows:
> for AltGDA, we must handle the nonsmooth terms that arise when the trajectories reach the boundary of the feasible set.
> In contrast, with a Legendre function, one avoids such nonsmoothness but instead need to deal with the "Bregman commutator" defined in (Wibisono et al., 2022).
> That is, the convergence analysis for alternating mirror descent algorithms with a Legendre function does not really parallel with AltGDA (and AltGDA is not necessarily the simpler case).
> Though, we think it is an interesting future direction to apply our current framework to study other alternating mirror descent algorithms.

---

### Official Review · Reviewer_xbjq · 2025-10-30

**Soundness:** 3
**Presentation:** 3
**Contribution:** 3
**Rating:** 8
**Confidence:** 4

**Summary:**

The paper studies alternating gradient descent ascent for bilinear games and provides three results, a rate of $O(1/T)$ for average convergence in games with interior Nash Equilibria, a local convergence of the same rate without the aforementioned restriction and a reformulation of the method under a performance estimation framework that allows the computation of an optimized stepsize. Regarding the first two contributions, the authors identify a non-decreasing energy function associated with the duality gap, that facilitates the analysis. The last contribution requires a series of transformations to reduce the original problem to a tractable form.

**Strengths:**

The paper addresses an important open problem and makes substantial progress towards resolving it. The approach for deriving the convergence rate may be based on a simple energy function but is quite technical and involved. The same holds true regarding the PEP framework. I also found the presentation clear.

**Weaknesses:**

The only weakness I can point to is that the existence of an interior NE is not a very weak assumption, like say the uniqueness of a NE.

**Questions:**

Could the authors elaborate on any approaches that they perhaps try but fail to generalize the result completely?

I will also add some minor comments:
In Lemma 8, you can conclude the monotonicity of the energy function without using the fact that the NE lies in the interior, since the third and fourth terms of equation 20 are jointly nonnegative.

In line 1418, “Lemma 12” instead of “12”.

---

> ### Author Response · Authors · 2025-11-20
>
> ### **Responses to questions**
>
> > Could the authors elaborate on any approaches that they perhaps try but fail to generalize the result completely?
>
> Yes, we did try a few approaches to generalize the results.
> The main bottleneck here is the increase of the energy function in the general case.
> Ideally, one can generalize the result by showing that the accumulated 'energy increase' is summable (i.e., prove that the sum of this energy increase is finite as $T$ goes to infinity).
> One possibility is to show that the number of times the trajectory `hits' the boundary of the feasible region is finite, or that the energy function decreases in an amortized sense.
> However, we found it very tricky to attain such results due to the nonsmoothness at the boundary and the resulting non-monotonicity.
>
> We also attempted to use the PEP framework to establish the
> $O(1/T)$ rate, which requires constructing dual variables that generalize to arbitrary $T$ while retaining (near)
> $O(1/T)$ decay of the dual objective.
> However, based on the patterns observed for small $T$, the required dual variables appear to be highly complex, making it difficult to infer any general closed-form structure.
>
> > I will also add some minor comments: In Lemma 8, you can conclude the monotonicity of the energy function without using the fact that the NE lies in the interior, since the third and fourth terms of equation 20 are jointly nonnegative.
>
> Thanks for this comment.
> As you pointed out, the interior NE condition is not required to derive the expression following Eq. (20).
> When an interior NE is not assumed, this expression can instead be written as an inequality.
> Though, to establish the monotonicity of the energy function, we still rely on the second item of Lemma 7, which requires an interior NE.
> We will revise this part of the proof to clarify this point.
>
> > In line 1418, “Lemma 12” instead of “12”.
>
> Thanks. We fixed it.

---

> > ### Comment · Reviewer_xbjq · 2025-11-26
> >
> > I thank the authors for responding to my question and comments. I have also read the other reviews and the authors' answers and I have no further questions or comments at this point of the discussion. I maintain my positive opinion about the paper.

---

> > > ### Author Response · Authors · 2025-12-04
> > >
> > > Thank you!

---

### Official Review · Reviewer_ppTd · 2025-10-31

**Soundness:** 3
**Presentation:** 3
**Contribution:** 3
**Rating:** 6
**Confidence:** 3

**Summary:**

The paper studies Alternating Gradient Descent–Ascent (AltGDA) in two-player zero-sum constrained bilinear games. While previous works have shown an empirical faster rate of AltGDA and some improved $O(T^{-2/3})$ convergence rate, this paper proves an $O(1/T)$ convergence rate when an **interior** Nash equilibrium (NE) exists, using a constant stepsize dependent on the value of the interior NE. Specifically, the sufficient condition is to have the stepsize no larger than the smallest non-zero equilibrium mass. It contrasts this with simultaneous GDA’s typical $O(1/\sqrt{T})$ behavior. The paper also shows a local $O(1/T)$ rate for non-interior games under a universal constant stepsize, with the bound expressed via a gap parameter linked to a maximal-support NE. Finally, a performance-estimation-programming (PEP) SDP is used to numerically optimize stepsizes and visualize a near $O(1/T)$ curve for finite $T$.

**Strengths:**

- The paper is well motivated by the empirical success of AltGDA and establishes the first $O(1/T)$ rate for AltGDA with constraints under an interior NE. This is a clear improvement over simultaneous GDA while using a constant stepsize. The proof leverages a monotone energy decay, which is neat conceptually and interesting in its own right.
- The use of the PEP framework is also interesting to show the benefit of using alternation. The SDP study provides tangible (though with approximations) finite-time evidence for the benefits of alternation and suggests structured.

**Weaknesses:**

- One main weakness is that even under the assumption that there exists an interior equilibrium, the stepsize $\eta$ depends on the unknown equilibrium masses. Specifically, the global $O(1/T)$ theorem (interior NE $(x,y)$) requires a stepsize scaled by $\min( \min_i x_i, \min_j y_j )$, which are unknown in practice and may be extremely small. No adaptive rule is provided that attains the same rate without such knowledge.
- In addition, if the interior NE is near the boundary, the admissible constant stepsize can be impractically tiny. There is no quantification of worst-case constants or a detection-and-retuning mechanism, and this will also leads to a bad convergence guarantee.
- The other concern is that the global analysis relies on the existence of an interior NE. When no interior NE exists, the same energy monotonicity fails; the paper does not provide an in-process diagnostic to decide which regime one is in.
- For the second result with local convergence rate, while the learning rate now is not dependent on $\min( \min_i x_i, \min_j y_j )$, this requires the initial point to be close to an NE, with the distance also dependent on these problem-dependent constants.

**Questions:**

- Is there an adaptive stepsize rule (e.g., backtracking based on duality-gap trends or energy-surrogate monotonicity) that achieves the same $O(1/T)$ rate when an interior NE exists, without using $\min\{\min_i x_i^*, \min_j y_j^*\}$ a priori?
- I wonder whether there is some way to detect the existence of an interior NE in an online fashion. For example, can persistent monotone decrease of the energy and support persistence be shown to imply interior-NE structure (or neighborhood), while non-monotonic energy indicates the alternative regime?
- Can the aurthors explain more on the choice of the learning rate $\eta$. Is that possible to bound the largest admissible constant stepsize using spectral/simplex geometry of $A$ alone, or give conditions under which $\min( \min_i x_i, \min_j y_j )$ are bounded away from zero?
- Can this analysis be generalized to two-player general-sum game?

---

> ### Author Response · Authors · 2025-11-20
>
> ### **Responses to questions**
>
> > Is there an adaptive stepsize rule (e.g., backtracking based on duality-gap trends or energy-surrogate monotonicity) that achieves the same $O(1/T)$ rate when an interior NE exists, without using $\min{\min_i x_i^\star, \min_j y_j^\star}$ a priori?
>
> Thanks for this question and the suggestions.
> Please see our response to Reviewer uo6Y's first question.
> We will also add a subsection in the revised PDF for this.
>
> > I wonder whether there is some way to detect the existence of an interior NE in an online fashion. For example, can persistent monotone decrease of the energy and support persistence be shown to imply interior-NE structure (or neighborhood), while non-monotonic energy indicates the alternative regime?
>
> Thanks for this comment and the suggestions.
> We think it is possible to detect whether there is an interior NE or non-interior NE based on the trajectory's behaviors.
> We elaborate one useful point here:
> with a sufficiently small constant stepsize, the game cannot have any non-interior NE if the trajectory is cycling within the interior of the probability simplex like the one (after reaching a steady state) in the left plot of Figure 2.
> This is because the energy function w.r.t. any non-interior NE is strictly decreasing (one can see this point from Lemma 8 and Reviewer xbjq's comment) if the trajectory no longer hits the boundary.
> As the energy value only depends on the position of an iterate, if the trajectory revisits one point (which happens in the cycling), then by contradiction we can say that there is no non-interior NE in the game.
>
> > Can the aurthors explain more on the choice of the learning rate $\eta$. Is that possible to bound the largest admissible constant stepsize using spectral/simplex geometry of $A$ alone, or give conditions under which $\min( \min_i x_i, \min_j y_j )$ are bounded away from zero?
>
> For general two-player zero-sum games, to the best of our knowledge, there is no simple rule that bounds the constant step size away from zero using properties of $A$ alone.
> Nevertheless, there exist some well-studied classes of games for which specific bounds are attainable.
> For example, it is known that every harmonic game admits a uniformly mixed equilibria [1].
> In this case, $\frac{1}{\lVert A \rVert_2} \min( \frac{1}{n}, \frac{1}{m} )$ is an admissible constant stepsize.
> This is a good point.
> We will add a paragraph to give examples in the revised paper.
>
> [1] Candogan, Ozan, et al. "Flows and decompositions of games: Harmonic and potential games." Mathematics of Operations Research 36.3 (2011): 474-503.
>
> > Can this analysis be generalized to two-player general-sum game?
>
> Although it is interesting to generalize this analysis to a broader class of games, we remark that one should not expect an analogous $O(1/T)$ convergence rate in two-player general-sum games.
> In particular, [Rub16] ruled out the possibility of computing an
> $\varepsilon$-Nash equilibrium in two-player general-sum games within a
> $\textnormal{poly}(m,n,\frac{1}{\varepsilon})$ running time, where $m$ and $n$ are input sizes (the sizes of the strategy spaces of the two players) of the games.
>
> [Rub16] Rubinstein, Aviad. "Settling the complexity of computing approximate two-player Nash equilibria." ACM SIGecom Exchanges 15.2 (2017): 45-49.

---

> ### Comment · Reviewer_ppTd · 2025-11-27
>
> Thanks for the detailed responses in the rebuttal. My questions are well-addressed in general and I keep my current positive score.

---

> > ### Author Response · Authors · 2025-12-04
> >
> > Thank you!

---

### Official Review · Reviewer_o6a6 · 2025-11-01

**Soundness:** 2
**Presentation:** 2
**Contribution:** 1
**Rating:** 2
**Confidence:** 4

**Summary:**

This paper studies the AltGDA algorithm in the constrained min-max setting. It gives a convergence rate $O(1/T)$ under interior NE assumptions.

**Strengths:**

The paper gives $O(1/T)$ convergence for the AltGDA algorithm under interior NE assumptions in the constrained case.

**Weaknesses:**

The algorithm is known, so the main claimed novelty is the analysis. Lemma 2 is the key new piece, which follows once interior NE existence is assumed. The result and proof are straightforward, leading the contribution to appear incremental since Theorem 1 follows almost immediately.

The authors fail to sufficiently discuss long-known theoretical results of $O(1/T)$ for the constrained min-max setting, with only a brief mention of optimistic methods, though there is a rich line of literature here (e.g. [1-3]), nearly all of which has been ignored by the authors.

The authors' results hold only for games with an interior Nash equilibrium, or else for local convergence. Previous first-order algorithms achieve $O(1/T)$ rates without these conditions. The authors should explain this point better and clarify how their first-order results are weaker than previous first-order results.

[1] Korpelevich, Galina M. "The extragradient method for finding saddle points and other problems." Matecon 12 (1976): 747-756.

[2] Nemirovski, Arkadi. "Prox-method with rate of convergence O (1/t) for variational inequalities with Lipschitz continuous monotone operators and smooth convex-concave saddle point problems." SIAM Journal on Optimization 15, no. 1 (2004): 229-251.

[3] Nesterov, Yurii. "Dual extrapolation and its applications to solving variational inequalities and related problems." Mathematical Programming 109, no. 2 (2007): 319-344.

**Questions:**

Can the authors say more about how their analysis differs from the analysis in Bailey et al. (2020)?

Can the authors explain what they believe is the purpose of analyzing alternating first-order methods, since optimal first-order methods have already been known for several decades?

Can the authors explain what they believe to be the non-trivial parts of their analysis?

---

> ### Author Response · Authors · 2025-11-20
>
> ### **Responses to weaknesses**
>
> > The algorithm is known, so the main claimed novelty is the analysis. Lemma 2 is the key new piece, which follows once interior NE existence is assumed. The result and proof are straightforward, leading the contribution to appear incremental since Theorem 1 follows almost immediately.
>
> The alternating GDA algorithm we studied is famous and very useful in practice, which makes understanding its convergence behavior theoretically particularly appealing.
> Over the years, a line of work has focused on developing a progressively better theoretical understanding of the “alternation” trick.
> Yet, we still do not know any convergence rate better than $O(1/\sqrt{T})$ in the basic matrix game setting (i.e., constrained bilinear minimax problems).
> In this context, our paper establishes the first set of $O(1/T)$ convergence results in the literature, and also, our analysis indicates a possible approach to further study the convergence of alternating dynamics.
> Other reviewers have raised insightful questions for future work related to our proof strategy.
> As such, it is unfair to claim that the results and proofs are straightforward.
> For instance, Lemma 2 relies on Lemmas 7 and 8 in the appendix, which are based on new geometric insights and arguments.
>
> > The authors fail to sufficiently discuss long-known theoretical results of $O(1/T)$ for the constrained min-max setting, with only a brief mention of optimistic methods, though there is a rich line of literature here (e.g. [1-3]), nearly all of which has been ignored by the authors.
> The authors' results hold only for games with an interior Nash equilibrium, or else for local convergence. Previous first-order algorithms achieve $O(1/T)$ rates without these conditions. The authors should explain this point better and clarify how their first-order results are weaker than previous first-order results.
>
> We would like to highlight that the main focus of this work is to study alternating dynamics in constrained minimax/game settings, which are commonly used as go-to algorithms in many applications.
> Achieving a $O(1/T)$ convergence rate in constrained problems remained an open problem prior to our paper, and is now open beyond the "interior NE" setting.
> We do not aim to propose an algorithm with strictly better theoretical guarantees.
>
> Regarding "optimistic methods," (i.e. proximal point methods, extragradient methods, optimistic GDA, etc): we are well-aware that these methods achieve a $O(1/T)$ rate. This is not really the point of the alternation line of literature though. The point is to understanding the power of the alternation trick. You can see this in earlier papers in this line of literature as well.
> We agree that the contextualization may have been too brief though; we provide additional context on this line of literature in the revised version.

---

> > ### Comment · Reviewer_o6a6 · 2025-11-25
> >
> > I appreciate the authors' response, and have updated my score. I still strongly recommend that the authors provide a more thorough discussion of the existing literature, and to emphasize the limitations of their work relative to these previous results.

---

> > > ### Author Response · Authors · 2025-12-04
> > >
> > > Thank you for the suggestions and the score updates! In the revised PDF, we have expanded the discussion of existing extragradient and optimistic-type algorithms and emphasized that AltGDA is studied as an appealing algorithmic choice that is widely used in practice.

---

> ### Author Response · Authors · 2025-11-20
>
> ### **Responses to questions**
>
> > Can the authors say more about how their analysis differs from the analysis in Bailey et al. (2020)?
>
> In Bailey et al. (2020),
> the per-iteration regret term can be decomposed into the difference of two consecutive squared norms (as shown in their Lemma 1), and the finite bound then follows by telescoping.
> In contrast,
> our paper handles nonsmoothness, which causes the additional positive residual terms in the decomposition.
> As a result, direct telescoping yields nonnegative terms that grow with $T$, creating a major obstacle to obtaining an $O(1/T)$ convergence rate.
>
> At a high level, our problem lies in the domain of nonsmooth optimization, which is fundamentally different from—and generally more challenging than—the smooth setting.
>
> > Can the authors explain what they believe is the purpose of analyzing alternating first-order methods, since optimal first-order methods have already been known for several decades?
>
> The study of alternating dynamics is motivated by the strong practical performance of learning dynamics that utilize alternation (see, e.g., [1,2]).
> For example, if you look at the extensive-form game solving literature, you will see that learning dynamics based on alternation (often with no optimism-type modification such as those in optimistic GDA, extragradient methods, etc.) are among the preferred methods in practice [3].
> Notably, the algorithm used to create superhuman poker AIs was an alternating non-optimistic algorithm [4,5].
> Similarly, the strong practical performance of alternation has been highlighted in the unconstrained setting [1,2].
> This is why earlier papers such as [1,2,6] study alternation without optimism.
>
>
> > Can the authors explain what they believe to be the non-trivial parts of their analysis?
>
> Firstly, please note from the list of citations in our answer to your preceding question that alternation has been studied by a meaningful number of researchers over the last five years (theoretically) and the last 15 years (experimentally).
> Even so, our result is the first result to show an $O(1/T)$ rate for any non-optimistic alternating method (including alternating mirror descent algorithms); the strongest prior result was $O(1/T^{5/6})$ [6].
> Even if you think that our analysis is trivial, it is clearly not a well-understood problem, and in fact a simple analysis would be quite surprising considering that no prior paper had managed to achieve an $O(1/T)$ convergence rate.
>
> Secondly, we do not agree that the analysis is simple.
> Here, we point out several non-standard components:
> - As we mentioned before, a major obstacle to obtaining an $O(1/T)$ convergence rate is the presence of the positive residual terms caused by the non-smoothness of the problem.
>     This issue is absent in most of the existing literature on alternating dynamics, which typically considers either the unconstrained case or Legendre functions (therefore the trajectories never hit the boundary of the feasible region).
>     The idea of wrapping up the sum of the residual terms by an energy function (which has meaningful interpretation in the experiments) is new and not direct to see, since the residual terms are not the consecutive differences of that energy function.
>     Furthermore, even assuming an interior Nash equilibrium, it is not straightforward to bound the residual terms by the difference of the energy function (see Lemmas 7 and 8).
> - To derive symmetric representations of the residual terms, one must consider and sum two descent inequalities corresponding to two consecutive iterates. This is also non-standard in the analysis of convex minimization; usually a single descent inequality is summed.
> - The construction and verification of the two local regions are non-trivial: it must be shown that any “energy increase” in the outer local region is bounded, so that the trajectory cannot escape from the inner region to outside the outer region (see Lemma 4 and the lemmas referenced in its proof).
>
> [1] Andre Wibisono, Molei Tao, and Georgios Piliouras. "Alternating mirror descent for constrained min-max games." Advances in Neural Information Processing Systems 35 (2022): 35201-35212.
>
> [2] Guodong Zhang, et al. "Near-optimal local convergence of alternating gradient descent-ascent for minimax optimization." International Conference on Artificial Intelligence and Statistics. PMLR, 2022.
>
> [3] Oskari Tammelin, et al. "Solving Heads-Up Limit Texas Hold'em." IJCAI. Vol. 15. 2015.
>
> [4] Noam Brown and Tuomas Sandholm. Superhuman AI for heads-up no-limit poker: Libratus beats
> top professionals. Science, 359(6374):418–424, 2018.
>
> [5] Noam Brown and Tuomas Sandholm. Superhuman AI for multiplayer poker. Science, 365(6456):
> 885–890, 2019.
>
> [6] Jonas Katona, Xiuyuan Wang, and Andre Wibisono. "A symplectic analysis of alternating mirror descent." arXiv preprint arXiv:2405.03472 (2024).

---

### Official Review · Reviewer_uo6Y · 2025-11-03

**Soundness:** 3
**Presentation:** 3
**Contribution:** 2
**Rating:** 6
**Confidence:** 3

**Summary:**

This paper studies alternating gradient descent–ascent with Euclidean projections in two-player zero-sum bilinear games over convex compact sets. They focus on constrained matrix games where alternation is widely used in practice but poorly understood theoretically. The authors first show that when the game admits an interior Nash equilibrium, AltGDA with a suitably small constant stepsize enjoys an ergodic  $\mathcal{O}(1/T)$ convergence rate in duality gap. Giving the first result where plain alternation provably improves over simultaneous GDA, they prove a local $\mathcal{O}(1/T)$ convergence result for general bilinear games. Beyond these analytic results, the paper develops a performance estimation problem (PEP) formulation that encodes AltGDA’s worst-case behavior as an SDP and numerically optimizes stepsizes. The PEP evidence suggests AltGDA may in fact achieve an $\mathcal{O}(1/T)$ rate more broadly for finite horizons, while simultaneous GDA appears limited to $\mathcal{O}(1/\sqrt{T})$, and experiments on random matrix games corroborate the faster empirical convergence of AltGDA.

**Strengths:**

* The combination of a global result under an interior NE and a local result around a maximal-support NE gives a fairly complete picture.
* The energy termed $\mathcal{E}$ used to capture the two-phase dynamics (boundary hits vs. interior cycling) and to bound residual terms is conceptually interesting.
* Experiments verify the theoretical results.

**Weaknesses:**

* The global convergence rate requires an interior NE, which is quite restrictive in game applications.
* The PEP-based evidence for global convergence in general convex compact sets is compelling but not a proof.
* The analysis assumes deterministic gradients. Many game-solving methods operate with sampling or bandit feedback, it is unclear whether the energy-based analysis or PEP insights extend to noisy gradients.

**Questions:**

* In Theorem 1, $\eta$ depends on $\min x_i, \min y_i$, which are unknown. Can you provide data-driven or adaptive rules that maintain the $\mathcal{O}(1/T)$ rate?
* Are there lower bounds showing that (a) AltGDA cannot do better than $\mathcal{O}(1/T)$ in this setting, or (b) the constants in Theorems 1–2 are not too pessimistic?
* Which parts of your proof crucially use bilinearity? Could the energy-based argument or the Lemma 1–2 telescoping scheme extend to smooth convex–concave problems with Lipschitz gradients but non-linear coupling?

---

> ### Author Response · Authors · 2025-11-20
>
> ### **Responses to weaknesses**
>
> > The PEP-based evidence for global convergence in general convex compact sets is compelling but not a proof.
>
> Indeed, the PEP-based evidence for global convergence in general convex compact sets does not establish a convergence rate of $O(1/T)$ for generic $T$. However, for the finite horizons ranging $T=1,2,\ldots,50$, the tightness of the convergence bounds computed numerically can be proven
> using the numerical dual multipliers associated with the final semidefinite optimization problem (Problem (63) in the Appendix D).
> For more details on constructing a valid proof via PEP,
> we refer to [1].
>
> [1] Baptiste Goujaud, Aymeric Dieuleveut, and Adrien Taylor. ``On fundamental proof structures in first-order optimization.'' In 2023 62nd IEEE Conference on Decision and Control (CDC), pp. 3023-3030. IEEE, 2023.
>
> > The analysis assumes deterministic gradients. Many game-solving methods operate with sampling or bandit feedback, it is unclear whether the energy-based analysis or PEP insights extend to noisy gradients.
>
> While we agree that it is interesting to study the stochastic case for the AltGDA dynamics, we would like to highlight that there is an extensive literature on studying the deterministic case of learning dynamics  (in our view significantly more extensive than the stochastic literature).
> The stochastic case is very interesting, but results tend to be significantly weaker.

---

> ### Author Response · Authors · 2025-11-20
>
> ### **Responses to questions**
>
> > In Theorem 1, $\eta$ depends on $\min_i x_i, \min_j y_j$, which are unknown. Can you provide data-driven or adaptive rules that maintain the $\mathcal{O}(1/T)$ rate?
>
> Yes, we can design some adaptive stepsize rule that searches for an admissible stepsize, or even leads to better performance in practice.
>
> Perhaps the most simple approach is to employ a monotonically decreasing stepsize update rule.
> We brief describe this approach here, and will include a full description in the revised PDF for this.
> Note that, if we are using an admissible stepsize, i.e., a stepsize $\eta \leq \eta^\star := \frac{1}{\lVert A \rVert_2}\min\{ \min_{i} x^\star_{i}, \max_{j} y^\star_{j} \}$ for some interior NE $(x^\star, y^\star)$, by Lemma 2, the residual terms $r_t$ is summable and the sum of them is at most $\mathcal{E}(x^{k, 0}, y^{k, 0}) \leq 8 + \eta^0 \lVert A \rVert_2$.
> With this fact in hand, we can design the following stepsize update rule:
> - Starting with some $\eta^0 > 0$, we track $r_t$ over time and decrease the stepsize by $\frac{1}{2}$ when the sum of $r_t$ exceeds $8 + \eta^0 \lVert A \rVert_2$ (and after stepsize update we restart the counting of $r_t$).
>
> In this way, we can find an admissible $\hat{\eta}$ in (round($\log_2(\eta^0/\eta^\star)$) + 1) epochs.
> Within each epoch, we use a constant step size $\eta^k \geq \hat{\eta}$, by Eqs.(3-4) and the definition of $r_t$, we have the sum of Eq.(3) + Eq.(4) over all iterations within the epoch provides an upper bound for $\hat{\eta} \sum_{t=0}^{T^k} ( y^\top A x^t - (y^t)^\top A x )$ that is independent of $T^k$.
> After we find the $\hat{\eta}$ after (round($\log_2(\eta^0/\eta^\star)$) + 1) epochs, the regret will remain finitely bounded by Theorem 1.
> Summing $\hat{\eta} \sum_{t=0}^{T^k} ( y^\top A x^t - (y^t)^\top A x )$ over all epochs yields an $O(1/T)$ convergence rate.
>
> > Are there lower bounds showing that (a) AltGDA cannot do better than $\mathcal{O}(1/T)$ in this setting, or (b) the constants in Theorems 1–2 are not too pessimistic?
>
> To the best of our knowledge, there are no lower bounds for the duality gap of the averaged iterate of AltGDA in the two-player zero-sum game setting.
> We remark that the numerical experiment results in Section 7 matches the $O(1/T)$ convergence rate, indicating this convergence rate might be tight.
> Moreover, as existing fast first-order methods (e.g., optimistic GDA, EG) have an $O(1/T)$ convergence rate-it would be very surprising if AltGDA can reach an even better rate.
>
> We also remark that it appears unlikely that a bound better than $O(1/T)$ exists:
> First, the last iterate is cycling, so proving the convergence rate in the last iterate duality gap is not possible.
> Second, because the denominator in the uniform average is on the order of $T$, we cannot average out a bad strategy at a rate faster than $O(1/T)$. So, it is at least not possible to do better than $O(1/T)$ without moving to some averaging scheme that is in-between uniform averaging and last iterate.
>
> The magnitude of the constants in Theorems 1-2 are mainly depending the stepsize $\eta$.
> We added plots to the revised PDF to indicate the dependence of $\eta$ is tight in the experiments.
> To derive a better constant,
> we need an improved stepsize choices, which can be an interesting future direction.
>
> > Which parts of your proof crucially use bilinearity? Could the energy-based argument or the Lemma 1–2 telescoping scheme extend to smooth convex–concave problems with Lipschitz gradients but non-linear coupling?
>
> Thanks for this question and the suggestion on extending our proof techniques.
>
> We assume that you are referring to constrained minimax problems with smooth convex–concave payoff functions,
> because truly “smooth” problems should not involve constraints, as the constraints introduce nonsmoothness.
> Moreover, AltGDA in the unconstrained minimax setting has already been studied extensively in the literature.
>
> Our current analysis is conducted based on our bilinear setting,
> which is arguably the most important game setting.
> In particular, to construct our new energy functions and connect them to the convergence analysis, bilinearity is used extensively.
> While it would be interesting to explore whether our techniques and insights extend to the more general convex–concave setting,
> it is not immediately clear how to do so.

---

### Author Response · Authors · 2025-11-21

We thank all reviewers for their valuable feedback. Our responses are provided in the corresponding official comments.

---

> ### Author Response · Authors · 2025-12-04
>
> [duplicated comments]

---

### Author Response · Authors · 2025-12-04

**We thank all reviewers again for their suggestions and feedback! We have incorporated their comments as well as our additional results into the revised paper, with changes highlighted in blue (in the main text). After the initial rebuttal period, reviewer o6a6 was convinced by our discussion and substantially increased their score, as can be seen from our interaction with them.**

---

### Meta-Review · Area_Chair_2aqZ · 2026-01-17

**Summary:**

This work analyzes alternating gradient descent-ascent (AltGDA) for two-player zero sum games. AltGDA has gained significant recognition recently. This work improves the rate of convergence for this method to O(1/T)  when there is an interior solution to the problem. Indeed, the interior solution assumption makes the problem closer to the unconstrained problem which is analyzed before by Bailey et al., 2020. Still, all the reviewers appreciated this fundamental advance which will hopefully lead to a more general result in the future even when the solution lies on the boundary which is a more interesting case for constrained problems.

**Reviewer Concerns:**

Most of the reviewers argued for acceptance from the beginning except Reviewer o6a6. The authors addressed the concerns of the reviewer on the difficulty of the analysis as well as comparison with optimistic methods. Authors explain that AltGDA receives interest despite the existence of optimistic method and that their analysis has additional technical tools adding to the literature, to handle issues coming from having constraints.

**Reviewer Scores:**

I think that Reviewer o6a6 would have increased their score since their concerns with optimistic methods and technical novelty are addressed in detail by the authors. All the other reviewers voted for acceptance with minor concerns that do not constitute any reason for rejecting the paper.

---

### Decision · Program_Chairs · 2026-01-26

Accept (Poster)